# Influence-Guided Symbolic Regression: Scientific Discovery via LLM-Driven Equation Search with Granular Feedback

Evgeny S. Saveliev [* 1]   Samuel Holt [* 1]   Nabeel Seedat [2 1]   David L. Bentley [3]   Jim Weatherall [4]
Mihaela van der Schaar [1]

## Abstract

Large Language Models (LLMs) offer a promising avenue for scientific discovery, yet their application to symbolic regression is often constrained by inefficient search strategies and coarse feedback signals. Current methods typically guide LLMs using scalar metrics (e.g., global Mean Squared Error), which fail to identify which components of a proposed equation are driving performance or causing error. We introduce *Influence-Guided Symbolic Regression* (IGSR), a method that frames equation discovery as an iterative two-step process combining diverse term generation with rigorous selection: an LLM generates candidate basis functions $\psi_j(\mathbf{x})$ for a linear model, which are then evaluated using granular influence scores $\Delta_j$. These scores quantify each term's marginal contribution to generalization accuracy, enabling an influence-guided pruning process that systematically refines the model structure. Integrating this mechanism into a Monte Carlo Tree Search (MCTS) enables navigating the combinatorial search space while balancing exploration of novel functional forms with exploitation of high-influence components. We demonstrate IGSR's effectiveness on a diverse suite of benchmarks, including LLM-SRBench, pharmacological PKPD models, an epidemiological simulation, and real-world genomic data. Notably, we validate the framework's capacity for genuine discovery in a case study using a high-dimensional biological dataset, in which IGSR identified a novel relationship between DNA methylation and RNA Polymerase II pausing; a hypothesis that was subsequently supported via wet-lab experimentation.

---
[*]Equal contribution   [1]University of Cambridge [2]Thomson Reuters Foundational Research [3]U. Colorado, Anschutz Medical Campus [4]AstraZeneca. Correspondence to: Evgeny S. Saveliev <es583@cam.ac.uk>.

*Proceedings of the 43$^{rd}$ International Conference on Machine Learning*, Seoul, South Korea. PMLR 306, 2026. Copyright 2026 by the author(s).

## 1. Introduction

The quest for interpretable and generalizable mathematical models is a cornerstone of scientific advancement. In disciplines ranging from epidemiology to high-energy physics, concise closed-form equations that accurately predict phenomena offer mechanistic insights that black-box approximations cannot provide. This paper addresses the challenge of discovering a model of the form $f(\mathbf{x}) = \sum_j w_j \psi_j(\mathbf{x})$, where $\psi_j(\mathbf{x})$ are basis functions representing potentially nonlinear and complicated transformations of the input features $\mathbf{x}$, and $w_j$ are their corresponding weights. The equation is linear in these (possibly nonlinear) terms, so each retained term carries an explicit, interpretable weight. We therefore target compact closed-form equations rather than the deeply nested or recurrent dynamics of fully differentiable approaches.

Large Language Models (LLMs) have emerged as potent accelerators for this task, leveraging vast pre-trained knowledge to propose plausible functional forms (Ma et al., 2024; Holt et al., 2024). However, existing LLM-driven approaches are often constrained by the coarseness of their feedback signals. Typically, these systems guide the search using a single scalar metric, such as global Mean Squared Error (MSE). While this indicates *if* a proposed equation is accurate, it fails to explain *why*, or *which specific components* are driving or degrading performance. Without granular credit assignment, the discovery process becomes inefficient trial-and-error, relying heavily on the LLM's generative priors rather than data-driven reasoning.

To bridge this gap, we introduce Influence-Guided Symbolic Regression (IGSR). Our key insight is that the successful construction of symbolic models requires decoupling the *generation* of diverse functional hypotheses from the *selection* of valid components. IGSR operationalizes this by providing a mechanism for granular, *per-term influence feedback*. In our framework, an LLM agent proposes candidate basis functions $\psi_j(\mathbf{x})$. Rather than evaluating the equation as a monolithic whole, we compute influence scores $\Delta_j$ for each term, quantifying its marginal contribution to generalization accuracy.

*Table 1.* **Comparison of Modeling Paradigms.** We contrast IGSR with Classical Symbolic Regression (SR), Black-Box methods, Automated Feature Engineering (AFE), and other LLM-based equation discovery tools. *Key Differences:* Unlike AFE, which typically augments data for downstream black-box models, IGSR targets a globally interpretable equation. Unlike other LLM approaches that rely on coarse global loss, IGSR uses granular influence scores to navigate the search space. See also extended comparison in Appendix A.8.

| Method Paradigm | High Dim ($d \gg 20$) | Feature Generation ($\psi_j$) | Feedback Signal | Output Structure & Interpretability |
|---|---|---|---|---|
| Classical Symbolic Reg. (e.g., GP-SR) [1] | Poorly | Evolutionary / Genetic | Scalar Loss | ✓ Closed-Form Eq. |
| Black-Box ML (NN, GBDT) [2] | ✓ | Implicit / Learned | Scalar Loss / Gradients | ✗ Opaque Predictor |
| Neural ODEs [3] | ✓ | Implicit | Scalar Loss / Gradients | ✗ Black-Box Dynamics |
| Automated Feat. Eng. (AFE) [4] | ✓ | Predefined Operators | Loss / Feat. Imp. | ✗ Augmented Dataset[†] |
| LLM for Equations (e.g., D3, LLM-SR) [5] | Limited | LLM (code/text) | Loss / Code Errors | ✓ Closed-Form Eq. / ODE |
| **IGSR** | ✓[‡] | **LLM Proposes $\psi_j$** | **Per-term Influence ($\Delta_j$)** | ✓ **Sparse Linear Eq. ($\sum w_j \psi_j$)** |

**References:** [1] Stephens (2015); Cranmer (2023), [2] Chen & Guestrin (2016); Gorishniy et al. (2021), [3] Chen et al. (2018), [4] Horn et al. (2019); Zhang et al. (2023), [5] Holt et al. (2024); Shojaee et al. (2025a)
**Abbreviations:** GP (Genetic Programming), GBDT (Gradient Boosted Decision Trees), ODE (Ordinary Differential Equation).
† AFE typically outputs a transformed feature matrix optimized for tree ensembles; exact interpretability depends on the downstream model.
‡ IGSR handles high dimensionality by using the LLM to *select* and *prune* relevant features from the large pool, rather than using all simultaneously.

This detailed feedback drives a rigorous *propose-and-prune* cycle. While this cycle can support an agentic mode (where a second LLM interprets the scores to make semantic pruning decisions) we find that a *deterministic pruning* strategy based strictly on $\Delta_j$ provides a robust, hallucination-free default that efficiently identifies the system's underlying mechanisms. By informing the generative agent of which specific terms were retained or discarded in previous iterations, the system fosters an iterative refinement loop that progressively converges on the optimal structure.

To navigate the combinatorial landscape of potential equations effectively, we integrate this cycle into a *Monte Carlo Tree Search (MCTS)* framework. While iterative refinement can improve a single hypothesis, it is prone to local optima. MCTS allows for the systematic exploration of the solution space, balancing the *exploitation* of high-influence terms with the *exploration* of novel functional branches (Kocsis & Szepesvári, 2006; Silver et al., 2016).

We validate IGSR through a rigorous empirical suite. On top of a diverse set of real and synthetic biological/clinical datasets, we evaluate the system on *LLM-SRBench* (Shojaee et al., 2025b), a large, recent benchmark designed to mitigate memorization and test genuine discovery capabilities. Furthermore, we demonstrate the real-world scientific utility of IGSR through a case study on RNA Polymerase II pausing. In this domain, IGSR not only recovered known biological mechanisms but generated a novel hypothesis regarding the role of DNA methylation, which was subsequently supported via a targeted wet-lab experiment involving cell treatment and sequencing

**Contributions**:
① **Conceptual Innovation**: We reframe LLM-driven symbolic discovery by introducing *per-term influence scores* as a mechanism for granular credit assignment. This shifts the paradigm from optimizing global scalar loss to refining specific structural components, bridging the gap between generative LLM priors and rigorous statistical selection.
② **Methodological Framework**: We propose a framework, IGSR, that integrates LLM-driven term generation with influence-guided pruning. We implement this within a Monte Carlo Tree Search (MCTS) to enable efficient exploration of the equation space.
③ **Empirical Validation**: We demonstrate that IGSR, on average, outperforms baselines on a set of six biological/clinical datasets, as well as on a comprehensive *LLM-SRBench* evaluation. Crucially, we validate IGSR's capacity for genuine discovery by its identifying a novel biological relationship in genomic data, which we then support via a prospective wet-lab experiment.

We situate IGSR within the landscape of relevant modeling paradigms in Table 1.

## 2. Methodology

Influence-Guided Symbolic Regression (IGSR)[1] is a framework that leverages the generative priors of Large Language Models (LLMs) to discover interpretable, closed-form mathematical models. We target a class of *sparse, interpretable* models of the form $f(\mathbf{x}) = \sum_{j=1}^{M} w_j \psi_j(\mathbf{x})$, where $\psi_j(\mathbf{x})$ are nonlinear basis functions[2] proposed by the LLM, and $w_j$ are weights optimized via linear regression. Keeping the model linear in the $\psi_j$ preserves global interpretability, while the proposed terms still capture nonlinear structure.

The core innovation of IGSR is the integration of granular, structure-aware feedback into the discovery process. Rather than relying solely on a global scalar loss (e.g., MSE) which obscures the contribution of individual terms, IGSR computes *per-term influence scores*, $\boldsymbol{\Delta}$. These scores quantify the marginal contribution of each basis function to the model's generalization accuracy. This feedback drives a rigorous *propose-and-prune* cycle: an LLM generates candidate terms based on scientific context and history, and

---

[1]IGSR code can be found at: github.com/DrShushen/IGSR and github.com/vanderschaarlab/IGSR.

[2]Standard IGSR treats these basis functions as fixed, we investigate *IGSR-TLO (Term-Local Optimization)* in Appendix G.14, a variant that allows for optimizing parameters *within* terms.

a selection mechanism refines this set by discarding low-influence terms.

In the default *IGSR* configuration, this refinement is performed via *deterministic pruning* based strictly on the influence scores, decoupling the creative generation of the LLM from the data-driven selection process. We also introduce a variant, *IGSR-Agent*, which employs a second LLM agent to interpret these scores and make semantic pruning decisions. This iterative cycle is governed by a Monte Carlo Tree Search (MCTS) algorithm, enabling systematic exploration of the combinatorial equation space (see Figure 1). Full algorithmic details are provided in Appendix B, pseudocode in Appendix B.5, and additional settings in Appendix I.

---

**Default IGSR Workflow at a Glance**

Given a dataset and a task description, IGSR repeats a *propose-and-prune* cycle, explored with MCTS:

1. **Propose:** LLM suggests $\sim 5$ candidate basis functions $\psi_j$ (10 in first round), conditioned on the task context and a history of previously kept/dropped terms.
2. **Fit:** an OLS linear model $f(\mathbf{x}) = \sum_j w_j \psi_j(\mathbf{x})$ is fit to the current candidate set.
3. **Score:** each term's per-term influence $\Delta_j$ is computed on validation (*no-refit*: set $w_j \rightarrow 0$ with the other weights held fixed).
4. **Prune:** retain the top-$K$ ($K=6$) terms by $\Delta_j$ (for multi-output, rank by $\max$ over outputs).
5. **Search:** explore via heuristic MCTS (immediate node reward = negative val. set MSE; budget 30 expansions, 5 successors per node, exploration constant $c=\sqrt{2}$).

---

## 2.1. Core Iterative Step: Propose-and-Prune Cycle

The engine of IGSR is a propose-and-prune cycle, which constitutes a single iteration of model refinement. This cycle takes a current set of basis functions and a history of past attempts, generating a new, refined equation state. It consists of three distinct phases: term generation, influence evaluation, and term pruning (see Algorithm 2 in Appendix B.5).

**1) Term Generation Phase.** The process begins by invoking the "Propose" LLM agent to suggest new candidate basis functions. To ground the LLM's generation in the scientific reality of the task, the prompt includes extensive context: a domain-specific description of the variables, the current set of active terms, and the best equation found so far. Crucially, we provide a *history of previous iterations*, summarizing which terms were previously kept or dropped and the resulting impact on MSE. This memory mechanism enables the LLM to perform in-context learning, avoiding repeated mistakes and iteratively improving its proposals (Appendix B.7). The output is a list of valid mathematical expressions.

**2) Candidate Evaluation and Influence Computation.** The newly proposed terms are aggregated with the existing

basis functions to form an expanded candidate set. A linear model (e.g., Ordinary Least Squares) is fitted to the training data using these terms, yielding weights $\mathbf{w}$. We then compute the *per-term influence score*, $\Delta_j$, for each candidate term $\psi_j$ on the validation set. Defined as the increase in validation MSE if term $j$ were removed (i.e., $w_j \rightarrow 0$) while holding other coefficients fixed, $\Delta_j$ acts as a granular credit assignment signal (see Appendix B.2 for calculation details). Unlike global error metrics, $\Delta_j$ isolates the specific utility of each component, allowing the system to distinguish between critical drivers of the dynamics and redundant noise. For a derivation of the influence score on validation data and an empirical comparison against refit-aware alternatives, please refer to Appendix G.11. We further verify that reusing the validation split for both influence-based pruning and the MCTS reward does not induce search-time overfitting (Appendix G.10).

**3) Term Pruning Phase.** This phase refines the expanded candidate set back down to a parsimonious model size (governed by the sparsity constraint $K$). IGSR supports two modes for this selection:

- **Deterministic Pruning (Default):** In the standard configuration, pruning is strictly data-driven. Terms are ranked by their aggregate influence score $\Delta_j$, and the top $K$ terms are retained. This approach is computationally efficient and eliminates the risk of an LLM hallucinating the removal of statistically significant terms. This influence-based selection remains reliable even under multicollinearity among candidates and for interaction-only (epistasis-like) signals, as stress-tested in Appendix G.12.
- **Agentic Pruning (IGSR-Agent):** In this variant, the feedback tuple $(\psi_j, w_j, \Delta_j)$ is passed to a second "Prune" LLM agent. This agent is instructed to use the influence scores as a primary heuristic ("$\Delta_j \approx 0 \implies$ drop") while also applying semantic reasoning to judge term plausibility. This allows for the injection of domain knowledge constraints at the selection stage.

Finally, the retained terms are used to fit the final model for this iteration. The results, including the specific terms kept and dropped, are appended to the history buffer to inform the "Propose" agent in subsequent steps.

## 2.2. Search Strategy for Equation Discovery

While the propose-and-prune cycle provides a mechanism for local refinement, discovering the optimal functional form requires navigating a vast and combinatorial search space. To do this effectively, IGSR embeds the iterative cycle within a *Monte Carlo Tree Search (MCTS)* framework.

In this formulation, each node in the search tree represents a distinct equation state defined by a set of basis functions and their optimized weights. The edges of the tree correspond to transitions generated by the propose-and-prune cy-

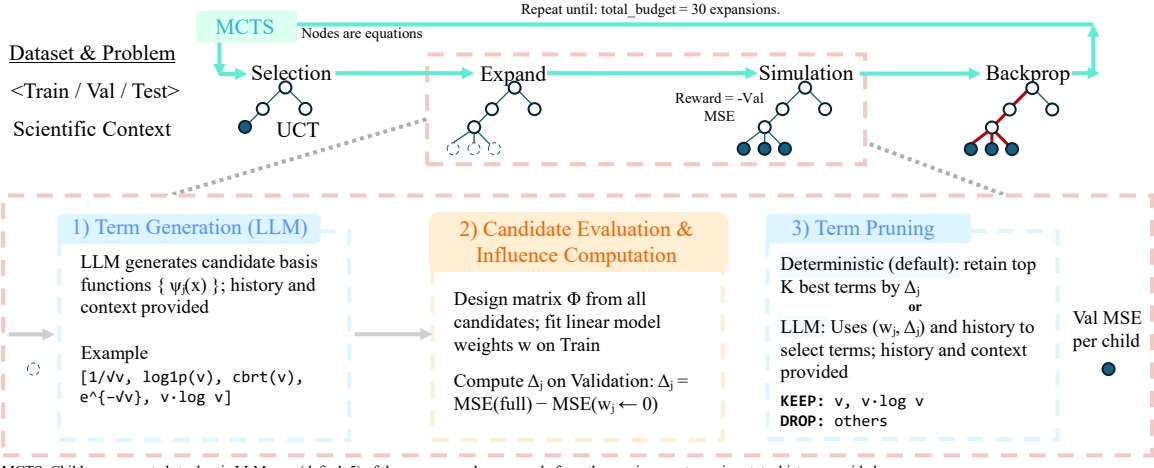

*Figure 1.* Conceptual overview of the Influence-Guided Symbolic Regression (IGSR) framework. The system operates via a *Propose-and-Prune Cycle* embedded within a *Monte Carlo Tree Search*. An LLM agent generates candidate basis functions $\psi_j(\mathbf{x})$ based on problem context. These terms are evaluated to compute weights $w_j$ and per-term influence scores $\Delta_j$. A deterministic selector retains terms with the highest influence (or in the *IGSR-Agent* variant, a secondary LLM performs this selection). This feedback loop allows the system to iteratively refine the equation structure, balancing exploration of new functional forms with exploitation of high-influence components.

cle. From a given parent equation, the "Propose" agent can generate multiple distinct successor states (children) by sampling from the LLM's stochastic output distribution. This branching allows the system to explore alternative refinement pathways simultaneously – for instance, one branch might explore adding trigonometric terms while another explores interaction terms.

The MCTS algorithm balances *exploration* (visiting less-sampled parts of the equation space) and *exploitation* (refining promising equations) using the Upper Confidence Bound for Trees (UCT) criterion (Kocsis & Szepesvári, 2006). The "reward" signal used to evaluate nodes is the negative validation MSE of the discovered equation. While full rollouts (simulating multiple future steps) are supported, our default configuration employs a *heuristic MCTS* approach where the immediate reward of a newly expanded node is back-propagated directly. This focuses the computational budget on breadth and immediate quality, which we found efficient for symbolic regression tasks (see Appendix G.4 on this, and Appendix B.4 for full implementation details).

Although MCTS is the primary search strategy, IGSR can also operate in a *Linear Iterative Refinement* mode. In this simplified setting, the system follows a single greedy chain of propose-and-prune steps. While computationally cheaper, our ablation studies (Section 5.2) demonstrate that the structured exploration of MCTS significantly improves convergence and final model accuracy by preventing the system from becoming trapped in local optima.

## 3. Related Work

Our work, IGSR, intersects with and differentiates itself from several research areas, as summarized in Tables 1 and 5. We extend the discussion of related works in Appendix A.

**Symbolic Regression (SR).** Traditional SR techniques, such as genetic programming (GP) (Koza, 1994; Schmidt & Lipson, 2009) and sparse regression methods like SINDy for dynamical systems (Brunton et al., 2016), aim to find explicit mathematical equations. While effective for certain problems, these methods often operate on a predefined set of input features and basic mathematical operations. They can struggle with high-dimensional inputs or require significant manual feature engineering to define relevant transformations. Modern SR approaches like PySR (Cranmer, 2023) incorporate sophisticated search algorithms and a wider range of operators but typically do not leverage the semantic reasoning and priors of LLMs for proposing complex basis functions. IGSR differentiates itself by employing an LLM to actively generate these basis functions $\psi_j(\mathbf{x})$, which can be arbitrarily complex, while using rigorous influence feedback to validate them.

**Black-Box Models.** Machine learning models like neural networks (NNs) (Chen et al., 2018; Gorishniy et al., 2021) and gradient-boosted decision trees (GBDTs) (Chen & Guestrin, 2016) are highly effective at fitting complex patterns but lack inherent interpretability. While techniques such as SHAP (Lundberg & Lee, 2017) or LIME (Ribeiro et al., 2016) can provide post-hoc explanations, they do not yield a concise, closed-form mathematical model. IGSR, in contrast, aims to produce inherently interpretable equations where the contribution of every component is explicitly weighted.

**LLM-driven Equation Discovery.** The application of LLMs to scientific discovery is a rapidly advancing field. Systems like AI Feynman (Udrescu & Tegmark, 2020) use NNs to guide symbolic decomposition, while Eureka (Ma et al., 2024) employs LLMs to write reward functions for reinforcement learning. The D3 framework (Holt et al., 2024) utilizes a multi-agent LLM system for discovering ODEs, utilizing feedback such as code execution errors and global loss. Other recent methods leverage LLMs in distinct ways: ICSR (Merler et al., 2024) uses in-context learning with scalar fitness scores to iteratively refine equations; LLM-SR (Shojaee et al., 2025a) treats equations as programs within an evolutionary search; and LaSR (Grayeli et al., 2024) evolves a library of abstract concepts (e.g., "exponential decay") to bias a genetic algorithm.

IGSR's novelty lies in its distinct feedback and selection mechanism. Unlike the abstract concepts in LaSR or the scalar fitness scores in ICSR and LLM-SR, IGSR computes a direct, quantitative vector of *per-term influence scores* ($\Delta_j$). This feedback measures each basis function's marginal contribution to validation accuracy. By utilizing this granular metric, IGSR effectively decouples the generative creativity of the LLM (used to propose terms) from the selection process (which uses $\Delta_j$ for rigorous pruning). This allows for a more analytical refinement loop than standard generative approaches, preventing the retention of statistically insignificant terms hallucinated by the model.

**Influence Functions and Model Interpretability.** The concept of influence functions (Cook & Weisberg, 1980) traditionally measures the impact of individual data points on model parameters. Our use of "influence scores" applies this concept to the *structural components* of the model itself. It assesses the importance of a basis function $\psi_j$ to the overall fit, analogous to a leave-one-out analysis performed at the term level. This provides the principled signal required to navigate the combinatorial search space of equation discovery.

**Automated Feature Engineering (AFE).** IGSR bridges symbolic regression and AFE: it constructs nonlinear feature transformations like AFE, but commits to a single interpretable equation rather than an augmented feature set. It thus differs from prior AFE, both non-LLM (AutoFeat Horn et al., 2019, OpenFE Zhang et al., 2023) and LLM-driven (CAAFE Hollmann et al., 2023, FeatLLM Han et al., 2024, OCTree Nam et al., 2024), which engineer features for a downstream model (often a GBDT) and select them by global loss or tree-based importances; IGSR instead fits a sparse linear model guided by *per-term influence scores* $\Delta_j$. Empirically, IGSR achieves the best mean MSE on 5/6 benchmarks against AutoFeat, OpenFE, SyMANTIC, and CAAFE (Table 23) and surpasses a traditional AFE baseline on LLM-SRBench (Table 24); see Appendices A.6

and G.15.

# 4. Experiments and Evaluation

We evaluate IGSR on a comprehensive suite of tasks designed to assess its ability to discover accurate, interpretable, and scientifically valid models. Our experimental framework spans three categories: (1) a diverse set of six biological and clinical benchmark datasets; (2) the *LLM-SRBench* for large-scale evaluation of scientific discovery capabilities; and (3) a real-world biological case study. Full experimental details are provided in Appendices C to F.

**Benchmark Datasets.** We utilize six distinct datasets representing complex dynamical and static systems:

- **Cancer PKPD (3 Variants)**: Derived from a biomathematical model of lung cancer tumor growth (Geng et al., 2017). We simulate three scenarios: *Lung Cancer* (untreated growth), *Lung Cancer (with Chemo.)*, and *Lung Cancer (with Chemo. & Radio.)*. These require discovering coupled differential equations governing tumor volume and drug concentration.
- **COVID-19**: An agent-based epidemic simulation using Covasim (Kerr et al., 2021), modeling the spread of the virus under various intervention policies.
- **RNA Polymerase**: A high-dimensional bioinformatics dataset (263 features) derived from eNET-seq data (Fong et al., 2022), predicting RNA Polymerase II pausing from genomic sequence and chromatin markers.
- **Warfarin**: A real-world Pharmacokinetic (PK) dataset (Janssen et al., 2022) predicting drug clearance rates in patients based on clinical covariates.

**LLM-SRBench Evaluation.** To test discovery capabilities beyond LLM memorization of standard formulae, we evaluate on *LLM-SRBench* (Shojaee et al., 2025b). We utilize the 128 discovery-oriented problems spanning Physics, Biology, Material Science, and Chemistry. To ensure a fair comparison, all methods are restricted to a fixed computational budget of 300k tokens per run.

**Benchmark Methods.** We compare IGSR against a wide range of baselines spanning three paradigms:

1. *White-box Non-LLM*: Sparse regression (**SINDy**, Brunton et al., 2016) and genetic programming approaches (**GPLearn**, Stephens, 2015; **PySR**, Cranmer, 2023).
2. *White-box LLM*: Generative approaches including a zero-shot baseline (**ZeroShot**), zero-shot with parameter optimization (**ZeroOptim**), and iterative in-context learning without influence scores (**ICL**). We also compare against state-of-the-art frameworks: **D3** (Holt et al., 2024), **ICSR** (Merler et al., 2024), **LLM-SR** (Shojaee et al., 2025a), and **LaSR** (Grayeli et al., 2024).
3. *Black-box*: Neural ODEs (**DyNODE**, Chen et al., 2018), Recurrent Neural Networks (**RNN**), Transform-

ers (Vaswani et al., 2017), and Gradient Boosted Trees (**XGBoost**, Chen & Guestrin, 2016).

**Evaluation Metrics**. For the six benchmark datasets, we report the Mean Squared Error (MSE) on held-out test data, averaged over multiple independent seeds with 95% confidence intervals. For *LLM-SRBench*, we report both predictive metrics (Normalized MSE and Accuracy within tolerance $Acc_{0.1}$) and symbolic recovery metrics (Term Recall and Symbolic Accuracy) as defined in Shojaee et al. (2025b) and Appendix E.

# 5. Results

**Benchmark Datasets Evaluation.** Comprehensive evaluations across the benchmark datasets are presented in Table 2a, and the *IGSR-Agent* variant is compared in Appendix G.1. IGSR consistently demonstrates strong performance, achieving the lowest MSE among interpretable model classes in 5/6 datasets, and competitive results against black-box models. This indicates its ability to discover accurate and concise closed-form equations. Beyond the baselines in Table 2a, IGSR also outperforms a broad set of automated feature-engineering methods (AutoFeat, OpenFE, SyMANTIC, and the LLM-based CAAFE; Appendix G.15), and a flexible neurosymbolic baseline LIES (Appendix G.16) across these datasets.

**LLM-SRBench Evaluation.** To validate the capacity of IGSR for genuine scientific discovery beyond the retrieval of memorized physical laws, we conducted a comprehensive evaluation against LLM-powered methods on LLM-SRBench (Shojaee et al., 2025b). We utilized the discovery-oriented problems of the benchmark across four scientific domains: Physics, Biology, Material Science, and Chemistry. To ensure fair comparison, we enforced a fixed computational budget of 300k tokens per run. We report the average rank across all problems for both predictive metrics, specifically Normalized MSE (NMSE) and Accuracy within tolerance ($Acc_{0.1}$), and symbolic recovery metrics, specifically Term Recall and Symbolic Accuracy (metrics as used in Shojaee et al. (2025b), Term Recall as defined in Appendix E). As shown in Table 2b, IGSR achieves the best average rank in predictive performance across both in-distribution (ID) and out-of-distribution (OOD) test sets. This confirms that the influence-guided search effectively optimizes for generalization. The dual-agent variant, IGSR-Agent, ranks second in predictive accuracy, while the symbolic recovery metrics are comparable in both variants, and better than the baselines. We present detailed results, per-domain breakdown of performance, and fidelity–accuracy tradeoff investigation in Appendix E. We also confirm that IGSR outperforms a traditional AFE baseline (AutoFeat + LASSO) on LLM-SRBench in Table 24.

## 5.1. RNA Polymerase II Pausing Case Study

The discovery of quantitative rules governing biological processes is crucial for advancing our understanding of life. Automated machine learning approaches like IGSR offer a path to generate interpretable, data-driven hypotheses from complex biological datasets, potentially accelerating discovery. Here, we present a case study in which IGSR was used not only to replicate existing knowledge but to generate a novel hypothesis subsequently supported via wet-lab experimentation (see Appendix F for full details).

**Biological Background.** RNA polymerase II (Pol II) transcription is a fundamental process involving initiation, elongation, and termination (Cramer, 2019). Transcription speed is non-uniform, influenced by frequent Pol II pausing (Noe Gonzalez et al., 2021; Jonkers & Lis, 2015; Danko et al., 2013; Bentley, 2014; Zamft et al., 2012). Pause sites, particularly at G residues preceding T/C on the non-template DNA strand, are key determinants of elongation speed (Fong et al., 2022; Gajos et al., 2021). Nucleosomes and histone modifications like H3K36me3 are also implicated (Bondarenko et al., 2006; Churchman & Weissman, 2011; Lee et al., 2025), but their precise roles and the sequence determinants of pausing in human cells remain incompletely understood.

DISCOVERY OF SEQUENCE AND CHROMATIN FACTORS

**Dataset and Setup.** We analyzed eNET-seq data mapping Pol II pause sites in human cells at single-base resolution (Fong et al., 2022). The dataset contained 263 features, including local DNA sequence context, MNase-seq signal (nucleosome occupancy), and histone modification signals (e.g., H3K36me3). We applied IGSR on a balanced dataset of pause and control sites to distinguish pause regions from background.

**Findings.** The equation discovered by IGSR (detailed in Appendix F.1) successfully captured known biological priors while proposing specific sequence dependencies. An expert review by an Anonymous Biologist confirmed the biological plausibility of these findings.

> **Expert Review of IGSR Discovered Equation**
>
> ▶**Confirms existing knowledge:** The model identified a positive association of nucleosome occupancy ($signal_{MNase}$) and H3K36me3 with pausing, and higher pausing in termination regions, consistent with prior studies (Bondarenko et al., 2006; Churchman & Weissman, 2011; Gromak et al., 2006).
> ▶**Discovers novel sequence elements:** Beyond the known GT element at positions 0/+1 (Fong et al., 2022; Gajos et al., 2021), the model identifies C at positions 0 and -1, and T at -1, -2, -3 as significant negative predictors of pausing. These represent new, testable hypotheses about sequence-dependent pausing mechanisms.

*Table 2.* **2a**, **Benchmark Datasets Evaluation.** Test MSE (mean±95 % CI) on held-out data for six benchmarks. IGSR demonstrates competitive or superior performance, particularly among interpretable models. Results on 25 seeds; GPT-4o as LLM backbone. Three vertical subsections correspond to baseline classes: (1) *white-box non-LLM baselines*, (2) *white-box LLM baselines*, (3) *black-box baselines* (dark gray font). Dashes (—) indicate "unable to run", reason clarified in footnote. Average rank computed for white-box methods only. **2b**, **LLM-SRBench Evaluation.** Performance aggregated across 128 discovery problems in Physics, Biology, Material Science, and Chemistry. We report the average rank achieved by each method (best result in **bold**, second best underlined) under a fixed token budget. IGSR variants outperform the baselines in both predictive performance (NMSE and $Acc_{0.1}$) and structural recovery metrics (Term Recall and Symbolic Accuracy), all metrics defined in Appendix E. GPT-4o-mini LLM backbone, 5 seeds per problem.

*(a)* Benchmark Datasets Evaluation.

| Method | LC MSE ↓ | LC-C MSE ↓ | LC-CR MSE ↓ | C-19 MSE ↓ | RNAPol MSE ↓ | Warf MSE ↓ | Avg. Rank ↓ |
|---|---|---|---|---|---|---|---|
| SINDy | 335±3.14 | 0.838±6.35e-3 | 0.664±5.14e-3 | 1.43e-7±4.48e-9 | —† | 0.914±0.169 | 5.50 |
| GPLearn | 7.56±1.11 | 46.8±15.5 | 46.8±4.91 | 7.13e-4±5.06e-4 | 0.0204±5.55e-4 | 2.53±0.169 | 7.33 |
| PySR | 669±7.86 | 0.12±0.0697 | 0.399 ± 0.123 | 4.23e-7±8.3e-8 | —† | 29.9±0.859 | 6.17 |
| ZeroShot | 2.13e13±4.35e13 | 4.97e3±3.67e3 | 2.54e3±2.74e3 | 1.34e8±2.09e8 | 1.35e5±1.75e5 | 5.3e3±1.07e4 | 10.50 |
| ZeroOptim | 0.142±0.119 | 86.2±27.3 | 122±6 | 1.41e-7±1.04e-7 | 0.0130±2.87e-4 | 0.861±0.177 | 5.50 |
| ICL | 0.0557±0.0486 | 21.2±9.8 | 63.3±16.5 | 9.35e-8±1.77e-8 | 0.0119±3.52e-4 | 0.784±0.193 | 3.83 |
| D3-w.b. | 1.01e4±1.27e4 | 45±28.9 | 253±273 | 7.81e-6±2.48e-7 | 0.043±0.0366 | 1.15±0.343 | 8.00 |
| LLM-SR | 33.4±0 | 42.2±35.6 | 32.1±48.4 | 4.53e-4±9.12e-4 | 1.44±1.91 | 1.24±0.564 | 7.00 |
| LaSR | 658±7.31 | 1.71±1.16 | 3.97±3.21 | 2.59e-6±8.66e-7 | 0.0172±6.49e-4 | 30.1±0.992 | 6.33 |
| ICSR | 0.407±0.244 | 0.688±0.39 | 6.1±1.05 | 1.03e-7±1.6e-8 | —‡ | 0.497±0.0646 | 4.17 |
| DyNODE | 497±18.4 | 355±3.63 | 398±14.2 | 7.48e-5±5.58e-7 | 0.0136±6.7e-5 | 0.593±0.0769 | - |
| RNN | 1.33e-3±4.85e-4 | 6.85e-3±2.2e-3 | 0.013±0.00552 | 5.88e-8±1.5e-9 | 7.28e-3±4.79e-5 | 0.756±0.139 | - |
| Transformer | 0.0375±6.53e-3 | 0.179±0.0135 | 0.366±0.0269 | 4.05e-7±5.05e-8 | 7.24e-3±3.3e-5 | 0.804±0.158 | - |
| XGBoost | 0.012±9.64e-4 | 0.155±4.88e-3 | 0.322±8.84e-3 | 8.43e-8±5.36e-9 | 7.58e-3±3.79e-5 | 1.67±0.473 | - |
| **IGSR** | **5.64e-5±6.79e-5** | **0.0013±0.0010** | **0.0141±0.0087** | **5.01e-8±1.78e-9** | **0.0111±0.0004** | **0.565±0.113** | **1.17** |

*(b)* LLM-SRBench Evaluation.

| Method | ID Test Set | | OOD Test Set | | Symbolic Metrics | |
|---|---|---|---|---|---|---|
| | NMSE | $Acc_{0.1}$ | NMSE | $Acc_{0.1}$ | Term Recall | Symb. Acc. |
| ICL | 3.91 | 2.53 | 4.60 | 2.41 | 3.50 | 1.35 |
| D3-w.b. | 6.89 | 4.16 | 6.27 | 4.34 | 3.16 | 1.34 |
| LLM-SR | 3.71 | 2.38 | 3.92 | 2.33 | 3.13 | 1.29 |
| LaSR | 5.53 | 3.94 | 4.77 | 3.33 | 4.60 | 1.30 |
| ICSR | 4.21 | 3.34 | 4.22 | 2.71 | 3.70 | 1.33 |
| IGSR-Agent | 2.41 | 1.50 | 2.45 | 1.49 | **1.98** | **1.16** |
| **IGSR** | **1.35** | **1.12** | **1.77** | **1.21** | 1.99 | 1.17 |

**Abbreviations:** D3-w.b.: D3-white-box; LC: Lung Cancer, LC-C: Lung Cancer (with Chemo.), LC-CR: Lung Cancer (with Chemo. & Radio.), C-19: COVID-19, RNAPol: RNA Polymerase, Warf: Warfarin PK. **Footnotes:** †: Method did not finish within maximum runtime of 3 hours per seed due to the large number of features in this dataset (263). ‡: ICSR implementation fails with this dataset due to the large number of features in this dataset (263).

FROM SYMBOLIC MODEL TO WET-LAB VERIFICATION

**Setup and Hypothesis Generation.** To further assess IGSR's capability for genuine discovery, we conducted a subsequent study targeting the *intensity* of pausing rather than its mere presence. We applied IGSR to a modified regression task: predicting the magnitude of the pause score exclusively at identified pause sites. We also introduced replicate DNA CpG methylation mapping by Bisulphite-seq in HCT116 cells (Du et al., 2021).

The resulting equation (see Appendix F.3) achieved a superior MSE compared to previous iterations. Crucially, the discovered coefficients indicated a notable overall *negative* effect of the methylation proxy. In other words, IGSR suggested a novel biological hypothesis: "*DNA methylation in gene bodies acts to suppress transcriptional pausing.*" While promoter methylation is known to suppress initiation (Deaton & Bird, 2011), the role of intragenic methylation in elongation is less defined and often debated (Lorincz et al., 2004; Yang et al., 2014).

**Wet-Lab Verification.** To test this AI-generated hypothesis, we performed a targeted experiment using HCT116 human colorectal carcinoma cells. We treated cells with 5-azacytidine, a potent hypomethylating agent, to deplete DNA methylation, and then mapped transcriptional pausing using eNET-seq.

The experimental data revealed a significant *increase* in both the frequency and strength of Pol II pauses in the hypomethylated cells relative to DMSO-treated controls, consistently across two biological replicates ($\sim$3000 gene-body pause sites per condition; Wilcoxon $p=4.3 \times 10^{-52}$ for pause frequency and $p=3.5 \times 10^{-33}$ for pause strength).

This inverse relationship (removing methylation increases pausing) empirically supports the hypothesis of IGSR; the full experimental protocol and statistics are reported in Appendix F.3.

> **Wet-Lab Verification of IGSR Hypothesis**
>
> ▶ **IGSR Prediction:** The discovered equation contained methylation terms with negative coefficients, predicting that methylation *suppresses* pausing.
> ▶ **Experimental Observation:** Pharmacological depletion of DNA methylation (using 5-azacytidine) caused a marked *increase* in transcriptional pausing.
> ▶ **Conclusion:** The wet-lab results support the IGSR-derived hypothesis that intragenic CpG methylation facilitates elongation by suppressing pausing, demonstrating the framework's ability to drive expanding scientific knowledge.

**Interpretation.** The case study illustrates IGSR's main value in complex biology: in this high-dimensional setting, with no ground truth available, the discovered equation is best read as an *interpretable approximation* that generates concrete, testable hypotheses rather than an exact mechanism. The methylation finding it surfaces is robust, recovered in 7/10 seeds on the original replicate, 6/10 on an independent biological replicate, and 8/10 under random 100-feature subsets (Table 10). Where ground truth is known, as in the PKPD benchmarks, IGSR instead recovers the true structure closely (Section 5.3).

**5.2. Impact of Influence Feedback and Search Strategy**

To understand the contributions of IGSR's core components, we performed ablation studies on the Lung Cancer (with Chemo. & Radio.) benchmark dataset. We compared the full IGSR model against variants where either the MCTS

tree search component was disabled (falling back to iterative refinement) or the detailed influence based feedback was removed (term pruning step replaced with an LLM pruner that is given the val. MSE and asked to select terms to keep), or both. Results are shown in Table 3. The reported NMSE values are for the Lung Cancer (with Chemo. & Radio.) dataset, averaged over 25 seeds.

*Table 3.* **Ablation of Influence Feedback and MCTS.** We quantify the impact of removing the MCTS component and/or the influence feedback term pruning mechanism on the Lung Cancer (with Chemo. & Radio.) dataset. Results are test NMSE (lower is better) averaged over 25 seeds with 95% confidence intervals.

| Variant | MCTS | Influence Feedback | NMSE ↓ |
|---|:---:|:---:|---|
| **Full IGSR** | ✓ | ✓ | **0.000787**±0.000532 |
| w/o MCTS (Iterative + Influence Feedback) | ✗ | ✓ | 0.626±1.35 |
| w/o Influence Feedback (MCTS + Basic Feedback) | ✓ | ✗ | 0.293±0.64 |
| w/o MCTS or Influence Feedback (Iterative + Basic Feedback) | ✗ | ✗ | 4.85±2.36 |

The results clearly indicate that both the influence-based feedback and the MCTS tree search contribute significantly to IGSR's performance. Removing influence feedback (even with tree search) leads to a substantial increase in NMSE (from 0.000787 to 0.293), demonstrating that granular, per-term signal is crucial for effective term selection and equation refinement. Figure 2 further demonstrates that this advantage leads to improved convergence. Similarly, removing the tree search (even with influence feedback, NMSE increases to 0.626) also results in degraded performance, confirming that systematic exploration helps in finding better solutions compared to a purely linear iterative approach. The variant without both components performs the worst (NMSE of 4.85), underscoring the synergistic benefits of both design choices. Appendix G.2 provides further detail.

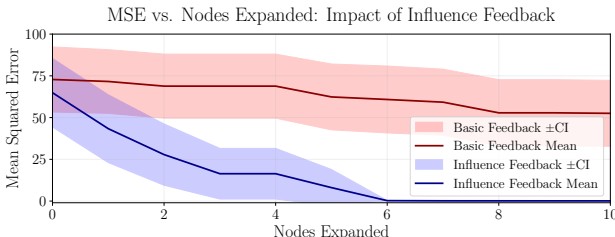

*Figure 2.* **Influence Feedback Improves Convergence.** Best test MSE achieved, shown over MCTS nodes expanded (iterations). "*Influence Feedback*" is full IGSR, and "*Basic Feedback*" is the ablation where the pruner is guided by a scalar loss value only. Use of per-term influence feedback leads to faster convergence and a lower final MSE. Lung Cancer (with Chemo. & Radio.) dataset, 25 seeds, 95% CIs.

These experiments underscore the importance of IGSR's core principles: (i) utilizing per-term influence values (*rich, structured feedback at the component level*) to (ii) *systematically search* over the LLM-proposed basis functions that comprise the complex landscape of possible equations.

## 5.3. Computational Efficiency and Method Robustness

**Computational Costs and Scaling.** A pervasive challenge in symbolic regression is the combinatorial explosion of the search space as the number of input features ($d$) increases. IGSR avoids this by using the LLM's semantic priors to propose relevant terms and the influence-based pruning mechanism to promptly filter them. To quantify this efficiency, we evaluated IGSR's wall-clock runtime on the RNA Polymerase dataset while varying the number of available input features from 10 to the full 263 (see Figure 3). We find that the method exhibits sub-linear scaling: increasing the feature count by over $25\times$ (10 to 263) results in only a marginal increase in runtime ($\approx 25\%$). Peak memory likewise stays modest on this largest dataset (peak RSS $2.782 \pm 0.003$ GB over 5 seeds; Appendix H.2).

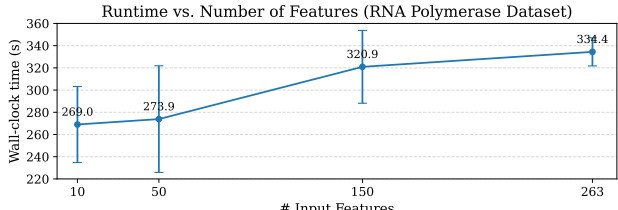

*Figure 3.* **Scalability to High-Dimensional Spaces.** Total wall-clock time on the RNA Polymerase dataset (mean $\pm$ 95% CI, 10 seeds) is shown as the number of available input features increases. Runtime scales sub-linearly up to the full 263 features, avoiding the combinatorial explosion typical of traditional symbolic search.

We also find that on Lung Cancer (with Chemo. & Radio.) dataset, IGSR is more computationally efficient than other LLM-based methods, achieving superior accuracy at an equivalent LLM token budget, and show that per-iteration, IGSR converges faster than traditional Genetic Programming methods like GPLearn (Appendices G.5 and H.3).

**Stability and Reliability of Discovered Equations** A noteworthy question for any discovery method is whether it recovers a stable structure or merely one of many equally-scoring fits. On Lung Cancer (with Chemo. & Radio.), where the ground-truth PKPD equations are known, the terms IGSR selects are highly consistent across 25 seeds (Figure 4): the most frequent terms are all ground-truth, and the cross-seed variation that remains lies in how a Gompertz nonlinearity is approximated. A pairwise-similarity analysis in Appendix G.13 corroborates this.

This stability rests on reliable pruning. In controlled stress tests (Appendix G.12), the default no-refit selection recovers all true signal groups under heavy multicollinearity, and retains genuine interaction terms even when their constituents are individually uninformative (an epistasis-like setting). IGSR's equations are thus both stable across seeds and robust to correlated or interaction-only signal.

| Term | Freq. | GT | Role in true equation |
|------|-------|-----|----------------------|
| $C$ | 25/25 | ✓ | Drug clearance |
| $x \cdot d$ | 25/25 | ✓ | Radio cell kill |
| $x \cdot C$ | 24/25 | ✓ | Chemo cell kill |
| $u^c$ | 21/25 | ✓ | Drug infusion |
| $\sqrt{x}$ | 13/25 | | |
| $\log(x)$ | 12/25 | | |
| $x \cdot \log(x)$ | 12/25 | ✓ | Gompertz saturation |
| $x$ | 12/25 | ✓ | Gompertz growth |
| $(u^c)^2$ | 4/25 | | |
| $x \cdot (C+d)$ | 1/25 | | |
| $x^{0.33}$ | 1/25 | | |

(a) Canonical term frequencies. "GT" = ground-truth, i.e. terms present in the known PKPD equations. All four terms at $\geq 80\%$ are GT.

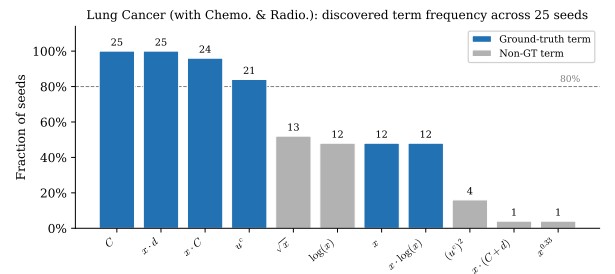

(b) Bar chart of (a). Blue = GT term, grey = non-GT. Dashed line = 80% threshold.

| | Seeds | Val. MSE | $\bar{J}$ | GT/seed$^{\dagger}$ | Characterization |
|--|-------|----------|-----------|---------------------|------------------|
| *Tier 1* | 12/25 | $<10^{-3}$ | 0.95 | 5.9/6 | Near-exact recovery; all 12 seeds share the same 6 GT terms |
| *Tier 2* | 13/25 | $\approx 0.023$ | 0.81 | 3.7/6 | Approximates Gompertz term $x \cdot \log x$ via $\sqrt{x}$, $\log x$; all four core GT interaction terms retained |
| *Overall* | 25/25 | | 0.65 | | 86 of 300 seed pairs share identical term sets |

$^{\dagger}$ Out of 6 discoverable GT terms. The 7th GT term, $xd^2$ (quadratic radiotherapy kill), is never discovered by any seed; its coefficient $\beta_r = 0.00398$ is an order of magnitude below the linear coefficient $\alpha_r = 0.0398$, making its contribution negligible at the clipped dose range $d \in [0, 2]$ Gy, used in the simulations.

(c) Stratification by performance. Inspecting per-seed validation MSE reveals a natural split into two groups.

*Figure 4.* **Cross-seed stability of discovered equation terms** on Lung Cancer (with Chemo. & Radio.), 25 seeds. Variables: $x$ = tumor volume, $C$ = chemo. drug concentration, $u^c$ = chemo. dosage, $d$ = radiotherapy dose. After canonicalizing variants (e.g. `np.log1p` → `np.log`), only 11 unique canonical terms remain. The four highest-frequency terms (84–100%) are all ground-truth (GT) from the known PKPD equations. IGSR's discovered equations are structurally stable across seeds: the core mechanistic terms (drug interactions, Gompertz growth) are consistently recovered, and cross-seed variation is primarily in how the Gompertz growth nonlinearity is approximated.

**Method Robustness and Generalization.** We demonstrate that IGSR's performance advantage is maintained across nine different backbone LLMs, including open-weight models (Appendix G.6); that it's robust to inclusion of a large number of irrelevant features (Appendix G.7); that it can successfully discover equations for unusual synthetic model structures (Appendix G.8); and that the discovered RNA Pol. pausing models generalize robustly to an unseen experimental replicate (Appendix G.9). We investigate the framework's sensitivity to prompt variation in Appendix G.19, and to the "proposed terms count" and "rollout depth" hyperparameters in Appendices G.4 and G.18. We quantify the diversity of generated candidate terms (term proposal stage) in Appendix G.20.

## 6. Discussion

We have presented Influence-Guided Symbolic Regression (IGSR), a framework that reframes the equation discovery problem by decoupling the generative creativity of Large Language Models from the rigorous selection pressure of statistical influence analysis. By shifting the feedback mechanism from a global scalar loss to granular, per-term influence scores ($\Delta_j$), IGSR enables a search process that is both semantically rich and data-efficient. Our empirical evaluation across diverse benchmarks demonstrates that this approach, on average, delivers superior performance relative to both established evolutionary approaches and many recent LLM-driven discovery frameworks. Furthermore, the method demonstrates robust scaling in high-dimensional biological settings, a regime that frequently challenges traditional symbolic regression techniques.

A central objective of scientific machine learning is the discovery of mechanisms that withstand physical validation. A particularly important contribution of this work is the demonstration of a complete discovery loop, transitioning from *in silico* hypothesis generation to *in vitro* verification. In our RNA Polymerase II case study, IGSR identified a novel relationship regarding the role of DNA methylation in transcriptional pausing, which was subsequently supported via targeted wet-lab experimentation. This validates the premise that LLM-driven search, with granular feedback, can act as a reliable partner in the scientific method.

**Limitations and Future Work.** Despite these advances, IGSR operates within specific constraints. First, the discovery capability is bounded by the generative priors of the underlying LLM; if a required functional form or transformation is completely absent from the LLM's pre-training distribution or context, the system is unlikely to propose it. Second, the framework fundamentally assumes a structure of $f(\mathbf{x}) = \sum w_j \psi_j(\mathbf{x})$. While our Term-Local Optimization (IGSR-TLO) variant mitigates this by tuning internal constants, discovering fully nested nonlinear dynamical systems remains a challenge compared to differentiable approaches. In addition, while the current implementation optimizes for validation MSE, incorporating explicit parsimony penalties (e.g., term counts or complexity metrics) directly into the value estimation could allow the search to more effectively navigate the Pareto frontier between accuracy and interpretability. Future research could tackle these limitations, aiming to refine the optimization of nested functional compositions or to formalize the multi-objective search for parsimonious yet accurate scientific models.

# Acknowledgements

We sincerely thank Nova Fong (University of Colorado, Anschutz Medical Campus), who performed all the wet-lab experiments, for this crucial contribution to validating IGSR.

This work was supported by Azure sponsorship credits granted by Microsoft AI for Good Research Lab, with special thanks to Dr. Juan M. Lavista Ferres, Director of Microsoft AI for Good Lab, for his support.

ES is funded by Dinwoodie Charitable Company and University Hospital Southampton NHS Foundation Trust.

# Impact Statement

This paper presents work whose goal is to advance the field of Machine Learning, specifically in the domain of interpretable scientific discovery. By developing Influence-Guided Symbolic Regression (IGSR), we aim to move AI-driven science away from opaque black-box predictions toward transparent, symbolic representations. This inherent interpretability is a positive step toward trustworthy AI, allowing domain experts to inspect and critique the logic governing model predictions.

However, the application of Generative AI to scientific discovery carries potential risks. The primary ethical concern is the risk of "scientific hallucination", where LLMs may propose equations that are mathematically elegant and statistically accurate on limited data, yet scientifically spurious. To mitigate this risk, the IGSR framework, particularly within its agentic pruning setup, allows for the direct injection of scientific domain knowledge and constraints via prompting. This capability helps ground the LLM's decision-making in established physical or biological principles, reducing the likelihood of accepting plausible but incorrect models.

Despite these safeguards, users might mistakenly interpret these data-driven hypotheses as validated causal laws. Given that our benchmarks include sensitive domains such as cancer treatment (PKPD), epidemiology (COVID-19), and genomics, we strongly emphasize that models discovered by IGSR should be treated as candidate hypotheses requiring rigorous expert review and empirical verification (as demonstrated by our wet-lab validation) before being considered for real-world medical or policy decision-making. Blind reliance on unverified symbolic models in clinical or public health settings could lead to adverse societal outcomes. Finally, we acknowledge that IGSR relies on pre-trained LLMs, which may introduce biases present in their training corpora into the scientific search process.

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

# Appendix

## Table of Contents

# A. Additional Related Work

Influence-Guided Symbolic Regression (IGSR) synergizes concepts from symbolic regression, the rapidly evolving capabilities of Large Language Models (LLMs) in scientific reasoning, machine learning interpretability, and advanced search techniques. This extended related work section aims to provide a more comprehensive contextualization of IGSR, elaborating on its distinctions and contributions by delving deeper into these intersecting domains.

## A.1. Symbolic Regression: Foundations and Evolution

Symbolic Regression (SR) is the problem of identifying a mathematical expression that best fits a given dataset, without assuming a pre-specified model structure. This inherently seeks interpretable models.

**Traditional Approaches:** Genetic Programming (GP) has historically been a dominant paradigm for SR (Koza, 1994; Schmidt & Lipson, 2009; Stephens, 2015). GP-based SR typically evolves a population of candidate expressions (often represented as trees) using evolutionary operators like crossover and mutation. While powerful, traditional GP can face challenges such as premature convergence, code bloat (expressions becoming overly complex), and difficulties in efficiently exploring vast search spaces, especially with high-dimensional data or when a diverse set of mathematical operators is required (Cranmer, 2023; Zeng et al., 2023). The search can be computationally intensive, and the quality of discovered equations can be sensitive to the choice of initial function sets and hyperparameters.

**Sparse Symbolic Regression:** Methods like SINDy (Sparse Identification of Nonlinear Dynamics) (Brunton et al., 2016) leverage sparse regression. SINDy constructs a library of candidate (often nonlinear) functions of the state variables and uses techniques like LASSO or sequentially thresholded least-squares to find a sparse combination of these functions that best describes the system's dynamics, primarily for ODEs. While effective for systems where the basis functions are well-chosen, the library of candidate functions is often pre-defined by the user, which might limit discovery of truly novel functional forms not anticipated by the domain expert.

**Modern Advancements in SR:** More recent SR methods have introduced innovations to tackle these challenges. PySR (Cranmer, 2023) incorporates techniques from simulated annealing, genetic algorithms, and a highly optimized search process with a broad library of operators to find Pareto-optimal equations (balancing accuracy and complexity). AI Feynman (Udrescu & Tegmark, 2020; Udrescu et al., 2020) introduced a recursive divide-and-conquer strategy inspired by physics problem-solving techniques. It attempts to discover symmetries, separability, and other properties of the target function to break it down into simpler components, often using neural networks to guide these decompositions. Deep Symbolic Regression (DSR) approaches (Petersen et al., 2021; Biggio et al., 2021; Kamienny et al., 2022) often use recurrent neural networks (RNNs) to generate expressions token by token, framing SR as a sequence generation problem, sometimes guided by reinforcement learning. Other neural approaches like SymbolNet (Tsoi et al., 2025) focus on scalability to high-dimensional inputs and model compression by dynamically pruning operators and features.

**IGSR's Differentiation in SR:** IGSR distinguishes itself from these SR paradigms in several key aspects. Unlike traditional GP or sparse SR methods that rely on pre-defined or combinatorially generated basis functions, IGSR tasks an LLM with proposing candidate basis functions $\psi_j(\mathbf{x})$. These can be arbitrarily complex and draw upon the LLM's vast pre-trained knowledge of mathematical and scientific relationships. Crucially, IGSR provides a mechanism for calculating highly granular, *per-term influence scores* $\Delta_j$. This allows for a rigorous *propose-and-prune* cycle where terms are generated by an LLM but selected based on their explicit quantitative contribution, rather than relying solely on evolutionary fitness or global loss signals. While AI Feynman uses NNs for specific decomposition tasks, IGSR employs LLMs broadly for generative proposals, coupled with a structure-aware selection mechanism.

## A.2. Large Language Models in Scientific Discovery and Equation Formulation

LLMs are increasingly being explored as powerful tools for accelerating scientific discovery (OpenAI et al., 2024; Brown et al., 2020). Their ability to process and generate human language, understand complex instructions, and synthesize information from vast training corpora makes them suitable for tasks ranging from hypothesis generation to experimental design and data analysis (Boiko et al., 2023).

**LLMs for Equation Discovery and System Modeling:** Several works have specifically investigated LLMs for discovering mathematical models. Eureka (Ma et al., 2024) leverages LLMs, particularly their code-writing capabilities, to design reward functions for reinforcement learning agents that then perform symbolic regression or other optimization tasks. While

Eureka uses LLMs effectively for reward generation, IGSR employs LLMs as the direct architects of the symbolic equations themselves, iteratively proposing and refining basis functions. The D3 framework (Holt et al., 2024) uses LLMs to discover Ordinary Differential Equations (ODEs) for pharmacological systems. D3 employs multiple LLM agents for modeling, feature acquisition, and evaluation in an iterative loop. While sharing the iterative, LLM-driven discovery spirit with IGSR, the feedback mechanism in D3 is generally coarser (e.g., overall model fit, code execution errors, qualitative evaluation). IGSR's unique contribution is the fine-grained, per-term influence scores $\Delta_j$, which enable precise credit assignment. This metric drives the deterministic pruning of the equation's structure and provides granular historical context to the LLM for subsequent iterations, contrasting with the coarser feedback loops in D3.

**Broader LLM Capabilities for Science:** LLMs have also shown promise in generating computer code for simulations (Li et al., 2022), assisting in mathematical reasoning (Imani et al., 2023; Lewkowycz et al., 2022), and forming hypotheses from literature. IGSR specifically harnesses the LLM's pattern recognition and generative capabilities to propose scientifically plausible basis functions and its reasoning capabilities to interpret and act upon the structured influence feedback for model refinement. This structured interaction, providing quantitative evidence for the utility of each proposed component, pushes the LLM beyond simple generation towards a more analytical role.

### A.3. Interpretability in Machine Learning: Inherent vs. Post-Hoc

The demand for interpretable machine learning models is growing, especially in high-stakes domains like science and medicine where understanding the "why" behind a prediction is as important as the prediction itself (Rudin, 2019; Doshi-Velez & Kim, 2017).

**Black-Box Models and Post-Hoc Explanations:** Many high-performing machine learning models, such as deep neural networks (Gorishniy et al., 2021; Chen et al., 2018) and gradient-boosted decision trees (Chen & Guestrin, 2016), are often considered "black boxes" due to their complex internal structures. To shed light on their predictions, post-hoc explanation methods have been developed. Prominent examples include LIME (Local Interpretable Model-agnostic Explanations) (Ribeiro et al., 2016) and SHAP (SHapley Additive exPlanations) (Lundberg & Lee, 2017). LIME approximates the black-box model locally with a simpler, interpretable model. SHAP uses concepts from cooperative game theory (Shapley values) to attribute the prediction to individual features. While these methods provide valuable insights, they offer explanations of an already-trained (and often opaque) model rather than producing a model that is inherently transparent. Limitations can include computational cost, potential instability of explanations, and the fact that the explanation itself is an approximation of the original model's behavior (Rudin, 2019; Kumar et al., 2020; Alvarez-Melis & Jaakkola, 2018).

**IGSR's Pursuit of Inherent Interpretability:** IGSR directly addresses the need for interpretability by aiming to discover models of the form $f(\mathbf{x}) = \sum_j w_j \psi_j(\mathbf{x})$. Each basis function $\psi_j(\mathbf{x})$ is a symbolic expression, and its contribution to the final output is explicitly weighted by $w_j$. This structure is inherently interpretable, allowing domain experts to examine, understand, and potentially validate or refute the discovered relationships based on their domain knowledge. The per-term influence scores $\Delta_j$ further enhance this by quantifying the contribution of each $\psi_j$ to the model's predictive power during the discovery process itself.

### A.4. Neurosymbolic AI

Neurosymbolic AI seeks to combine the strengths of neural networks (e.g., learning from data, pattern recognition) with symbolic reasoning (e.g., logic, explicit knowledge representation, interpretability) (Garcez & Lamb, 2023; Chaudhuri et al., 2021; Besold et al., 2021). This integration aims to create AI systems that are more robust, generalizable, interpretable, and capable of incorporating existing domain knowledge.

IGSR can be viewed as a neurosymbolic system. The LLM, a large neural network, acts as the "neuro" component, responsible for proposing candidate symbolic basis functions based on problem context and history. The symbolic regression task itself – the manipulation of mathematical expressions and the resulting interpretable equation $f(\mathbf{x}) = \sum_j w_j \psi_j(\mathbf{x})$, represents the "symbolic" component. The per-term influence scores act as a critical bridge, translating numerical performance data into a structured signal that drives term selection and informs the LLM's future proposals. We empirically compare IGSR against a dedicated neurosymbolic regressor (LIES) in Appendix G.16.

## A.5. Influence Analysis: From Data Points to Model Components

The concept of "influence" in IGSR, referring to the impact of individual terms $\psi_j(\mathbf{x})$ on the model's predictive performance, draws an analogy to classical influence functions in statistics and more recent feature importance techniques.

**Classical Influence Functions:** Influence functions, introduced in (Hampel, 1974) and further developed in (Cook & Weisberg, 1980), measure the effect of an individual data point on a model's parameters or predictions. They are valuable for outlier detection and understanding model sensitivity to specific observations.

**Feature Importance and Leave-One-Out Analysis:** In machine learning, various methods assess feature importance. Permutation feature importance (Breiman, 2001; Fisher et al., 2019) measures the decrease in model performance when a feature's values are randomly shuffled. Leave-One-Feature-Out (LOFO) importance involves retraining the model with one feature omitted and observing the performance change (Lei et al., 2018; Erdem et al., 2025). Game-theoretic global importance measures, such as SAGE (Covert et al., 2020), instead use Shapley values to attribute a model's overall predictive performance across features in a way that accounts for feature interactions. These methods help identify which input features are most critical for a model's predictions.

**IGSR's Term Influence Scores:** IGSR's per-term influence scores $\Delta_j$ adapt this concept to the components of the *discovered equation itself*. Instead of assessing the impact of raw input features or individual data points, $\Delta_j$ quantifies how much the removal of a specific basis function $\psi_j(\mathbf{x})$ (and its corresponding weight $w_j$) would affect the model's validation MSE. This provides a direct, interpretable measure of each term's contribution to the model's accuracy, akin to a leave-one-term-out analysis. This granular credit assignment drives the pruning phase (identifying which terms to retain or discard) and populates the history buffer. This enables the LLM to perform in-context learning, refining its future proposals based on the specific utility of past components.

## A.6. Relationship to Automated Feature Engineering

| Method | LLM / semantic signal | Generation mechanism | Feedback & selection | Typical predictor | Output representation |
|---|---|---|---|---|---|
| AutoFeat (Horn et al., 2019) | ✗ (no LLM; optional physical units) | Predefined analytic transforms and pairwise operators on numeric features | Correlation filtering and L1-regularised linear models with stability selection | Linear regression / classification | Small set of analytic feature expressions used in a linear model |
| OpenFE (Zhang et al., 2023) | ✗ | Expand–and–reduce over unary/binary operators (incl. aggregations and group-by) | FeatureBoost residual fitting + two-stage pruning with GBDT feature importance | GBDT / LightGBM | Augmented tabular feature matrix consumed by tree ensembles |
| RuleFit (Friedman & Popescu, 2008) | ✗ (no LLM; tree-ensemble rules) | Decision rules extracted from ensembles of trees and converted to binary features | Sparse linear model (e.g., L1-regularised regression) over rule indicators | Sparse linear model on rule features (rules derived from trees) | Set of human-readable decision rules with weights capturing non-linear interactions |
| CAAFE (Hollmann et al., 2023) | ✓ (LLM + dataset description) | LLM-generated Python feature-engineering code and textual explanations | Keep only transformations that improve validation metrics of downstream classifiers | Logistic regression, random forests, TabPFN | Dataset with additional engineered columns plus natural-language rationales |
| FeatLLM (Han et al., 2024) | ✓ (LLM + few-shot examples) | LLM-generated per-class logical rules converted to binary features | Linear model trained on rule indicators; learned weights used to rank rules | Linear classifier on rule features | Weighted rule set defining class-specific decision heuristics |
| OCTree (Nam et al., 2024) | ✓ (LLM + decision-tree reasoning) | LLM-generated rule-based column generators iteratively refined | Validation score of prediction model + CART-based natural-language reasoning | XGBoost, MLP, HyperFast | Set of rule-defined columns that boost generic tabular predictors |
| **IGSR** | ✓ (LLM + structure-guided feedback) | LLM-generated symbolic basis functions $\psi_j$ inside a linear equation | Global per-term influence scores $\Delta_j$ and pruning within a propose–and–prune loop | Sparse linear equation model | Closed-form symbolic equation $g(\mathbf{x}) = \sum_j w_j \psi_j(\mathbf{x})$ with ranked term contributions |

*Table 4.* Conceptual comparison between IGSR and representative automated feature engineering methods for tabular data. All approaches can be seen as generating additional features, but they differ in how the search space is specified, what feedback is used for refinement, which downstream predictors they target, and whether the final output is a generic feature set or an explicit symbolic equation.

Automated feature engineering (AFE) aims to construct informative transformations of raw inputs, because the performance of many machine learning models depends heavily on the quality of the input representation. Classical approaches include polynomial expansions and other predefined transformations, while deep neural networks can be viewed as implicitly performing feature engineering by learning hierarchical representations of the data. However, these learned representations are typically not symbolic and are difficult to interpret analytically.

While IGSR is framed as a symbolic equation discovery method, it can also be viewed as an automated feature discovery mechanism for linear models: the basis functions $\psi_j(\mathbf{x})$ play the role of engineered features and the symbolic equation $g(\mathbf{x}) = \sum_j w_j \psi_j(\mathbf{x})$ is a sparse linear model on top of them. This naturally connects IGSR to the literature on automated feature engineering and to tree-based tabular learners.

**Classical automated feature engineering.** AutoFeat (Horn et al., 2019) provides a scikit-learn style regression/classification interface in which a large pool of non-linear transformations of the inputs (e.g., polynomial, logarithmic and rational combinations) is constructed and then aggressively pruned using correlation filters and L1-regularised linear models with stability selection. The aim is to retain a small set of analytic features that make a linear model competitive while remaining globally interpretable, with optional use of physical units to filter dimensionally inconsistent expressions. OpenFE (Zhang et al., 2023) follows an expand–and–reduce strategy on tabular data: it enumerates candidate features using a library of unary and binary operators (including aggregations and group-by operators), and evaluates them via the FeatureBoost procedure together with a coarse-to-fine pruning scheme driven by gradient boosting decision trees (GBDTs) and tree-based feature attributions. In both cases, the search space is predefined, the optimisation is driven purely by predictive performance, and the final predictor is either a linear model (AutoFeat) or a tree ensemble (OpenFE).

**Rule-based feature generation.** Methods such as RuleFit (Friedman & Popescu, 2008) generate features in the form of decision rules. An ensemble of decision trees (e.g., gradient-boosted trees) is first trained on the original features, then each internal path is converted into a binary rule feature indicating whether a sample satisfies the corresponding conjunction of threshold conditions. A sparse linear model (typically with L1 regularisation) is then fitted on both the original features and these rule-based features. This yields an interpretable model that can capture non-linearities and feature interactions through a relatively small set of weighted rules derived from trees, sitting at the intersection of rule-based AFE and linear modelling.

**LLM-based feature generation.** Recent work combines AFE with large language models. CAAFE (Hollmann et al., 2023) prompts an LLM with a natural-language description of the dataset and prediction task and asks it to generate executable Python code that adds new columns, together with free-form textual justifications of their utility. The proposed transformations are accepted only if they improve validation performance of downstream classifiers such as logistic regression, random forests or TabPFN. FeatLLM (Han et al., 2024) targets few-shot tabular learning: an LLM is prompted with a task description and a small labelled set and asked to produce per-class rule sets (e.g., conjunctions of simple threshold conditions) that are converted into binary features. A linear model is then trained on these rule indicators, yielding both a prediction rule and a ranking of human-readable rules by their learned weights. OCTree (Nam et al., 2024) views feature engineering as optimisation over rule-based column generators. An LLM proposes rules for new columns, which are iteratively refined using two feedback channels: validation scores of prediction models such as XGBoost and MLPs, and natural-language decision-tree "reasoning" extracted from CART models fitted on the current dataset. Compared to classical AFE, these LLM-based methods replace a hand-crafted operator library with a language prior and use natural-language context (column names, descriptions, decision-tree paths) as additional guidance, while still primarily optimising black-box predictive performance on tabular benchmarks.

**IGSR as structure-guided feature engineering for symbolic equations.** Viewed through the AFE lens, IGSR also uses an LLM to propose new "features" $\psi_j(\mathbf{x})$ and keeps them only if they improve a downstream model. However, there are two key differences to the methods above. First, IGSR constrains the downstream model to be a sparse linear equation

$$g(\mathbf{x}) = \sum_{j=1}^{m} w_j \, \psi_j(\mathbf{x}),$$

which is itself the object of interest: the set of basis functions and their weights are meant to be read as a compact, global symbolic model of the system under study rather than as a feature set for an arbitrary classifier. In contrast to typical rule-generation algorithms that operate over simple threshold rules, IGSR's LLM-driven basis function proposal is not limited

to rules and can express complex symbolic compositions of inputs, including non-linear operations and domain-specific transformations that would be difficult to enumerate by hand. The subsequent linear combination $\sum_j w_j \psi_j(\mathbf{x})$ maintains global interpretability, while the influence-guided pruning ensures that only valuable, LLM-generated transformations are retained.

Second, the feedback that drives the search in IGSR is explicitly structure-aware: per-term influence scores $\Delta_j$ quantify the change in global mean-squared error when each term is removed. This allows the framework to deterministically prune ineffective terms and provides the LLM with detailed feedback on which specific transformations drove model performance, complementing black-box feature importance methods.

**Relationship to tree-based models and explainability.** Strong tree ensembles such as XGBoost (Chen & Guestrin, 2016) and LightGBM (Ke et al., 2017) remain among the best-performing predictors for tabular data and underpin many AFE systems, including OpenFE and OCTree. In these methods, interpretability is typically provided either at the feature level (via split counts or impurity-based importance) or through local explanation methods such as SHAP values computed on the engineered features. RuleFit also leverages tree ensembles, but distils their structure into a sparse linear model over rule indicators, bringing it closer in spirit to IGSR's linear formulation while still relying on tree-based rule extraction. IGSR instead optimises a single sparse linear equation and reports global term-level contributions through the coefficients $w_j$ and influence scores $\Delta_j$. This sacrifices some expressive power relative to deep ensembles of trees, but yields a model class in which the full predictive mechanism can be inspected analytically and related directly to scientific priors (e.g., additivity, saturation, or known limiting behaviours). Conceptually, IGSR is therefore complementary to tree-based pipelines: IGSR can be seen as a structure-guided feature generator whose discovered terms may also be fed into tree ensembles, while tree-based methods provide powerful black-box baselines against which the fidelity of IGSR's symbolic approximations can be assessed.

## A.7. Iterative Refinement and Search Strategies in Complex Spaces

Discovering optimal symbolic equations is a challenging search problem over a vast and complex space.

**Iterative Refinement and Feedback:** Many AI systems employ iterative refinement, where solutions are progressively improved based on feedback. The nature and granularity of this feedback are crucial. IGSR's iterative propose-and-prune cycle, guided by per-term influence scores, provides a structured mechanism for self-improvement. The LLM learns from its past decisions (which terms were kept/dropped and their impact) and the specific utility of each component in the current proposal. This contrasts with approaches where feedback is only a single scalar loss, offering less guidance for targeted improvement. LLM self-correction and refinement, often through in-context learning, is an active area of research (Madaan et al., 2023; Shinn et al., 2023). IGSR provides a domain-specific instantiation of this principle with a highly structured feedback signal.

**Monte Carlo Tree Search (MCTS):** For more systematic exploration, IGSR can integrate its propose-and-prune cycle into an MCTS framework. MCTS is a heuristic search algorithm that has achieved remarkable success in domains like game playing (e.g., AlphaGo/AlphaZero (Silver et al., 2016; 2017)) and other optimization problems. In IGSR, MCTS treats the equation discovery process as navigating a tree where nodes represent (sets of) basis functions and edges represent propose/prune actions. MCTS balances exploration of new equation structures with exploitation of promising ones. While MCTS has been explored for program synthesis and, more recently, symbolic regression (Shojaee et al., 2023; Lu et al., 2021), IGSR's novelty lies in combining MCTS with LLM-driven proposal/pruning stages that are informed by the detailed influence-based feedback. The work (Shojaee et al., 2023) also uses MCTS with transformers for symbolic regression, but their feedback and reward structure within MCTS differs from IGSR's influence-score-driven pruning by an LLM agent.

By leveraging detailed, interpretable feedback and powerful search strategies, IGSR aims to make significant strides in the automated discovery of accurate and understandable symbolic models from data.

## A.8. Extended Comparison of Modeling Paradigms

To further contextualize Influence-Guided Symbolic Regression (IGSR) within the broader landscape of machine learning and scientific discovery, we provide an extended comparison of modeling paradigms in Table 5.

While Table 1 highlights the fundamental methodological differences, specifically regarding feedback mechanisms and output interpretability, this supplementary analysis focuses on the operational characteristics of each approach. We categorize

*Table 5.* **Extended Comparison: Context and Implementation.** This table provides complementary details to Table 1, focusing on data requirements, system types, search mechanisms, and typical domains. Note that while IGSR and AFE both handle feature generation, they serve distinct operational goals (Scientific Discovery vs. Predictive Accuracy on Tabular Data).

| Method Paradigm | Typical Data Regime | System Type Focus | Primary Search Mechanism | Typical Applications |
|---|---|---|---|---|
| Classical Symbolic Reg. (e.g., GP-SR) [1] | Small-Med | Static | Genetic / Evolutionary | Physics laws, simple regressions |
| Black-Box ML (NN, GBDT) [2] | Large | Static / Dyn | Gradient / Greedy Tree | Image recog., Tabular prediction |
| Neural ODEs [3] | Large | Dynamical | Gradient Descent | Modeling physical processes |
| Automated Feat. Eng. (AFE) [4] | Med-Large | Static | Expansion & Filtering | Kaggle/Industry Tabular tasks |
| LLM for Equations (e.g., D3, LLM-SR) [5] | Small-Med | Static / Dyn | Iterative Refinement | Bio/Pharm dynamics discovery |
| **IGSR** | **Small-Med** | **Static / Dyn** | **LLM Agent + MCTS** | **Genomics, PKPD, Biomarkers** |

**References:** [1] (Stephens, 2015; Cranmer, 2023), [2] (Chen & Guestrin, 2016; Gorishniy et al., 2021), [3] (Chen et al., 2018), [4] (Horn et al., 2019; Zhang et al., 2023), [5] (Holt et al., 2024; Shojaee et al., 2025a)
**Search Mechanisms:** *Evolutionary*: Population-based mutation/crossover. *Gradient*: Differentiable backpropagation. *MCTS*: Monte Carlo Tree Search.

methods based on their typical data regimes (e.g., small-data scientific discovery versus large-scale industrial prediction), their suitability for static versus dynamical systems, and the primary search algorithms they employ.

This breakdown clarifies the distinction between IGSR and adjacent fields like Automated Feature Engineering (AFE). While both paradigms leverage generative mechanisms to construct new features, they serve fundamentally different objectives. AFE frameworks typically target predictive maximization in static, large-scale tabular tasks, often producing augmented feature matrices for downstream black-box models. In contrast, IGSR is explicitly designed for scientific discovery in high-dimensional, often data-constrained environments, utilizing LLM-driven search to distill complex relationships into concise, interpretable symbolic equations.

## B. Method Details

This section provides further details on the Influence-Guided Symbolic Regression (IGSR) framework, complementing the main description in Section 2. We elaborate on the core components, including the definition and calculation of influence scores, and the specifics of the Monte Carlo Tree Search (MCTS) implementation.

### B.1. Overview of IGSR

As outlined in Section 2 and depicted in Figure 1, IGSR operates as an iterative symbolic modeling pipeline driven by Large Language Models (LLMs).

The framework employs the LLM agents:

- A **"Propose" Agent**: This LLM agent is responsible for generating new candidate basis functions $\psi_j(\mathbf{x})$. It receives contextual information about the dataset, the scientific problem, the current set of basis functions, the best equation discovered so far, and a history of past decisions and outcomes to guide its suggestions. This agent is used in all IGSR variants.

- An optional *"Prune" Agent* in *IGSR-Agent* variant only: After candidate terms are evaluated, this LLM agent refines the set of basis functions. It is provided with the candidate terms, their fitted weights $w_j$, and crucially, their per-term influence scores $\Delta_j$. Based on this granular feedback and its general knowledge, it decides which terms to keep or discard. This agent is not used in the default IGSR variant, which used deterministic pruning instead.

The propose-and-prune cycle forms the core iterative step. While IGSR can execute these cycles in a linear iterative refinement loop, a more efficient search is achieved when integrating them within a Monte Carlo Tree Search (MCTS) framework for a more systematic and robust exploration of the equation space. MCTS is used as the default search strategy in IGSR. Any experiments or ablations that use the linear iterative refinement variant are labeled as the *Iterative* variant of the method.

The overall objective is to discover accurate and interpretable models of the form $f(\mathbf{x}) = \sum_j w_j \psi_j(\mathbf{x})$. The hyperparameters related to LLM prompting, such as `terms_per_round` (default: 5), `first_round_n_candidates` (default: 10), and `keep_n_terms` (default: 6, but can be disabled to allow keeping *any* number of terms), guide the framework's behavior during term generation and pruning.

### B.2. Influence Score ($\Delta_j$) Details

The per-term influence score, $\Delta_j$, is a cornerstone of the IGSR framework, providing the granular signal necessary for the pruning step: deterministically in default *IGSR* and via prompting for "Prune" LLM agent to make informed decisions in the *IGSR-Agent* variant.

**Definition**: For a linear model $f(\mathbf{x}) = \sum_{k=1}^{M} w_k \psi_k(\mathbf{x})$ with $M$ basis functions, the influence score $\Delta_j$ for a specific term $\psi_j(\mathbf{x})$ is defined as the change in the model's Mean Squared Error (MSE) on the validation set if that term were removed from the model, *while all other term weights $w_k$ ($k \neq j$) remain fixed at their originally fitted values*.

Let $\mathbf{\Phi}_{\text{val}}$ be the design matrix evaluated on the validation set using all $M$ candidate basis functions, and $\mathbf{y}_{\text{val}}$ be the corresponding true target values. Let $\mathbf{w} = (w_1, w_2, \ldots, w_M)^T$ be the vector of weights obtained by fitting the full model to the training data (e.g., using Ordinary Least Squares, OLS). The prediction of the full model on the validation set is $\hat{\mathbf{y}}_{\text{val}} = \mathbf{\Phi}_{\text{val}} \mathbf{w}$. The MSE of the full model on the validation set is $\text{MSE}_{\text{full}} = \text{mean}((\mathbf{y}_{\text{val}} - \hat{\mathbf{y}}_{\text{val}})^2)$.

Now, consider removing term $\psi_j$. This is equivalent to setting its weight $w_j$ to zero. The predictions of this reduced model, $f_{-j}(\mathbf{x})$, on the validation set are $\hat{\mathbf{y}}_{\text{val},-j} = \hat{\mathbf{y}}_{\text{val}} - \phi_{\text{val},j} w_j$, where $\phi_{\text{val},j}$ is the $j$-th column of $\mathbf{\Phi}_{\text{val}}$ (i.e., the evaluations of $\psi_j(\mathbf{x})$ on the validation set). The MSE of this reduced model is $\text{MSE}_{-j} = \text{mean}((\mathbf{y}_{\text{val}} - \hat{\mathbf{y}}_{\text{val},-j})^2)$. The influence score $\Delta_j$ is then:

$$\Delta_j = \text{MSE}_{-j} - \text{MSE}_{\text{full}}$$

A higher positive $\Delta_j$ indicates that removing term $\psi_j$ increases the validation MSE, implying that the term is important for the model's predictive accuracy on unseen data. A $\Delta_j \approx 0$ suggests the term has little unique contribution to reducing validation MSE under the current model. This calculation is performed for each target variable if the model is multi-output.

For a rigorous derivation of the influence score on validation data (where the orthogonality of residuals assumed in the simplified OLS formula does not strictly hold) and an empirical comparison against refit-aware alternatives, please refer to Appendix G.11.

**Calculation**: The calculation steps are:

1. Fit the full linear model (e.g., OLS) using the training data $(\mathbf{\Phi}_{\text{train}}, \mathbf{y}_{\text{train}})$ to obtain the weights $\mathbf{w}$.

2. Calculate the predictions $\hat{\mathbf{y}}_{\text{val}}$ and the baseline $\text{MSE}_{\text{full}}$ on the validation set $(\mathbf{\Phi}_{\text{val}}, \mathbf{y}_{\text{val}})$.

3. For each term $\psi_j$ (from $k = 1$ to $M$):
   (a) Calculate the predictions of the model without term $j$ (effectively $w_j = 0$, other $w_k$ fixed): $\hat{\mathbf{y}}_{\text{val},-j} = \hat{\mathbf{y}}_{\text{val}} - \phi_{\text{val},j} w_j$.
   (b) Calculate $\text{MSE}_{-j}$ using $\hat{\mathbf{y}}_{\text{val},-j}$.
   (c) Compute $\Delta_j = \text{MSE}_{-j} - \text{MSE}_{\text{full}}$.

**Computational Cost**:

- Fitting the initial OLS model: Typically $O(N_{\text{train}} M^2 + M^3)$ if $N_{\text{train}} > M$ or $O(N_{\text{train}}^2 M)$ if $M > N_{\text{train}}$.

- Calculating $\hat{\mathbf{y}}_{\text{val}}$: $O(N_{\text{val}} M)$.

- Calculating $\text{MSE}_{\text{full}}$: $O(N_{\text{val}})$ (assuming multi-output $m$ is small, otherwise $O(N_{\text{val}} m)$).

- For each of the $M$ terms, calculating $\Delta_j$:

  - Prediction $\hat{\mathbf{y}}_{\text{val},-j}$ adjustment: $O(N_{\text{val}})$.
  - $\text{MSE}_{-j}$ calculation: $O(N_{\text{val}})$.
  - Total for all $\Delta_j$: $M \cdot O(N_{\text{val}})$.

The calculation of influence scores is therefore efficient once the initial model is fitted and its predictions on the validation set are obtained.

**Justification and Relation to Other Measures**:

- **Direct Relevance to Predictive Performance**: $\Delta_j$ directly quantifies how much a term contributes to reducing error on unseen (validation) data, which is a primary goal.

- **Efficiency**: Calculating $\Delta_j$ by fixing other weights is much more computationally efficient than refitting the model $M$ times (once for each term's removal). This makes it practical for iterative refinement loops with many candidate terms.

- **Interpretability for LLM Guidance**: The concept of "change in error if term is removed" is intuitive and can be effectively communicated to an LLM, especially with heuristics like "$\Delta_j \approx 0 \implies$ drop". This is applicable to the *IGSR-Agent* variant.

- **Differentiation from other importance measures**:
  - It is a form of "leave-one-out" importance but applied to model terms (basis functions) rather than individual data points (like statistical influence functions, e.g., Cook's distance).
  - It differs from SHAP values (Lundberg & Lee, 2017), which explain the contribution of features to individual predictions rather than the global impact of a term on overall model MSE.
  - It is distinct from feature importance measures derived from tree-based ensembles (e.g., Gini importance or permutation importance on raw input features), as $\Delta_j$ is specific to the contribution of pre-defined or LLM-proposed basis functions $\psi_j(\mathbf{x})$ within a linear model structure.
  - By not refitting the model for each term removal, $\Delta_j$ measures the unique contribution of a term *given the current set of other terms and their weights*. This is a deliberate choice to assess the marginal utility of a term in the specific context of the current full model. If terms are highly collinear, this score might be low for some of them, aiding in pruning redundant terms.

The use of validation data for calculating $\Delta_j$ is critical for assessing generalization and mitigating overfitting, guiding the LLM to select terms that are robustly beneficial.

We further explore and experimentally investigate alternative approaches to computing influence scores in Appendix G.11.

### B.3. Deterministic Pruning Strategy

The IGSR framework is inherently modular. *IGSR-Agent* variant employs a dual-agent architecture – using a "Propose" LLM to generate terms and a "Prune" LLM to refine them. The "Prune" agent in this case is tasked with interpreting the per-term influence scores ($\Delta_j$) alongside domain descriptions to make selection decisions. However, in common scenarios where predictive performance, computational efficiency, and immunity to potential LLM hallucinations are priority over provision of domain/dataset knowledge at the pruning step, the pruning step can be purely deterministic. This is the default approach adopted by *IGSR*. Instead of the "Prune" LLM agent, a deterministic selection mechanism based strictly on the quantitative feedback provided by the influence scores is used. The core iterative cycle remains largely unchanged (see Section 2.1), but the *Term Pruning Phase* is purely algorithmic and does not employ an LLM. Since $\Delta_j$ explicitly quantifies the marginal contribution of basis function $\psi_j$ to the validation accuracy, it serves as a robust ranking metric.

In multi-output scenarios where the model predicts multiple target variables simultaneously, a single term $\psi_j$ may have a distinct influence score for each target. To ensure that terms critical to *any* dimension of the system are preserved, we define the aggregate influence of a term as its maximum influence across all targets: $\Delta_j^{\mathrm{agg}} = \max_m(\Delta_{j,m})$. Using this aggregate metric, we define two deterministic pruning strategies:

1. **Top-K Selection:** The candidate terms are ranked in descending order of their aggregate influence scores $\Delta_j^{\mathrm{agg}}$. The top $K$ terms are retained, where $K$ corresponds to the sparsity constraint (the `keep_n_terms` hyperparameter). This is approach we employ in this work.

2. **Threshold Selection:** Terms are retained if their aggregate influence score exceeds a specific threshold, $\Delta_j^{\mathrm{agg}} > \epsilon$. This allows the model complexity to adapt dynamically to the signal-to-noise ratio of the discovered terms. This is an alternative approach worth exploring, but we do not investigate it in this work.

This approach effectively decouples the generative creativity of the LLM (used in the "Propose" phase) from the rigorous, data-driven selection process. It eliminates the token cost and latency associated with the second LLM call and removes the possibility of the LLM mistakenly discarding a high-influence term due to a failure in reasoning or instruction following. However, it does not allow for potentially more sophisticated reasoning that LLMs *may* employ at the pruning stage, or for provision of domain knowledge (e.g. "prefer these kinds of terms because of scientific reason X") during pruning.

### B.4. Monte Carlo Tree Search (MCTS) Implementation Details

IGSR utilizes a Monte Carlo Tree Search (MCTS) algorithm to navigate the complex hypothesis space of symbolic equations. This approach allows for a more structured exploration than simple iterative refinement.

- **State (Node Representation)**: Each node in the MCTS tree represents a specific state in the equation discovery process. This state is defined by the current set of selected basis functions $\{\psi_j(\mathbf{x})\}$, the corresponding fitted weights $\{w_j\}$ that form the current best equation for that path, and the history of decisions (proposals, prunings, and feedback) that led to this state.

- **Action (Transition)**: An action involves transitioning from a parent node (current equation state) to a child node (a refined equation state). This transition is achieved by executing one full **propose-and-prune cycle** as described in Section 2.1. Specifically, from a selected leaf node, the "Propose" agent suggests new terms, these are evaluated, influence scores are calculated, and the pruning step refines the term set. This results in a new equation that defines a child node.

- **Expansion**: During the expansion phase of MCTS, if a selected leaf node is not a terminal state (e.g., maximum depth not reached), child nodes are generated. IGSR can generate multiple distinct successor states (child nodes) from a single parent node by repeatedly invoking the propose-and-prune cycle. The hyperparameter `n_successors` (default: 5) controls how many child nodes are attempted to be generated from a parent node during a single expansion step.

Variation between these successors arises from the inherent stochasticity in the LLM's responses to the propose (and, optionally, prune) prompts, even when given the same historical context up to the parent node. Each such generated child represents a distinct path for exploration.

- **Reward and Value Estimation (Simulation/Rollout Phase)**: The quality or *value* of a node (equation) needs to be estimated to guide the MCTS search.

  - In the default IGSR configuration, referred to as **Heuristic MCTS** (when the hyperparameter `rollout_is_just_node_reward` is set to `True`, which is the default), no explicit rollout (simulation beyond the newly expanded node) is performed. Instead, the reward for a newly expanded child node is its immediate, directly computed quality. This quality is typically the negative validation Mean Squared Error ($-\text{MSE}_{\text{val}}$) of the equation associated with that child node (after the pruning step). A higher value (lower MSE) indicates a better node. This immediate reward is then directly backpropagated up the tree.
  - If `rollout_is_just_node_reward` is set to `False`, a simulation phase (rollout) would be executed from the newly expanded node for a number of steps defined by `rollout_depth` (default: 1, see Appendix G.4 for an investigation). The reward from the end of this simulated trajectory would then be backpropagated. However, the primary results in this paper use the Heuristic MCTS approach.
  - Other reward signals beyond simply the negated validation set MSE could be used, though are not explored here, e.g. incorporating alternative accuracy metrics or accounting for equation parsimony (e.g. as $R = -\text{MSE}_{\text{val}} - \alpha \cdot Complexity$, where complexity may be defined as the number of terms, operator count, etc.).

- **Backpropagation**: After a node's value is determined (either through direct evaluation in Heuristic MCTS or via a rollout), this value is backpropagated up the tree from the expanded node to the root. The visit counts and average reward (or value) of all visited nodes along the path are updated.

- **Selection (Tree Policy)**: The MCTS algorithm uses the Upper Confidence Bound applied to Trees (UCT) formula to balance exploration and exploitation when selecting which node to traverse down the tree. From a current node, the child $i$ selected is the one that maximizes:

$$\text{UCT}(i) = Q(i) + c\sqrt{\frac{\ln N(p)}{N(i)}}$$

  where:

  - $Q(i)$ is the current estimated average reward (exploitation term) for child $i$.
  - $N(p)$ is the number of times the parent node $p$ has been visited.
  - $N(i)$ is the number of times child node $i$ has been visited.
  - $c$ is the exploration constant (hyperparameter, default: $\sqrt{2}$). A higher $c$ encourages more exploration of less-visited nodes.

  This selection process is repeated from the root until a leaf node (a node that has not been expanded, or a terminal node) is reached.

- **Termination**: The MCTS process continues until a predefined computational budget is exhausted. This is primarily controlled by the `total_budget` hyperparameter (default: 30), which defines the total number of MCTS iterations (selection, expansion, simulation/evaluation, backpropagation cycles). A `depth_limit` (default: 10) can also restrict the maximum depth of the tree.

- **Output**: After the MCTS process completes, the equation corresponding to the node that yielded the best validation MSE throughout the entire search is typically selected as the final discovered model.

- **Computational Cost**: The main computational costs in the MCTS-based IGSR are the LLM API calls (for term proposal and pruning at each expansion, potentially multiple times if `n_successors > 1`), the evaluation of basis functions on data, and the fitting of linear models. The total number of MCTS iterations (`total_budget`) is the primary driver of the overall computational expense.

This MCTS framework allows IGSR to systematically explore diverse pathways of equation refinement, potentially avoiding local optima that a simpler iterative approach might encounter.

## B.5. IGSR Pseudocode

Below is a high-level pseudocode outlining the IGSR framework. Algorithm 1 describes the overall MCTS-driven search. Algorithm 2 details the core propose-and-prune cycle (PAPC).

For brevity, Algorithm 1 illustrates the Heuristic MCTS case, but this can be easily adapted to the rollout with `rollout_depth` case.

---

**Algorithm 1** Influence-Guided Symbolic Regression (IGSR) - MCTS Loop

---

1: **Input:** Dataset $\mathcal{D}$ (containing train, validation, test sets), Problem Description $P_d$, MCTS Budget $B_{\text{MCTS}}$, LLM Agents ($LLM_P, [LLM_R]$), MCTS Parameters (exploration const $c$, num successors $n_s$, depth limit $d_{lim}$)
2: **Output:** Best discovered symbolic equation $f_{\text{best}}$
3: Initialize $f_{\text{best}} \leftarrow$ None
4: Initialize $best\_val\_mse \leftarrow \infty$
5: Initialize MCTS Tree $T$ with a root node (representing an initial empty state)
6: **for** $iteration = 1$ to $B_{\text{MCTS}}$ **do**
7:    $selected\_node \leftarrow$ SelectPolicy$(T, c)$ {Select a node using UCT}
8:    **if** $selected\_node$ is suitable for expansion (e.g., not terminal by $d_{lim}$, and not yet fully expanded for $n_s$ children) **then**
9:      *// Attempt to expand selected_node by generating up to $n_s$ children*
10:      **for** $i = 1$ to $n_s$ **do**
11:        $PAPC\_Inputs \leftarrow$ GatherInputsForPAPC$(selected\_node, P_d, \mathcal{D}, LLM_P[, LLM_R], f_{\text{best}})$
12:        $equation, val\_mse, new\_terms, history \leftarrow$ ProposeAndPruneCycle$(PAPC\_Inputs)$
13:        **if** $equation$ was successfully generated **then**
14:          $child\_node \leftarrow$ AddChildNode$(T, selected\_node, new\_terms, equation, val\_mse, history)$
15:          $current\_reward \leftarrow -val\_mse$   *// Heuristic reward for UCT*
16:          BackpropagateValue$(child\_node, current\_reward, T)$
17:          **if** $val\_mse < best\_val\_mse$ **then**
18:            $best\_val\_mse \leftarrow val\_mse$
19:            $f_{\text{best}} \leftarrow equation$
20:          **end if**
21:        **end if**
22:      **end for**
23:      Update expansion status of $selected\_node$ in $T$.
24:    **else**
25:      $current\_reward \leftarrow -selected\_node.stored\_val\_mse$   *// Use node's known value*
26:      BackpropagateValue$(selected\_node, current\_reward, T)$
27:    **end if**
28: **end for**
29: **if** $f_{\text{best}}$ is not None **then**
30:    Evaluate $f_{\text{best}}$ using test set from $\mathcal{D}$
31: **end if**
32: **return** $f_{\text{best}}$

---

---

**Algorithm 2** Propose-and-Prune Cycle

---

1: **Input:** $current\_terms_{in}$, $history_{in}$, $best\_equation\_so\_far$, $P_d$, $\mathcal{D}_{train}$, $\mathcal{D}_{val}$, $LLM_P$, $[LLM_R]$, $PruningMode$ (Deterministic/Agentic), $K$ (keep_n_terms)
2: **Output:** $equation_{out}$, $val\_mse_{out}$, $final\_terms_{out}$, $history_{out}$
3:
4: *// Term Generation Phase*
5: Prompt $LLM_P$ with $P_d$, current active terms $current\_terms_{in}$, $best\_equation\_so\_far$, $history_{in}$, and data preview.
6: $newly\_proposed\_terms \leftarrow LLM_P.generate\_terms()$
7: $candidate\_terms \leftarrow current\_terms_{in} \cup newly\_proposed\_terms$
8:
9: *// Candidate Evaluation and Feedback Preparation*
10: Evaluate all $\psi_j \in candidate\_terms$ on $\mathcal{D}_{train}$ to get design matrix $\mathbf{\Phi}_{train}$.
11: Fit linear model: $\mathbf{y}_{train} \approx \mathbf{\Phi}_{train}\mathbf{w}_{cand}$ to get candidate weights $\mathbf{w}_{cand}$.
12: Evaluate all $\psi_j \in candidate\_terms$ on $\mathcal{D}_{val}$ to get $\mathbf{\Phi}_{val}$.
13: Calculate full validation MSE: $\text{MSE}_{val,cand}$ using $\mathbf{w}_{cand}$ and $(\mathbf{\Phi}_{val}, \mathbf{y}_{val})$.
14: For each $\psi_j \in candidate\_terms$, calculate influence score $\Delta_j$ on validation set (as per Appendix B.2).
15:
16: *// Term Pruning Phase*
17: **if** $PruningMode$ is **Deterministic** (Default IGSR) **then**
18:     For each term $j$, compute aggregate influence: $\Delta_j^{\text{agg}} \leftarrow \max_m \Delta_{j,m}$ (across targets $m$).
19:     Sort $candidate\_terms$ by $\Delta_j^{\text{agg}}$ in descending order.
20:     $final\_terms_{out} \leftarrow$ Select top $K$ terms from the sorted list.
21: **else** {$PruningMode$ is **Agentic** (IGSR-Agent)}
22:     Prompt $LLM_R$ with $P_d$, $candidate\_terms$, weights $w_j$, influence scores $\Delta_j$, $\text{MSE}_{val,cand}$, $history_{in}$.
23:     $pruning\_decisions \leftarrow LLM_R.decide\_terms\_to\_keep\_drop()$   *// {keep: [...], drop: [...]}*
24:     $final\_terms_{out} \leftarrow$ terms marked "keep" in $pruning\_decisions$
25: **end if**
26:
27: *// Final Model Fitting and State Update*
28: Evaluate $final\_terms_{out}$ on $\mathcal{D}_{train}$ to get $\mathbf{\Phi}_{train,final}$.
29: Fit final linear model: $\mathbf{y}_{train} \approx \mathbf{\Phi}_{train,final}\mathbf{w}_{final}$ to get $\mathbf{w}_{final}$.
30: Form $equation_{out}$ using $final\_terms_{out}$ and $\mathbf{w}_{final}$.
31: Calculate $val\_mse_{out}$ for $equation_{out}$ on $\mathcal{D}_{val}$.
32: Record this cycle's details (terms before/after pruning, decisions/scores, MSEs, equation) into $history_{out}$.
33:
34: **return** $equation_{out}$, $val\_mse_{out}$, $final\_terms_{out}$, $history_{out}$

---

This pseudocode provides a conceptual blueprint. Actual implementations would involve detailed prompt engineering, error handling, and specific choices for the linear model fitting procedure (e.g., OLS, Ridge).

## B.6. LLM Details

Unless otherwise specified (e.g., in Appendix G.6 which details experiments across a range of different Large Language Models), the primary LLM employed for the core experiments presented in this work was a version of GPT-4o, specifically:

- `GPT-4o`: model identifier `gpt-4o`, version `2024-11-20`.

A notable exception is for LLM-SRBench evaluations, where GPT-4o-mini was used in order to manage costs (since the benchmark contains a large number of problems). Specifically:

- `GPT-4o-mini`: model identifier `gpt-4o-mini`, version `2024-07-18`.

## B.7. Prompt Details

The interaction with the LLM is central to the IGSR framework. We employ two primary types of interactions, mediated by distinct LLM agents: one for proposing new candidate basis functions (the "Propose" agent) and another for refining the set of terms based on evaluation feedback (the "Prune" agent), the latter used in the *IGSR-Agent* variant of the method only. Below, we detail the structure of these prompts and provide illustrative examples of LLM interactions.

### B.7.1. TERM GENERATION PHASE: THE "PROPOSE" AGENT

In this phase, the LLM is tasked with generating new candidate basis functions $\psi_j(\mathbf{x})$. The prompt provides comprehensive context, including the problem description, available features, the current set of basis functions (if any), the best equation found so far, and a history of previous interactions to facilitate learning from past attempts.

**Prompt Template for "Propose" Agent:** The following is a representative template for the prompt provided to the "Propose" agent. Specific details such as dataset descriptions, feature lists, and historical performance are dynamically inserted.

*Listing 1.* Prompt Template for "Propose" Agent

```
You are an automated assistant for proposing linear terms for the equations in a symbolic regression pipeline.

Your proposed terms will be:
1. Concatenated with the current candidate terms.
2. Sent to a LLM term pruner agent that will use various computed signals to decide which terms to keep and which to drop.

# Instructions:
Given below information, propose some candidate terms. The terms can be any valid numpy expressions.
Make use of the dataset and problem description to propose relevant terms.
Make sure to use the learnings from the history of previous rounds.

Return these between triple backticks and one term on each line.
The first backticks must be prepended with TERMS

Example output:

TERMS
```
x1
x2**2
np.sin(x3)
```

NOTES:
* Propose around {terms_per_round} terms, generally not too many unless this is the first round.
* If this *is* the first round, propose around {first_round_n_candidates} terms.
* You may propose no terms (nothing between the TERMS backticks) if you think it is appropriate.

DATASET AND PROBLEM DESCRIPTION
----------------------------
{dataset_and_problem_description}

CURRENT TERMS:
----------------------------
{current_terms}

# Current equation:
{current_equation}

HISTORY
----------------------------
{history}

=========
The input data and target variable(s) preview:
{input_preview}
```

The prompt template for the "Propose" agent dynamically incorporates several pieces of information. Key placeholders include:

**terms_per_round**: Hyperparameter specifying how many new basis functions the LLM should suggest in the current round. Default value: 5.

**first_round_n_candidates**: Hyperparameter specifying how many new basis functions the LLM should suggest in the first round. Default value: 10.

**dataset_and_problem_description**: Provides the LLM with context about the scientific problem and data, e.g.:

```
Prediction of Treatment Response for Combined Chemo- and Radiation-Therapy
for Non-Small Cell Lung Cancer Patients Using a Bio-Mathematical Model

Here you must model the state differential of **cancer_volume**, and **chemo_concentration**, which are **dv_dt** and **dc_dt**, driven
    by the input actions **chemo_dosage** and **radiotherapy_dosage**, together with relevant
laboratory and coagulation markers listed below.

Description of the variables
─────────────────────────────
* cancer_volume        : Volume of the tumour (cm^3)
* chemo_concentration  : Plasma concentration of vinblastine (mg m^-3)
* chemo_dosage         : Administered vinblastine dose rate (mg m^-3 day^-1)
* radiotherapy_dosage  : Delivered external-beam RT dose rate (Gy day^-1)

Time unit: **days**

Typical value ranges
──────────────────────
* cancer_volume       : 0.01433 - 1170.861 cm^3
* chemo_concentration : 0 - 9.9975 mg m^-3
* chemo_dosage        : 0 - 5 mg m^-3 day^-1
* radiotherapy_dosage : 0 - 2 Gy day^-1

Dataset summary
─────────────────
The training dataset consists of **1000** patients, each
followed for 60 days with daily resolution (delta_t = 1 day). Continuous variables are recorded once per day unless otherwise specified
    .
```

**current_terms**: Informs the LLM about terms already part of the active model in iterative steps, e.g.:

```
['cancer_volume', 'chemo_dosage', 'cancer_volume * chemo_concentration', 'cancer_volume * radiotherapy_dosage', 'np.log(cancer_volume +
    1)', 'np.sqrt(cancer_volume)']
```

**current_equation**: Shows the LLM the current best-performing model to potentially build upon or deviate from, e.g.:

```
dv_dt = 0.009194 cancer_volume - 0.0004414 chemo_dosage - 0.02797 cancer_volume * chemo_concentration - 0.04775 cancer_volume *
    radiotherapy_dosage - 1.214 np.log(cancer_volume + 1) + 1.202 np.sqrt(cancer_volume)
dc_dt = - 0.004518 cancer_volume + 0.9776 chemo_dosage - 0.001191 cancer_volume * chemo_concentration - 3.691e-06 cancer_volume *
    radiotherapy_dosage - 1.442 np.log(cancer_volume + 1) + 0.5209 np.sqrt(cancer_volume)
```

**history**: Gives a history of terms decisions, and performance, enabling in-context learning, e.g.

```
Round node_0_0:  KEEP=['cancer_volume', 'cancer_volume * chemo_concentration', 'cancer_volume * radiotherapy_dosage', 'np.log(
    cancer_volume + 1)', 'np.sqrt(cancer_volume)', 'chemo_dosage']  |  DROP=['chemo_concentration', 'radiotherapy_dosage', '
    chemo_dosage * radiotherapy_dosage', 'cancer_volume * chemo_dosage', 'cancer_volume * radiotherapy_dosage', 'np.log(cancer_volume
    + 1)', 'np.sqrt(cancer_volume)', 'radiotherapy_dosage']  |  MSE before pruning=0.005453 (per-output=[0.010905755578988435,
    6.33977252002678e-27])  |  MSE after pruning=0.900026 (per-output=[0.011005448429060318, 1.789047395904737])
Round node_0_0_1: ...
```

**Example LLM Response for "Propose" Agent:** Below is an example response from the LLM acting as the "Propose" agent, based on the template above.

*Listing 2.* Example LLM Response from "Propose" Agent

```
TERMS
```
cancer_volume
chemo_concentration
chemo_dosage
radiotherapy_dosage
np.log(cancer_volume + 1)
np.sqrt(cancer_volume)
chemo_dosage * radiotherapy_dosage
cancer_volume * chemo_concentration
radiotherapy_dosage / (cancer_volume + 1)
chemo_dosage / (chemo_concentration + 1)
```
```

### B.7.2. TERM PRUNING PHASE: THE "PRUNE" AGENT (IGSR-AGENT VARIANT)

After new terms are proposed and a model is fitted using the expanded set of basis functions, the "Prune" agent is invoked. This agent receives the full list of candidate terms, their fitted weights $w_j$, their per-term influence scores $\Delta_j$ (e.g., change in validation MSE if the term is removed), and the overall validation MSE. Its task is to decide which terms to keep or discard.

**Prompt Template for "Prune" Agent:** The following template illustrates the prompt given to the "Prune" agent.

*Listing 3.* Prompt Template for "Prune" Agent

```
You are an equation-pruning assistant for symbolic regression.

=======================================================
INPUT YOU RECEIVE
=======================================================

* A table (or dictionary/json representation) where each row has:

| field     | meaning                                                        |
|-----------|----------------------------------------------------------------|
| term      | Name of the symbolic basis function psi_k(x) (e.g. "x1", "sin(x2)", ...). |
| weight    | Fitted scalar coefficient w_k obtained by ordinary least squares (OLS).   |
| influence | Influence score for term k (see definition below).             |

* The validation set MSE.

**Influence definition (no refit)**

delta_k is the increase in mean-squared-error (MSE) if the k-th weight is deleted while all other weights stay fixed. For OLS this is

    delta_k = (w_k^2 / n) * sum phi_k(x_i)^2  (always >= 0).

* Note that the influence values are computed on the validation (rather than training) set, and thus may not always be >= 0.

Hence:
* If influence is large -> the term is important (its removal hurts the loss a lot).
* If influence ~= 0 -> the term is useless (its removal makes no noticeable difference).

=======================================================
YOUR TASK
=======================================================
1. **Inspect every row**.
2. **Decide "keep" or "drop"** for each term using the rule:

* Use the heuristic: "delta_k ~= 0 -> drop", "large delta_k -> keep" and your own judgement.

3. **Return** a python dictionary after "DECISION" with exactly the two keys

DECISION
```
{{
    "keep":  ["term_a", "term_b", ...],
    "drop":  ["term_c", "term_d", ...]
}}
````

Place each term name in either **keep** or **drop** - never both, never neither.

**IMPORTANT:**
* Make use of the dataset and problem description to make the best decision.
* Make sure to use the learnings from the history of previous rounds.
* (!) You must consider the generalization beyond the validation set and make decisions accordingly.
* (!) You should also consider BOTH the weights and the influence of the terms.
* (!) You MUST keep {keep_n_terms} terms at most, to keep the model interpretable.

_______________________________________________
CONVENTIONS & NOTES
_______________________________________________

# Important notes
* Treat terms independently; no need to refit or update weights.
* Note that everything was evaluated on the validation set to avoid overfitting.
* If there are multiple outputs (targets), you will see multiple tables, one for each target.
* Use all the information available, but keep in mind that you must only return one keep/drop decision even if there are multiple
    outputs.
* Keep only the most important terms for each output.

# Output format
* Feel free to comment briefly (<= 30 chars) about each decision, but keep the python dictionary in the right format.
* The dictionary MUST be provided between triple backticks, otherwise it cannot be parsed.
* It must be prepended with "DECISION", otherwise it cannot be parsed.

_______________________________________________
EXAMPLE
_______________________________________________

# INPUT TABLE(S):

y_1:
| term   | weight | influence |
| ------ | ------ | --------- |
| x1     | 3.00   | 12.21     |
| x2     | -1.96  | 5.14      |
| x1**2  | 0.53   | 0.91      |
| sin    | 0.93   | 0.52      |
| cos    | -0.05  | 0.0009    |

y_2:
| term   | weight | influence |
| ------ | ------ | --------- |
| x1     | 3.00   | 12.21     |
```

```
| x2     | -1.96 | 5.14      |

MSE (per-output): [0.217, 0.145]
MSE overall: 0.181

# Output:

DECISION
```
{{
    "keep": ["x1", "x2", "x1**2", "sin"],
    "drop": ["cos"]
}}
```

That's it - perform the keep/drop decision based on the information provided.

========================================================
DATASET AND PROBLEM DESCRIPTION
========================================================
{dataset_and_problem_description}

========================================================
CURRENT TERMS
========================================================
{current_terms}

# Current equation:
{current_equation}

========================================================
HISTORY
========================================================
{history}

========================================================
INPUT YOU RECEIVE
========================================================

INPUT TABLE(S):

{input}

MSE (per-output): [..., ...]
MSE overall: ...
```

The prompt template for the "Prune" agent is populated with the following key pieces of information to guide the pruning decision:

**keep_n_terms**: Hyperparameter specifying the maximum number of terms to keep. Default value: 6.

**dataset_and_problem_description**: Analogous to the "Propose" agent.

**current_terms**: Analogous to the "Propose" agent.

**current_equation**: Analogous to the "Propose" agent.

**history**: Analogous to the "Propose" agent.

**input**: This is a critical input, formatted as a table, showing each candidate basis function $\psi_j$, its fitted weight $w_j$, and its calculated influence score $\Delta_j$. For example:

```
INPUT TABLE(S):

dv_dt:
| term                                | weight      | influence    |
|:------------------------------------|------------:|-------------:|
| cancer_volume                       | 0.0092651   | 2.7752       |
| chemo_concentration                 | -0.0035792  | 0.000354906  |
| chemo_dosage                        | 0.00120966  | 9.91133e-06  |
| radiotherapy_dosage                 | 0.00212704  | -7.52432e-07 |
| cancer_volume * chemo_concentration | -0.027962   | 638.689      |
| cancer_volume * chemo_dosage        | -1.14907e-05| 9.16085e-05  |
| cancer_volume * radiotherapy_dosage | -0.0477543  | 197.164      |
| chemo_dosage * radiotherapy_dosage  | -0.000750627| 1.83055e-05  |
| np.log(cancer_volume + 1)           | -1.20511    | 17.425       |
| np.sqrt(cancer_volume)              | 1.19886     | 109.978      |

dc_dt:
| term                                | weight      | influence    |
|:------------------------------------|------------:|-------------:|
| cancer_volume                       | 4.8914e-16  | -5.36642e-27 |
| chemo_concentration                 | -0.5        | 6.62719      |
```

```
| chemo_dosage                          | 1           | 10.8058      |
| radiotherapy_dosage                   | -6.55278e-15 | 4.93846e-28 |
| cancer_volume * chemo_concentration   | 1.86065e-17 | -1.98136e-27 |
| cancer_volume * chemo_dosage          | -8.391e-18  | 7.54011e-28  |
| cancer_volume * radiotherapy_dosage   | -9.34055e-17 | 3.47305e-27 |
| chemo_dosage * radiotherapy_dosage    | -1.90453e-16 | 3.39238e-29 |
| np.log(cancer_volume + 1)             | 5.50181e-16 | -2.14874e-28 |
| np.sqrt(cancer_volume)                | -2.09539e-16 | 2.65579e-28 |
```

```
MSE (per-output): [0.010905755578988435, 6.33977252002678e-27]
MSE overall: 0.005453
```

**Example LLM Response for "Prune" Agent:**    An example of an LLM's response when acting as the "Prune" agent is shown below.

*Listing 4.* Example LLM Response from "Prune" Agent

```
To make the equation interpretable and preserve only the most meaningful terms, I will review the terms in both outputs ('dv_dt' and '
    dc_dt') based on their influence and weight. I'll aim to retain at most 6 terms.

---

### **Analysis for 'dv_dt':**
1. **Highly influential terms**:
    - 'cancer_volume * chemo_concentration' (delta = 638.509) is critical.
    - 'cancer_volume * radiotherapy_dosage' (delta = 197.209) is highly impactful.
    - 'np.sqrt(cancer_volume)' (delta = 110.587) has substantial influence.
    - 'np.log(1 + cancer_volume)' (delta = 17.6803).

2. **Moderately influential / less relevant**:
    - 'cancer_volume' (delta = 2.7081): core variable, but delta suggests moderate importance.
    - 'chemo_dosage', 'radiotherapy_dosage': limited delta values.

3. **Negligible**:
    - Terms involving weighted interactions: e.g., 'chemo_dosage / (1 + cancer_volume)' (delta ~= 0). These can be ignored safely.

---

### **Analysis for 'dc_dt':**
1. **Highly influential terms**:
    - 'np.log(1 + cancer_volume)' (delta = 22.3107): critical.
    - 'chemo_dosage' (delta = 18.8115): essential for modeling 'dc_dt'.
    - 'np.sqrt(cancer_volume)' (delta = 15.8203): quite significant.

2. **Moderately influential**:
    - 'radiotherapy_dosage': important as part of the model.
    - 'cancer_volume': relevant, but has relatively small delta compared to other terms.

3. **Negligible terms**:
    - Interactions like 'radiotherapy_dosage / (1 + cancer_volume)' (small delta).

---

Considering both outputs, I aim to balance interpretability and prioritization based on influence across both equations. Here is the
    final decision:

---

DECISION
```
{
    "keep": [
    "cancer_volume * chemo_concentration",  # Very high delta for dv_dt
    "cancer_volume * radiotherapy_dosage",  # Very high delta for dv_dt
    "np.sqrt(cancer_volume)",               # Significant delta for both dv_dt and dc_dt
    "np.log(1 + cancer_volume)",            # Critical delta for both dv_dt and dc_dt
    "chemo_dosage",                         # Crucial delta for dc_dt
    "radiotherapy_dosage"                   # Balanced importance for dc_dt
    ],
    "drop": [
    "cancer_volume",                        # Moderate delta in both outputs
    "chemo_dosage * chemo_concentration",   # Insignificant in dv_dt (<0.0003)
    "radiotherapy_dosage * chemo_concentration",  # Similar to above
    "chemo_dosage / (1 + cancer_volume)",   # Negligible delta
    "radiotherapy_dosage / (1 + cancer_volume)"   # Negligible delta
    ]
}
```
```

# C. Benchmark Dataset and Evaluation Details

Our evaluation utilizes several benchmark datasets, including three derived from a sophisticated biomedical Pharmacokinetic-Pharmacodynamic (PKPD) model, one from an epidemic simulator, one real-world clinical dataset, and one real-world biological dataset.

## C.1. Cancer PKPD Simulations

Three of our benchmark environments are based on a well-established biomedical Pharmacokinetic-Pharmacodynamic (PKPD) model of lung cancer tumor growth. This model simulates the combined effects of chemotherapy and radiotherapy and has been utilized in prior research (Geng et al., 2017; Seedat et al., 2022; Bica et al., 2020; Melnychuk et al., 2022; Holt et al., 2024). We use this model to generate data for three distinct scenarios: **Lung Cancer** (no treatments), **Lung Cancer (with Chemo.)** (chemotherapy only), and **Lung Cancer (with Chemo. & Radio.)** (both chemotherapy and radiotherapy). Each scenario results in a separately sampled dataset. The comprehensive *Lung Cancer (with Chemo. & Radio.)* scenario, based on the general Cancer PKPD model, is described below, followed by how the other variations are derived.

**General Cancer PKPD Model Structure.** This model describes the tumor volume $x(t)$ (in cm$^3$) over time $t$ (in days) following diagnosis. It incorporates the effects of radiotherapy, $u_t^r$, and chemotherapy, $u_t^c$. The tumor dynamics are governed by the differential equation:

$$\frac{dx(t)}{dt} = \left( \rho \log \left( \frac{K}{x(t)} \right) - \beta_c C(t) - (\alpha_r d(t) + \beta_r d(t)^2) \right) x(t)$$

where the first term $\rho \log \left( \frac{K}{x(t)} \right)$ represents tumor growth dynamics, the term $\beta_c C(t)$ models the effect of chemotherapy, and $(\alpha_r d(t) + \beta_r d(t)^2)$ models the effect of radiotherapy. The patient-specific parameters $K, \rho, \beta_c, \alpha_r, \beta_r$ are adopted from (Geng et al., 2017), and their values are summarized in Table 6.

*Table 6.* **Cancer PKPD model parameter values from (Geng et al., 2017).**

| Model Component | Variable | Parameter Symbol | Value |
|---|---|---|---|
| Tumor Growth | Growth rate | $\rho$ | $7.00 \times 10^{-5}$ day$^{-1}$ |
| | Carrying capacity | $K$ | 30 cm$^3$ |
| Radiotherapy | Linear cell kill | $\alpha_r$ | 0.0398 Gy$^{-1}$ |
| | Quadratic cell kill | $\beta_r$ | Set such that $\alpha_r/\beta_r = 10$ Gy |
| Chemotherapy | Chemotherapy cell kill | $\beta_c$ | 0.028 (mg/m$^3$)$^{-1}$ day$^{-1}$ |

The concentration of the chemotherapy drug, $C(t)$, is modeled by an exponential decay with a one-day half-life:

$$\frac{dC(t)}{dt} = -0.5C(t)$$

A chemotherapy action $u_t^c$ corresponds to an increase in $C(t)$ by $5.0$ mg/m$^3$ of Vinblastine administered at time $t$. The radiotherapy term $d(t)$ represents the daily dose, with $u_t^r$ corresponding to $2.0$ Gy fractions of radiotherapy delivered at timestep $t$.

**Time-Dependent Confounding.** To introduce realistic complexities, the administration of chemotherapy and radiotherapy is modeled as Bernoulli random variables, with probabilities $p_c(t)$ and $p_r(t)$ dependent on the average tumor diameter $\bar{D}(t)$:

$$p_c(t) = \sigma \left( \frac{\gamma_c}{D_{\max}} (\bar{D}(t) - \delta_c) \right) \qquad\qquad p_r(t) = \sigma \left( \frac{\gamma_r}{D_{\max}} (\bar{D}(t) - \delta_r) \right)$$

where $\sigma(\cdot)$ is the sigmoid function, $D_{\max} = 13$ cm is the maximum tumor diameter, $\delta_c = \delta_r = D_{\max}/2$. The parameters $\gamma_c = \gamma_r = 2$ control the strength of this time-varying confounding.

**Dataset Generation.** For each of the three Cancer PKPD scenarios, we sample $N = 1,000$ patient trajectories. Initial tumor volumes $x(0)$ are drawn from a uniform distribution $\mathcal{U}(0, 1149)$ cm$^3$. Patient trajectories are simulated for 60 days using the PKPD model (Appendix C.1) and the specified action policy, employing a Euler stepwise numerical solver. This

procedure generates one dataset sample. We create distinct training, validation, and test sets ($\mathcal{D}_{\text{train}}, \mathcal{D}_{\text{val}}, \mathcal{D}_{\text{test}}$) by repeating this sampling process with different random seeds. For each run of a benchmark method requiring a random seed, these datasets are re-sampled. The **Lung Cancer** scenario omits both treatment terms (effectively $C(t) = 0, d(t) = 0$). The **Lung Cancer (with Chemo.)** scenario omits the radiotherapy term only ($d(t) = 0$). The **Lung Cancer (with Chemo. & Radio.)** scenario uses the full model as described.

### C.2. COVID-19 Epidemic Simulation

We employ COVASIM, a detailed agent-based simulator for modeling COVID-19 epidemics (Kerr et al., 2021). COVASIM can simulate various non-pharmaceutical interventions (e.g., lockdowns, social distancing) and pharmaceutical interventions (e.g., vaccinations). In this model, each agent signifies an individual who can transition between states: susceptible, exposed, infectious, or recovered (which includes deaths).

We use COVASIM with its default parameters as provided in its open-source implementation[3]. To generate diverse epidemic trajectories, we simulate 24 distinct "countries" or populations. For each simulation, the population size is set to $1,000,000$ individuals, with each agent simulated individually (i.e., simulation rescaling is disabled). Each simulation starts with an initial number of infected individuals $I(0)$ sampled uniformly from $\mathcal{U}(10,000, 100,000)$, and the epidemic is simulated for 60 days.

This process is repeated with independent random seeds to generate training, validation, and test datasets ($\mathcal{D}_{\text{train}}, \mathcal{D}_{\text{val}}, \mathcal{D}_{\text{test}}$). For each benchmark method run that involves a random seed, these datasets are re-sampled.

### C.3. Warfarin Pharmacokinetics Dataset

We utilize a real-world clinical trial dataset focused on Warfarin pharmacokinetics (PK), which is publicly available (Janssen et al., 2022). This dataset, known as the NOMEN dataset, can be accessed from https://github.com/Janssena/SI-AIEP-paper. It comprises data from 32 patients who received a single dose of Warfarin. The dataset contains a total of 251 Warfarin concentration measurements, with a median of six measurements per patient. Warfarin was administered at $t = 0$, and concentration measurements were taken at predefined time points: $t \in \{0.25, 0.5, 1.0, 2.0, 4.0, 6.0, 12.0, 24.0, 48.0, 72.0, 96.0, 120.0\}$ hours. Covariates available for each patient include weight, age, and sex.

We adhere to the original pre-processing scripts provided with the dataset. The data is split into training, validation, and test sets using proportions of 70%, 15%, and 15%, respectively. These splits are performed chronologically to maintain temporal causality. This dataset is released under a GPL-3.0 license.

### C.4. RNA Polymerase II Pausing Dataset

This dataset, central to the case study presented in Section 5.1, focuses on predicting RNA Polymerase II (Pol II) pausing. It is derived from eNET-seq data that maps Pol II pause sites in human cells at single-nucleotide resolution (Fong et al., 2022). The primary objective is to predict the `pause_score`, a continuous variable between 0 and 1, where a higher score indicates increased Pol II pausing at a specific genomic location. The pause score is calculated as the ratio of sequencing reads at the pause site to the total reads in a 200bp window surrounding it.

The dataset comprises 263 features, including:

- **Genomic Coordinates and Context:** Features such as `start` (current position), `gene_start`, `gene_end`, `exon_intron_start`, `exon_intron_end`, distance to nearest downstream (`down_nuc_dist`) and upstream (`up_nuc_dist`) nucleosomes.
- **Chromatin Features:**
    - Nucleosome occupancy signal from MNase-seq (e.g., `SIGNAL_MNase_CONDITION_WT`).
    - Histone modification signals from ChIP-seq, such as H3K4me3 (e.g., `SIGNAL_ChIPseqH3K4me3_CONDITION_CDK7negWT`, where higher signal implies more H3K4me3 modification) and H3K36me3 under various conditions (e.g., `SIGNAL_ChIPseqH3K36me3_CONDITION_negU170K`, `SIGNAL_ChIPseqH3K36me3_CONDITION_posU170K`, where higher signal indicates more H3K36me3 modification).

---

[3]COVASIM is available at https://github.com/InstituteforDiseaseModeling/covasim.

- **RNA Structure Features:** DMS signal (e.g., `SIGNAL_DMS_CONDITION_WT_STRAND_neg`, where lower signal implies more RNA structure) and RNA structure scores (e.g., `SIGNAL_StructureScore_CONDITION_WT_STRAND_neg`, where higher score indicates more structure).
- **DNA Sequence Context:** One-hot encoded nucleotides at positions -20 to +20 relative to the potential pause site (e.g., `seq_neg1_A`, `seq_0_G`, `seq_1_T`). Possible nucleotide categories are `A`, `C`, `G`, `T`, and `N` (unknown).
- **Gene Region Annotations:** Categorical features like `chrom_<category>` (chromosome, e.g., `chr1`, `chrX`), `strand_<category>` (`pos` or `neg`), `gene_region_<category>` (one of `TSS`, `body`, `termination`), and `exon_intron_<category>` (one of `exon`, `intron`, `missing`). Categorical columns are one-hot encoded, with column names in the format `<column_name>_<category>`.

The target variable is `pause_score`. The dataset used for the experiments described in Section 5.1 was balanced, containing approximately 48,000 pause sites and a similar number of control sites (where pause score = 0), and was split into training, validation, and test sets. For each benchmark method run involving a random seed, this training/validation/test split was resampled.

## C.5. Evaluation Details

We assess the performance of all benchmark methods using the mean squared error (MSE) on a held-out test dataset, denoted as $\mathcal{D}_{\text{test}}$. The MSE is calculated based on the model's predictions against the true target values in this test set. Where there are multiple target variables, the mean MSE across all targets is used.

For each method, given per-seed test MSEs $x_1, \ldots, x_n$, we report two-sided 95% confidence intervals for the mean across seeds as $\bar{x} \pm t_{0.975,n-1} \frac{s}{\sqrt{n}}$, where $\bar{x}$ is the sample mean, $s$ is the unbiased sample standard deviation, and $t_{0.975,n-1}$ is the Student's $t$ critical value (see, e.g., Wasserman, 2004). Replicates are independent training runs (random seeds). We use the appropriate degrees of freedom for each method's $n$.

When a simulator is used to generate data (e.g., for the Cancer PKPD and COVID-19 benchmarks as described in Appendix C), new training, validation, and test datasets ($\mathcal{D}_{\text{train}}, \mathcal{D}_{\text{val}}, \mathcal{D}_{\text{test}}$) are independently generated for each random seed. Unless specified otherwise, the validation and test datasets are generated to contain a comparable number of samples or trajectories as the training set.

Each benchmark model is trained using its respective training dataset ($\mathcal{D}_{\text{train}}$). For methods that support it, early stopping (at patience 10) or default model selection heuristics are applied using the validation dataset ($\mathcal{D}_{\text{val}}$) to prevent overfitting and guide the learning process. For IGSR, the validation set is critically used for calculating per-term influence scores and guiding the pruning decisions, as detailed in Section 2. Maximum number of iterations or generations is set to 30 for all methods where this is applicable unless stated otherwise; IGSR maximum node expansion budget for MCTS is similarly set set to 30. For a complementary experiment comparing methods at a fixed computational (LLM token) budget, see Appendix H.3. The final reported performance for all methods is evaluated on the unseen test dataset ($\mathcal{D}_{\text{test}}$). This entire procedure of data generation (if applicable), training, validation-guided refinement (if applicable), and testing is repeated for each random seed to ensure robust and reliable comparisons. We use 25 random seeds for IGSR results and other LLM-based method results, using the same underlying LLM. Given the evaluation is exactly the same as the baseline and datasets for some results of the baselines, we use previous values averaged over 10 random seeds for the non-LLM baselines from paper (Holt et al., 2024).

# D. Benchmark Method Details

Our comparative evaluation includes several established benchmark methods to contextualize the performance of IGSR. These methods span both black-box and white-box modeling paradigms. Hyperparameters and other configuration details are detailed further in Appendix I.

## D.1. White-box non-LLM baselines

**SINDy** (Sparse Identification of Nonlinear Dynamics) is a white-box method designed to discover governing differential equations directly from time-series data (Brunton et al., 2016). It operates by constructing a library of candidate nonlinear functions of the state variables and then employs sparse regression techniques (typically sequentially thresholded least-squares or Lasso) to identify a minimal set of active terms that best describe the observed dynamics. The result is an interpretable, parsimonious differential equation model.

**GPLearn** (Stephens, 2015) is a genetic programming approach for symbolic regression that algorithmically discovers mathematical expressions to model a given dataset. It works by evolving a population of candidate equations over a number of generations, applying genetic operators such as crossover and mutation to iteratively refine solutions based on a fitness metric, typically related to prediction accuracy. In the experiments, unless stated otherwise, GPLearn was configured with a `population_size` of 1000, run for 30 `generations`, and a `parsimony_coefficient` of 1.0. The `parsimony_coefficient` of 1.0 was selected to apply a considerable penalty to the complexity of the evolved expressions. This encourages the discovery of more concise equations, which aligns with the general aim of finding parsimonious models (e.g., around six terms in the primary IGSR experiments), thus making shorter programs preferable during the evolutionary search.

**PySR** (Cranmer, 2023) is a high-performance symbolic regression toolkit that leverages a multi-population evolutionary algorithm (regularized evolution) to discover interpretable mathematical expressions. It identifies a Pareto frontier of equations that balance predictive accuracy with complexity. In our experiments, we utilized PySR version 0.19 with a population size and iteration count as described in Appendix I, employing the "best" model selection strategy to determine the final equation.

## D.2. White-box LLM baselines

To ensure fair comparison, we set the maximum number of terms allowed to be six (unless otherwise stated) in IGSR, and ensured this was comparably set in all other corresponding methods. We use the same underlying LLM across all LLM-based methods (`GPT-4o` version `2024-11-20`, unless otherwise stated).

**ZeroShot** represents a baseline LLM-driven approach where the Large Language Model is prompted to generate a symbolic equation model in a single pass, based solely on the provided problem description and dataset characteristics. This method does not involve any iterative refinement or feedback based on the model's performance on actual data. Consequently, the parameters of the equation generated by the ZeroShot method are used as proposed by the LLM without any subsequent optimization against the training dataset. The number of terms in the equation generated by the LLM was constrained via prompting, typically to a maximum of six terms (unless stated otherwise), to align with the parsimony goals of IGSR.

**ZeroOptim** builds directly upon the ZeroShot approach. An LLM first generates an initial symbolic equation model in a zero-shot manner, based on the problem description. However, unlike the pure ZeroShot baseline, the structural form of this LLM-proposed equation is then taken, and its constituent parameters (weights $w_j$) are subsequently optimized by fitting them to the training data. This optimization process is performed analogously to the final model fitting stage within the IGSR framework, aiming to find the best parameter values for the LLM's proposed equation structure. As with ZeroShot, the LLM was prompted to generate equations with a limited number of terms, typically up to six (unless stated otherwise), for comparability.

**ICL** is an LLM-based iterative equation discovery method that serves as a key ablation and comparator to IGSR. In this approach, the LLM iteratively proposes candidate equations (or modifications to existing ones). After each proposal, the equation's parameters are optimized analogously to IGSR, and its performance is evaluated, typically yielding a scalar metric such as the overall Mean Squared Error (MSE) on a validation set. This scalar MSE, along with a history of previously attempted equations and their outcomes, is now provided back to the LLM to guide its proposal for the next iteration, facilitating in-context learning. Crucially, ICL lacks the granular, per-term influence scores ($\Delta_j$) that are central to IGSR.

Furthermore, it does not incorporate IGSR's dedicated term pruning phase, where an LLM agent refines the equation structure based on these influence scores. For fair comparison with IGSR, the maximum number of terms in the equations generated by the LLM was limited via prompting, with a default of six terms in the experiments unless specified otherwise.

**D3-white-box** (Holt et al., 2024) refers to the Data-Driven Discovery framework when specifically configured to discover interpretable, white-box dynamical system models. This approach leverages Large Language Models (LLMs) within an iterative cycle involving three core agents: Modeling, Feature Acquisition, and Evaluation. For white-box discovery, the Modeling Agent is tasked by the LLM to propose and refine closed-form equation models. These equations are represented as executable Python code, typically PyTorch modules. After a model is proposed, its parameters are optimized against training data. The Evaluation Agent then assesses the model, providing feedback that includes quantitative metrics like validation Mean Squared Error (MSE) and qualitative verbal reflections on model structure and plausibility. This feedback informs the LLM for subsequent iterations of model generation and refinement. The Feature Acquisition Agent can also propose additional relevant features to incorporate, further guiding the discovery process, although for specific benchmark comparisons, it might be applied to a fixed set of predefined features. The overall D3 framework aims to autonomously navigate the model space, identify relevant system variables, and converge on accurate, interpretable equations. The implementation details and hyperparameters as per (Holt et al., 2024) were used for this benchmark. The maximum number of input features in the white-box model was constrained via prompting, with a default of six, unless specified otherwise.

**LLM-SR** (Scientific Equation Discovery via Programming with Large Language Models, Shojaee et al., 2025a) frames equation discovery as program synthesis. It leverages an LLM's scientific prior knowledge and code generation capabilities to propose equation skeletons as Python programs. These programs are then combined with an evolutionary search, where data-driven feedback on the overall program's fit is used to guide the iterative refinement process. The complexity hyperparameter `max_nparams` was set to 8, in excess of IGSR's six term limit. The rest of the hyperparameters were used as per (Shojaee et al., 2025a).

**LaSR** (Symbolic Regression with a Learned Concept Library, Grayeli et al., 2024) is a framework that enhances traditional genetic algorithms for symbolic regression by incorporating a learned library of abstract textual concepts. It uses an LLM to discover and evolve high-level concepts (e.g., "exponential decay") from successful equations. These concepts then guide the mutation and crossover operations in the search, biasing it towards scientifically plausible structures rather than refining the equations directly. The complexity hyperparameter `maxsize` of 30 was used to allow for equations at least as complex as the six-term limit used elsewhere. The total number of cycles was set to 75. Other hyperparameters used were as per (Grayeli et al., 2024).

**ICSR** (In-Context Symbolic Regression, Merler et al., 2024) employs an LLM as an optimizer within an iterative refinement loop. The LLM is prompted with a set of previously evaluated equations and their corresponding scalar fitness scores (which typically combine accuracy and a complexity penalty). Through in-context learning, the LLM is tasked with generating a new candidate equation that is expected to achieve a better score, effectively navigating the solution space based on a history of successes and failures. The hyperparameter from (Merler et al., 2024) were used, and `max_nodes` complexity parameter was set to 30 to allow for equations at least as complex as the six-term limit used in IGSR and elsewhere.

### D.3. Black-box baselines

**DyNODE** (Dynamical Neural Ordinary Differential Equations) is a method that learns the underlying dynamics of a system from observed data by parameterizing the derivative function of an ordinary differential equation (ODE) with a neural network (Chen et al., 2018). For systems involving external actions or interventions, the DyNODE framework can be extended to incorporate these action inputs directly into the learned dynamics, allowing it to model how treatments or other external factors influence the system's evolution over time (Alvarez et al., 2020). This approach offers a flexible black-box model for continuous-time dynamical systems.

**RNN** (Recurrent Neural Network) models are a class of neural networks well-suited for sequential data, including time-series (Rumelhart et al., 1986). Standard RNN architectures, such as those using LSTM (Long Short-Term Memory) (Hochreiter & Schmidhuber, 1997) or GRU (Gated Recurrent Unit) (Cho et al., 2014) cells, maintain an internal hidden state that captures information from past inputs, enabling them to model temporal dependencies. For prediction tasks, the RNN processes input sequences (e.g., trajectories of state variables and actions) to predict future states or system outputs. These are generally considered black-box models due to the complexity of their internal representations.

**Transformer** models, originally introduced for natural language processing tasks (Vaswani et al., 2017), have demonstrated

strong performance on a wide variety of sequential data, including time-series. The core mechanism of Transformers is the attention mechanism, particularly self-attention, which allows the model to weigh the importance of different elements in the input sequence when making predictions. This enables them to capture long-range dependencies effectively. Like RNNs, Transformer models are typically considered black-box due to their intricate architectures and large number of parameters.

**XGBoost** (Chen & Guestrin, 2016) is a scalable implementation of gradient-boosted decision trees (GBDTs) and serves as a robust black-box baseline. It constructs an ensemble of weak learners (decision trees) in a sequential manner to minimize a differentiable loss function, effectively capturing complex non-linear interactions in the data without producing an interpretable symbolic form. We employed the standard XGBoost regressor using the library's default hyperparameters.

# E. LLM-SRBench Evaluation: Further Detail and Results

**Benchmark Overview:** Existing symbolic regression benchmarks often rely on well-known physical equations (e.g., Feynman benchmark) that are likely present in the pre-training corpora of modern Large Language Models. This allows models to potentially solve problems via memorization rather than true scientific reasoning and data-driven discovery. LLM-SRBench (Shojaee et al., 2025b) addresses this by introducing a suite of 239 challenging problems designed to prevent trivial memorization. It includes transformed representations of physical models and synthetic, discovery-driven problems. The latter problems span four distinct scientific domains: Physics, Biology, Material Science, and Chemistry. Evaluating on LLM-SRBench ensures that IGSR's performance reflects genuine data-driven discovery rather than the retrieval of memorized equations.

**Experimental Setup:** In order to evaluate the the methods on scientific discovery tasks, where additional scientific context on the problem and its variables is available, we performed the evaluation on all problems in the four Scientific Domains, as available in the dataset: Physics (problem label "PO", 43 problems), Biology (problem label "BPG", 24 problems), Material Science (problem label "MatSci", 25 problems), and Chemistry (problem label "CRK", 36 problems), a large and diverse set of 128 total problems.

To ensure a strictly comparable computational footprint across all evaluated methods, as well as a manageable computational cost budget, we enforced a fixed budget of approximately 300,000 tokens per run, with iteration counts set in excess to ensure this token limit served as the active termination criterion. Each problem was evaluated over 5 random seeds using gpt-4o-mini (version 2024-07-18) as the underlying LLM, while all other method-specific hyperparameters were maintained consistent with the rest of the experiments in this work, see Appendix D.

**Evaluation Metrics:** To provide a comprehensive view of model performance, we report metrics covering predictive accuracy, out-of-distribution generalization, and symbolic structural correctness.

**Predictive Performance Metrics:** We first report standard numerical accuracy metrics:

- *Normalized Mean Squared Error (NMSE)*: We compute this metric as defined in (Shojaee et al., 2025b). It is evaluated for **both** in-distribution (ID) and out-of-distribution (OOD) test sets.

- *Accuracy to Tolerance 0.1 ($Acc_{0.1}$)*: We compute this metric as defined in (Shojaee et al., 2025b), representing the fraction of runs achieving an NMSE below 0.1. This is also evaluated for **both** ID and OOD test sets.

**Symbolic Metrics:** We report the following symbolic recovery metrics:

- *Symbolic Accuracy*: We report *LLM-assisted Symbolic Accuracy* as defined in (Shojaee et al., 2025b). A high capacity reasoning model, gpt-5 (version 2025-08-07) is used as a judge, to ensure better accuracy on the complex task of matching the predicted equation representation with the ground truth. We also include the following additional line in the judge prompt (compared to Shojaee et al. 2025b default): "Note: The ground truth expression is always correct, and you should not consider altering its constants/parameters". This was added to avoid the common misinterpretation by the LLM judge (which we observed empirically by inspecting the logs), of modifying the ground truth equation constants (to, e.g. 0, making certain ground truth terms disappear), in order to match the hypothesis – rather than the other way round. We observe fairly low values of Symbolic Accuracy across the methods, in single digit percentages.

- *Term Recall*: To evaluate the recovery of underlying dynamics independent of parameter precision, we define Term Recall. Let $E_{gt}$ and $E_{pred}$ be the ground truth and predicted expressions, decomposed into additive term sets. We define a skeletonization mapping, $\phi(\tau)$, which reduces a term $\tau$ to its canonical form by removing top-level scalar coefficients and replacing internal floating-point parameters with unity, while preserving integer exponents that define topology (e.g., $\phi(0.42\sin(1.01t)) \rightarrow \sin(t)$). Denoting the set of skeletonized terms as $S(E) = \{\phi(\tau) \mid \tau \in \text{terms}(E)\}$, the metric is defined as the normalized intersection $\text{TR} = |S(E_{gt}) \cap S(E_{pred})|/|S(E_{gt})|$. This strictly penalizes missing physical terms while remaining robust to numerical approximation. To enable these structural comparisons across methods with differing output formats – which range from raw symbolic strings to executable Python functions or class definitions we implemented a standardization step, where we utilized a high capacity LLM (gpt-5, version 2025-08-07) to translate each proposed solution into a canonical, SymPy-parsable format, ensuring a consistent representation for subsequent metric computation.

## E.1. Detailed Results and Per-Domain Breakdown

Table 2b reports the method average rank and Table 7 the metric values, aggregated across all the problems in the four Scientific Domains.

*Table 7.* **Performance and Symbolic Metrics on LLM-SRBench.**
We report the Normalized Mean Squared Error (NMSE) and Accuracy within 10% tolerance ($\text{Acc}_{0.1}$) for both In-Distribution (ID) and Out-Of-Distribution (OOD) test sets. We also report Term Recall and Symbolic Accuracy to evaluate the recovered equations. As NMSR varies greatly across problems and is affected by outliers, we report this as Median [IQR]. Overall results aggregated across the four Scientific Domains are shown.
**Formatting:** For each column, the **best** result is bolded and the second best is underlined.
**Values:** Metrics are reported as Median [IQR] for NMSE, and Mean $\pm$ Standard Deviation for others.

| Method | ID Test Set | | OOD Test Set | | Symbolic Metrics | |
|---|---|---|---|---|---|---|
| | NMSE $\downarrow$ | $\text{Acc}_{0.1}$ $\uparrow$ | NMSE $\downarrow$ | $\text{Acc}_{0.1}$ $\uparrow$ | Term Recall $\uparrow$ | Symbolic Accuracy $\uparrow$ |
| ICL | $6.19 \times 10^{-4}$ [$1.04 \times 10^{-2}$] | $0.391 \pm 0.422$ | $0.883$ [338] | $0.383 \pm 0.437$ | $0.167 \pm 0.165$ | 0.00 |
| D3-white-box | $0.400$ [$0.624$] | $0.0233 \pm 0.136$ | $29.2$ [$2.21 \times 10^3$] | $0.0496 \pm 0.182$ | $0.195 \pm 0.203$ | $0.00155 \pm 0.0176$ |
| LLM-SR | $4.54 \times 10^{-4}$ [$5.90 \times 10^{-3}$] | $0.391 \pm 0.427$ | $0.325$ [54.9] | $0.422 \pm 0.436$ | $0.205 \pm 0.165$ | $0.0140 \pm 0.0622$ |
| LaSR | $9.47 \times 10^{-3}$ [$0.0300$] | $0.0775 \pm 0.216$ | $1.50$ [61.6] | $0.251 \pm 0.379$ | $0.0653 \pm 0.123$ | $0.00775 \pm 0.0388$ |
| ICSR | $1.15 \times 10^{-3}$ [$2.91 \times 10^{-3}$] | $0.188 \pm 0.311$ | $0.870$ [24.4] | $0.327 \pm 0.420$ | $0.131 \pm 0.153$ | $0.00465 \pm 0.0393$ |
| IGSR-Agent | $1.69 \times 10^{-5}$ [$3.57 \times 10^{-4}$] | $0.640$ $\pm 0.436$ | $1.15 \times 10^{-3}$ [1.54] | $0.626$ $\pm 0.440$ | $0.321$ $\pm 0.271$ | **$0.0558$** $\pm 0.175$ |
| **IGSR** | **$7.51 \times 10^{-7}$** [$2.40 \times 10^{-5}$] | **$0.781$** $\pm 0.368$ | **$2.90 \times 10^{-5}$** [$0.127$] | **$0.744$** $\pm 0.394$ | **$0.344$** $\pm 0.297$ | **$0.0558$** $\pm 0.180$ |

Overall, on both the performance and symbolic recovery metrics, IGSR and its agentic pruning variant IGSR-Agent outperform the baseline methods across both In-Distribution (ID) and Out-Of-Distribution (OOD) evaluations. As shown in Table 2b, the IGSR variants secure the top two average ranks for both NMSE and $\text{Acc}_{0.1}$. Crucially, this performance advantage is maintained on the OOD test sets. Similarly, the IGSR variants occupy two average ranks for Term Recall and Symbolic Accuracy. In terms of the actual values of the metrics (Table 7), we observe a similar pattern, with IGSR variants outperforming the other methods on average. The non-LLM pruning variant (IGSR) outperforms the deterministic pruning IGSR-Agent variant on the performance metrics, and the two variants perform comparably on the symbolic metrics.

Here, we present the results broken down by the four LLM-SRBench Scientific Domains. Table 8 presents the method average rank and Table 9 contains the metric values.

We find that IGSR consistently outperforms the baselines on the predictive performance metrics in all four domains, and on the symbolic metrics in all but one domain – Material Science – where we find D3-white-box and LLM-SR to be somewhat better (though note the overall low symbolic recovery results across all methods in this domain, e.g. Symbolic Accuracy mostly 0.0). Qualitative inspection of the discovered equations in this domain suggests that D3 and LLM-SR more successfully retain the intact functional forms of domain-specific terms (e.g., shifted dependencies like $\beta(T - T_0)$ or Arrhenius-type rates), while IGSR tends to approximate these dynamics through linear expansions or alternative basis combinations. Consequently, this fragmentation of the ground-truth structure results in a lower symbolic recovery, even when predictive accuracy is maintained. We anticipate that our *IGSR-TLO* variant (detailed in Appendix G.14), which is designed to identify and optimize such internal constants, would bridge this structural gap; however, it was not deployed in this benchmark evaluation.

## E.2. Structural Fidelity vs Predictive Accuracy

To synthesize the relationship between symbolic reconstruction and predictive generalization, we visualize the trade-off between structural fidelity (Term Recall) and out-of-distribution accuracy (OOD NMSE) in Figure 5. In this analysis, the ideal model resides in the top-right quadrant, characterized by high term recall (recovering the correct physical terms) and low prediction error.

We qualitatively observe that *IGSR* tends to populate the top-right quadrant more so than the baselines, especially for the Physics, Biology and Chemistry domains. This indicates that IGSR effectively balances the discovery of mechanistic terms with numerical precision. A notable exception is observed in the Material Science domain, where IGSR results tend to populate the bottom-right quadrant, a point which aligns with the discussion in Appendix E.1.

*Table 8.* **Average Rank by LLM-SRBench Scientific Domain.** We report the average rank (lower is better) for ID (In-Distribution) and OOD (Out-Of-Distribution) predictive performance, as well as symbolic recovery metrics. The best performing method in each column is **bolded**, and the second best is underlined.

*(a)* Physics (PO)

| Method | ID Test Set | | OOD Test Set | | Symbolic Metrics | |
|---|---|---|---|---|---|---|
| | NMSE | $Acc_{0.1}$ | NMSE | $Acc_{0.1}$ | Term Recall | Symb. Acc. |
| ICL | 4.73 | 2.75 | 5.34 | 2.84 | 4.25 | 1.39 |
| D3-white-box | 6.91 | 3.41 | 5.98 | 3.59 | 4.52 | 1.39 |
| LLM-SR | 3.70 | 2.36 | 4.43 | 2.64 | 3.71 | 1.34 |
| LaSR | 5.75 | 3.02 | 4.41 | 3.11 | 5.05 | 1.25 |
| ICSR | 3.55 | 2.70 | 4.07 | 2.68 | 4.16 | 1.39 |
| IGSR-Agent | 2.30 | 1.64 | 2.36 | 1.64 | 1.91 | **1.14** |
| **IGSR** | **1.07** | **1.07** | **1.41** | **1.11** | **1.20** | 1.25 |

*(b)* Biology (BPG)

| Method | ID Test Set | | OOD Test Set | | Symbolic Metrics | |
|---|---|---|---|---|---|---|
| | NMSE | $Acc_{0.1}$ | NMSE | $Acc_{0.1}$ | Term Recall | Symb. Acc. |
| ICL | 3.67 | 2.67 | 5.08 | 2.67 | 3.38 | 1.50 |
| D3-white-box | 6.83 | 3.38 | 5.92 | 3.38 | 2.04 | 1.50 |
| LLM-SR | 5.13 | 2.79 | 4.96 | 2.46 | 2.46 | 1.46 |
| LaSR | 4.83 | 3.21 | 4.13 | 2.50 | 4.38 | 1.50 |
| ICSR | 3.58 | 3.17 | 3.42 | 2.71 | 3.75 | 1.50 |
| IGSR-Agent | 2.46 | 1.38 | 2.42 | **1.21** | **1.88** | 1.21 |
| **IGSR** | **1.50** | **1.21** | **2.08** | 1.25 | 2.13 | **1.04** |

*(c)* Material Science (MatSci)

| Method | ID Test Set | | OOD Test Set | | Symbolic Metrics | |
|---|---|---|---|---|---|---|
| | NMSE | $Acc_{0.1}$ | NMSE | $Acc_{0.1}$ | Term Recall | Symb. Acc. |
| ICL | 3.48 | 2.40 | 3.52 | 1.16 | 2.80 | 1.12 |
| D3-white-box | 6.80 | 5.16 | 6.60 | 6.40 | **2.20** | 1.12 |
| LLM-SR | 2.40 | 1.16 | 2.64 | 1.36 | 2.61 | **1.00** |
| LaSR | 5.60 | 5.40 | 5.68 | 3.64 | 3.32 | 1.12 |
| ICSR | 5.08 | 4.16 | 4.92 | 1.64 | 2.24 | 1.12 |
| IGSR-Agent | 2.64 | 1.28 | **2.16** | **1.08** | 2.72 | 1.12 |
| **IGSR** | **2.00** | **1.08** | 2.48 | 1.32 | 3.28 | 1.12 |

*(d)* Chemistry (CRK)

| Method | ID Test Set | | OOD Test Set | | Symbolic Metrics | |
|---|---|---|---|---|---|---|
| | NMSE | $Acc_{0.1}$ | NMSE | $Acc_{0.1}$ | Term Recall | Symb. Acc. |
| ICL | 3.36 | 2.28 | 4.14 | 2.58 | 3.14 | 1.36 |
| D3-white-box | 6.97 | 4.89 | 6.64 | 4.47 | 2.89 | 1.33 |
| LLM-SR | 3.67 | 2.97 | 3.50 | 2.53 | 3.27 | 1.31 |
| LaSR | 5.67 | 4.53 | 5.00 | 3.92 | 5.08 | 1.36 |
| ICSR | 4.83 | 3.67 | 4.44 | 3.50 | 4.11 | 1.31 |
| IGSR-Agent | 2.36 | 1.56 | 2.78 | 1.78 | **1.64** | **1.17** |
| **IGSR** | **1.14** | **1.14** | **1.50** | **1.22** | 1.97 | 1.19 |

*Table 9.* **Performance and Symbolic Metrics by LLM-SRBench Scientific Domain.**
We report the Normalized Mean Squared Error (NMSE) and Accuracy within 10% tolerance ($\text{Acc}_{0.1}$) for both In-Distribution (ID) and Out-Of-Distribution (OOD) test sets. We also report Term Recall and Symbolic Accuracy to evaluate the recovered equations. As NMSR varies greatly across problems and is affected by outliers, we report this as Median [IQR].
**Formatting:** For each column, the **best** result is bolded and the second best is underlined.
**Values:** Metrics are reported as Median [IQR] for NMSE, and Mean $\pm$ Standard Deviation for others.

*(a)* Physics (PO)

| Method | ID Test Set | | OOD Test Set | | Symbolic Metrics | |
|---|---|---|---|---|---|---|
| | NMSE $\downarrow$ | $\text{Acc}_{0.1}$ $\uparrow$ | NMSE $\downarrow$ | $\text{Acc}_{0.1}$ $\uparrow$ | Term Recall $\uparrow$ | Symbolic Accuracy $\uparrow$ |
| ICL | $6.30 \times 10^{-3}$ [0.0227] | $0.127 \pm 0.283$ | 0.116 [320] | $0.136 \pm 0.272$ | $0.211 \pm 0.158$ | 0.00 |
| D3-white-box | 0.160 [0.588] | 0.00 | 0.413 [1.86] | 0.00 | $0.180 \pm 0.169$ | 0.00 |
| LLM-SR | $1.63 \times 10^{-3}$ [0.0115] | $0.209 \pm 0.353$ | 0.0554 [54.7] | $0.195 \pm 0.349$ | $0.255 \pm 0.155$ | $4.55 \times 10^{-3} \pm 0.0302$ |
| LaSR | 0.0106 [0.0339] | $0.0591 \pm 0.186$ | 0.0507 [1.94] | $0.0864 \pm 0.222$ | $0.133 \pm 0.160$ | $0.0227 \pm 0.0642$ |
| ICSR | $1.95 \times 10^{-3}$ [$5.12 \times 10^{-3}$] | $0.145 \pm 0.284$ | 0.0184 [207] | $0.182 \pm 0.313$ | $0.215 \pm 0.154$ | 0.00 |
| IGSR-Agent | $2.62 \times 10^{-4}$ [$1.36 \times 10^{-3}$] | $0.414 \pm 0.434$ | $2.68 \times 10^{-3}$ [7.27] | $0.414 \pm 0.423$ | $0.440 \pm 0.241$ | **$0.0500 \pm 0.130$** |
| **IGSR** | **$1.70 \times 10^{-5}$** [$2.17 \times 10^{-4}$] | **$0.664 \pm 0.399$** | **$1.31 \times 10^{-4}$** [$4.25 \times 10^{-3}$] | **$0.668 \pm 0.393$** | **$0.543 \pm 0.263$** | $0.0273 \pm 0.0924$ |

*(b)* Biology (BPG)

| Method | ID Test Set | | OOD Test Set | | Symbolic Metrics | |
|---|---|---|---|---|---|---|
| | NMSE $\downarrow$ | $\text{Acc}_{0.1}$ $\uparrow$ | NMSE $\downarrow$ | $\text{Acc}_{0.1}$ $\uparrow$ | Term Recall $\uparrow$ | Symbolic Accuracy $\uparrow$ |
| ICL | $1.09 \times 10^{-3}$ [$5.82 \times 10^{-3}$] | $0.183 \pm 0.363$ | 163 [1680] | $0.158 \pm 0.339$ | $0.0861 \pm 0.143$ | 0.00 |
| D3-white-box | 0.625 [0.659] | $0.0333 \pm 0.127$ | 899 [20800] | $0.0750 \pm 0.203$ | $0.232 \pm 0.186$ | 0.00 |
| LLM-SR | 0.288 [0.604] | $0.142 \pm 0.267$ | 139 [710] | $0.192 \pm 0.341$ | $0.189 \pm 0.200$ | $0.0167 \pm 0.0817$ |
| LaSR | 0.0166 [0.0480] | $0.0333 \pm 0.113$ | 6.83 [135] | $0.167 \pm 0.352$ | 0.00 | 0.00 |
| ICSR | $1.23 \times 10^{-3}$ [$3.18 \times 10^{-3}$] | $0.0583 \pm 0.161$ | 4.57 [36.8] | $0.133 \pm 0.299$ | $0.0486 \pm 0.0958$ | 0.00 |
| IGSR-Agent | $2.15 \times 10^{-7}$ [$1.90 \times 10^{-5}$] | $0.658 \pm 0.431$ | $1.64 \times 10^{-3}$ [1.30] | $0.608 \pm 0.481$ | **$0.243 \pm 0.282$** | $0.108 \pm 0.250$ |
| **IGSR** | **$7.81 \times 10^{-8}$** [$6.00 \times 10^{-6}$] | **$0.717 \pm 0.421$** | **$4.00 \times 10^{-5}$** [5.96] | **$0.642 \pm 0.457$** | $0.232 \pm 0.277$ | **$0.167 \pm 0.316$** |

*(c)* Material Science (MatSci)

| Method | ID Test Set | | OOD Test Set | | Symbolic Metrics | |
|---|---|---|---|---|---|---|
| | NMSE $\downarrow$ | $\text{Acc}_{0.1}$ $\uparrow$ | NMSE $\downarrow$ | $\text{Acc}_{0.1}$ $\uparrow$ | Term Recall $\uparrow$ | Symbolic Accuracy $\uparrow$ |
| ICL | $3.21 \times 10^{-6}$ [$2.03 \times 10^{-4}$] | $0.768 \pm 0.287$ | $1.30 \times 10^{-3}$ [0.0575] | $0.968 \pm 0.125$ | $0.120 \pm 0.106$ | 0.00 |
| D3-white-box | 0.163 [0.216] | $0.0880 \pm 0.277$ | 17.5 [16.8] | $0.128 \pm 0.282$ | **$0.173 \pm 0.252$** | 0.00 |
| LLM-SR | $5.89 \times 10^{-8}$ [$1.80 \times 10^{-5}$] | $0.912 \pm 0.239$ | $9.40 \times 10^{-5}$ [$6.24 \times 10^{-3}$] | $0.928 \pm 0.251$ | $0.157 \pm 0.129$ | **$0.0240 \pm 0.0663$** |
| LaSR | $4.40 \times 10^{-3}$ [0.0107] | $0.104 \pm 0.272$ | 0.496 [1.18] | $0.720 \pm 0.316$ | $0.0900 \pm 0.122$ | 0.00 |
| ICSR | $7.19 \times 10^{-4}$ [$1.67 \times 10^{-3}$] | $0.304 \pm 0.347$ | 0.153 [0.374] | $0.936 \pm 0.170$ | $0.141 \pm 0.170$ | 0.00 |
| IGSR-Agent | $9.10 \times 10^{-8}$ [$3.70 \times 10^{-5}$] | $0.880 \pm 0.332$ | $8.50 \times 10^{-6}$ [$3.09 \times 10^{-3}$] | **$0.968 \pm 0.160$** | $0.103 \pm 0.141$ | 0.00 |
| **IGSR** | **$1.54 \times 10^{-8}$** [$8.00 \times 10^{-6}$] | **$0.928 \pm 0.230$** | **$7.23 \times 10^{-7}$** [$1.75 \times 10^{-3}$] | $0.952 \pm 0.202$ | $0.0807 \pm 0.107$ | 0.00 |

*(d)* Chemistry (CRK)

| Method | ID Test Set | | OOD Test Set | | Symbolic Metrics | |
|---|---|---|---|---|---|---|
| | NMSE $\downarrow$ | $\text{Acc}_{0.1}$ $\uparrow$ | NMSE $\downarrow$ | $\text{Acc}_{0.1}$ $\uparrow$ | Term Recall $\uparrow$ | Symbolic Accuracy $\uparrow$ |
| ICL | $8.11 \times 10^{-5}$ [$4.39 \times 10^{-4}$] | $0.589 \pm 0.388$ | 4.45 [2180] | $0.428 \pm 0.403$ | $0.200 \pm 0.195$ | 0.00 |
| D3-white-box | 0.623 [0.441] | 0.00 | 2580 [73600] | $0.0389 \pm 0.178$ | $0.203 \pm 0.219$ | $5.56 \times 10^{-3} \pm 0.0333$ |
| LLM-SR | $9.44 \times 10^{-5}$ [$8.81 \times 10^{-4}$] | $0.417 \pm 0.387$ | 0.960 [22.4] | $0.500 \pm 0.369$ | $0.189 \pm 0.163$ | $0.0167 \pm 0.0737$ |
| LaSR | $6.81 \times 10^{-3}$ [0.0275] | $0.111 \pm 0.255$ | 32.8 [130] | $0.183 \pm 0.339$ | $8.33 \times 10^{-3} \pm 0.0368$ | 0.00 |
| ICSR | $9.08 \times 10^{-4}$ [$1.69 \times 10^{-3}$] | $0.244 \pm 0.358$ | 15.2 [52.8] | $0.211 \pm 0.338$ | $0.0759 \pm 0.117$ | $0.0167 \pm 0.0737$ |
| IGSR-Agent | $5.96 \times 10^{-6}$ [$1.28 \times 10^{-4}$] | $0.739 \pm 0.388$ | $6.63 \times 10^{-2}$ [10.5] | $0.661 \pm 0.422$ | **$0.380 \pm 0.270$** | **$0.0667 \pm 0.214$** |
| **IGSR** | **$1.06 \times 10^{-7}$** [$9.00 \times 10^{-6}$] | **$0.867 \pm 0.321$** | **$4.84 \times 10^{-4}$** [1.05] | **$0.761 \pm 0.408$** | $0.358 \pm 0.267$ | $0.0556 \pm 0.176$ |

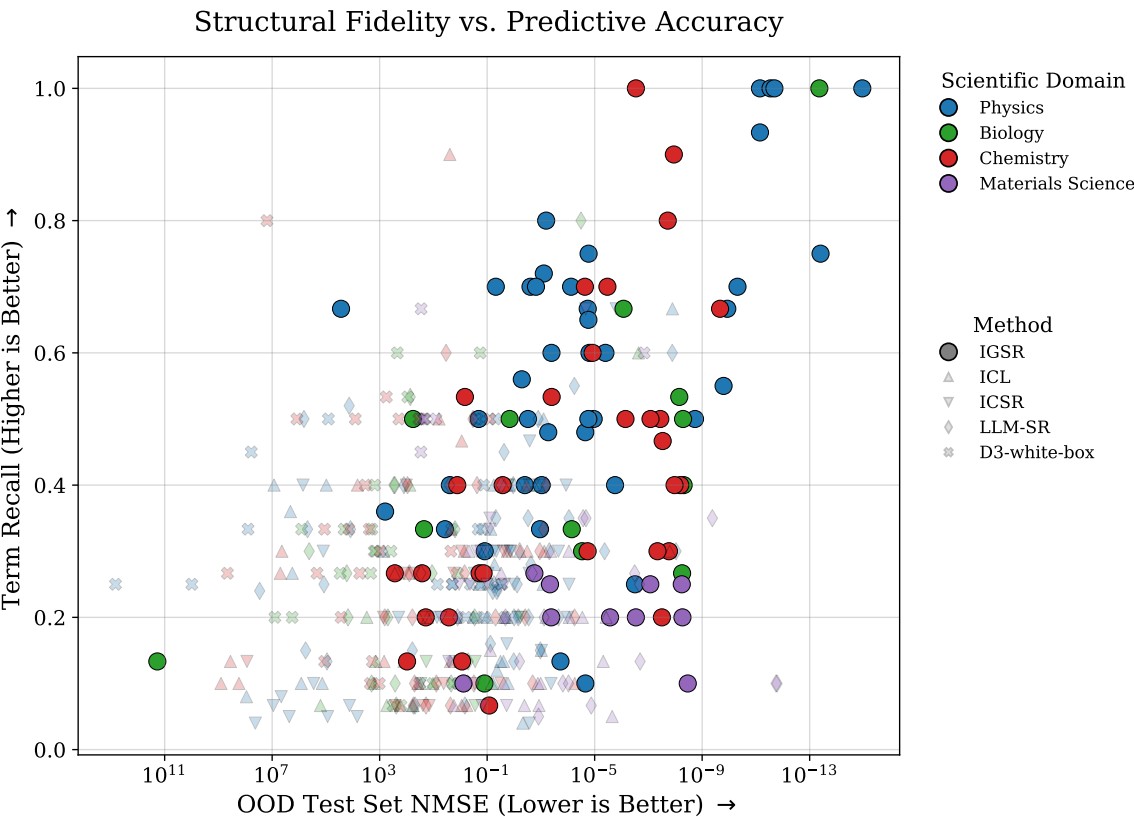

*Figure 5.* **Visualization of Structural Fidelity vs. Predictive Accuracy on LLM-SRBench.** We plot the Out-of-Distribution (OOD) NMSE (log scale, lower is better, axis reversed such that lower values are to the right) against Term Recall (higher is better) for all method-problem pairs. Markers distinguish the discovery methods, while the colors distinguish the Scientific Domains (Physics, Biology, Material Science, Chemistry). IGSR tends to occupy the ideal top-right quadrant more than the baselines, except for the Materials Science domain, where it tends to underperform on Term Recall.

# F. RNA Polymerase II Pausing Case Study: Further Detail, Discussion, and Results

In this section, we provide SHAP plots for the case study experiment (referred to as **Experiment 1** here), an additional experiment (referred to as **Experiment 2** here) conducted to further investigate the use of IGSR with the RNA Polymerase II pausing problem, and a discussion given both sets of results. The aim is to illustrate how IGSR can be used in scientific discovery and hypothesis generation process (e.g. note the extension to Experiment 2 from Experiment 1, given the results of the former). **Experiment 3** follows Experiment 2's problem setup, but uses additional biological signal features – resulting in a novel hypothesis regarding DNA methylation impact on pausing, subsequently confirmed in a wet-lab experiment, Appendix F.3. This case study is one of many potential use cases for IGSR, illustrating the potential power of this method.

**Configuration Note:** The case study Experiments 1-3 were performed with default IGSR hyperparameters except: `keep_n_terms=15` and LLM backbone of `o3` (version `2025-04-16`).

## F.1. Experiment 1

**Results.** IGSR, guided by influence-based feedback, discovered the following interpretable equation:

$$
\begin{aligned}
\text{pause\_score} = {} & 0.0178\,\ln\!\big(1 + \text{signal}_{\text{MNase}}\big) \\
& - 0.000246\,\text{signal}_{\text{H3K4me3}} + 0.00902\,\ln\!\big(1 + \text{signal}_{\text{H3K4me3}}\big) \\
& + 0.0194\,\ln\!\Big(1 + \sum_{\text{ds}} \text{signal}_{\text{H3K36me3,ds}}\Big) \\
& - 0.0291\,\big(\mathbb{1}_{\{\text{seq}_{-1}=A\}} + \mathbb{1}_{\{\text{seq}_0=A\}} + \mathbb{1}_{\{\text{seq}_1=A\}} \\
& \qquad + \mathbb{1}_{\{\text{seq}_{-1}=T\}} + \mathbb{1}_{\{\text{seq}_0=T\}} + \mathbb{1}_{\{\text{seq}_1=T\}}\big) \\
& + 0.0223\,\mathbb{1}_{\{\text{gene\_region}=\text{TSS}\}} + 0.0257\,\mathbb{1}_{\{\text{gene\_region}=\text{body}\}} + 0.0636\,\mathbb{1}_{\{\text{gene\_region}=\text{termination}\}} \\
& - 0.0146\,\big(\mathbb{1}_{\{\text{seq}_{-3}=T\}} + \mathbb{1}_{\{\text{seq}_{-2}=T\}} + \mathbb{1}_{\{\text{seq}_{-1}=T\}}\big) \\
& - 0.0402\,\big(\mathbb{1}_{\{\text{seq}_{-1}=G\}} + \mathbb{1}_{\{\text{seq}_{-1}=C\}}\big) + 0.0333\,\mathbb{1}_{\{\text{seq}_0=G\}} - 0.0397\,\mathbb{1}_{\{\text{seq}_0=C\}} \\
& + 0.0735\,\mathbb{1}_{\{\text{seq}_1=T\}} + 0.0243\,\mathbb{1}_{\{\text{seq}_{-1}=G\}},
\end{aligned}
$$

where "ds" sums over the values of H3K36me3 signals from several alternative data sources; $\text{seq}_i$ is the nucleotide at relative position $i$; $\mathbb{1}_{\{\cdot\}}$ is the indicator function.

To further interrogate the interpretable equation discovered by IGSR for RNA Polymerase II pausing (presented in Section 5.1), $f(\mathbf{x}) = \sum_j w_j \psi_j(\mathbf{x})$, we employed SHapley Additive exPlanations (SHAP) (Lundberg & Lee, 2017). SHAP is a game theoretic approach that explains the output of a model by assigning an importance value (SHAP value) to each of its input features for every individual prediction. In this application, the "features" provided to SHAP are the evaluated basis functions $\psi_j(\mathbf{x})$ that form the terms of the IGSR equation. The SHAP values therefore quantify how much each term $w_j \psi_j(\mathbf{x})$ contributes to pushing the model's output (the pause score) from its base (average) prediction to the actual predicted value for a given sample. This analysis allows us to visualize the magnitude, variability, and directional impact of each constituent term in the discovered equation.

While the IGSR equation $f(\mathbf{x}) = \sum_j w_j \psi_j(\mathbf{x})$ is already structured for interpretability as a weighted sum of basis functions, SHAP analysis provides a standardized framework to:

- Visualize the distribution of contributions for each basis function term $\psi_j(\mathbf{x})$ across all samples.
- Understand how the specific value of a basis function $\psi_j(\mathbf{x})$ (which itself is derived from the original input features like $\text{signal}_{\text{MNase}}$ or sequence indicators) influences its contribution to the final pause score.
- Confirm the relative importance and consistent impact of terms that were selected and weighted during the IGSR discovery process.

Essentially, SHAP helps to decompose the prediction into contributions from each $\psi_j(\mathbf{x})$ term, providing insights at both a global (overall term importance) and local (individual prediction explanation) level.

We generated two types of SHAP plots to visualize these explanations, as shown in Figure 6: a beeswarm plot and a custom term influence bar plot.

The **beeswarm plot** (Figure 6a) summarizes the SHAP values for the most influential basis function terms $\psi_j(\mathbf{x})$ in the equation. Each point on the plot represents a single term's SHAP value for a specific sample (a potential pause site). The terms are ranked along the y-axis by their global importance (sum of absolute SHAP values across all samples). The x-axis

shows the SHAP value, indicating the term's impact on the model output; positive SHAP values contribute to a higher predicted pause score, while negative values contribute to a lower score. The color of each point corresponds to the *value of the basis function* $\psi_j(\mathbf{x})$ for that sample (typically, red indicates high values of $\psi_j(\mathbf{x})$ and blue indicates low values). This coloring reveals how the magnitude of each basis function's evaluation influences its contribution to the pause score.

The **custom term influence bar plot** (Figure 6b) provides an alternative view of global term importance and the general direction of each term's influence. The length of each bar corresponds to the mean absolute SHAP value for that basis function term $\psi_j(\mathbf{x})$, signifying its overall importance in the equation. The color and direction of the bar indicate the predominant direction of the term's influence on the pause score. This directionality is determined by calculating the Spearman's rank correlation coefficient ($\rho$) between the *values of the basis function* $\psi_j(\mathbf{x})$ for each sample and their corresponding SHAP values for that term. Spearman's correlation is employed here because:

- It assesses monotonic relationships. The SHAP value of a term $w_j\psi_j(\mathbf{x})$ will have a monotonic relationship with $\psi_j(\mathbf{x})$ (the direction determined by the sign of $w_j$). Spearman captures this robustly.
- It is robust to outliers in the evaluated basis function values or the SHAP values.
- It operates on the ranks of the data, suitable even if distributions are non-normal.

A positive Spearman correlation (typically shown in red, extending to the right) suggests that as the value of the basis function $\psi_j(\mathbf{x})$ increases, its SHAP value (and thus its effective contribution $w_j\psi_j(\mathbf{x})$ relative to its mean) tends to push the predicted pause score higher. Conversely, a negative correlation (typically shown in blue, extending to the left) suggests that an increasing value of $\psi_j(\mathbf{x})$ tends to push the pause score lower. This plot helps to quickly identify which terms in the IGSR-discovered equation are most impactful and whether higher values of these terms generally promote or inhibit Pol II pausing.

### F.2. Experiment 2 (a Follow-up)

**Experimental Setup.** Experiment 1 (discussed in Section 5.1) tasked IGSR with predicting pause sites from an artificially balanced dataset comprising equal numbers of actual pause sites and control non-pause sites within genes. While this approach identified key features distinguishing pause sites from non-pause regions, a limitation is that pause sites are naturally much less frequent than non-pause sites. Furthermore, this setup primarily addresses the *presence* of a pause rather than the *strength* or characteristics of different pause sites relative to each other.

To delve deeper into the factors modulating the intensity of Pol II pausing, Experiment 2 adopted a different approach. Here, IGSR was tasked with identifying terms that distinguish pause sites based on their varying pause scores, using a dataset consisting exclusively of identified pause sites (i.e., non-pause sites were excluded). This focuses the analysis on understanding what makes some pauses stronger or weaker than others, given that a pause event is already occurring. This also addresses any potential confounding effects of artificial balancing between pause and non-pause sites in Experiment 1.

**Results.** The equation discovered by IGSR in this context is presented below.

$$
\begin{aligned}
\text{pause\_score} = & -0.01711 \ln(1 + \text{signal}_{\text{MNase}}) \\
& -0.0003407 \, \text{signal}_{\text{H3K4me3}} \\
& -0.00216 \ln(2 + \text{signal}_{\text{H3K4me3}}) \\
& -0.0005086 \sum_{\text{cond}} \text{signal}_{\text{H3K36me3, cond}} \\
& +0.01206 \ln(1 + \text{gene\_length}) \\
& -0.08414 \, \mathbb{1}_{\{\text{gene\_region=TSS}\}} \\
& +0.0003421 \, \mathbb{1}_{\{\text{gene\_region=TSS}\}} \cdot \text{signal}_{\text{H3K4me3}} \\
& +0.02143 \, \mathbb{1}_{\{\text{seq}_0=G\}} \\
& +0.0006817 \ln(1 + \text{down\_nuc\_dist}) \\
& +0.01968 \ln(1 + \text{down\_nuc\_dist} + |\text{up\_nuc\_dist}|) \\
& +0.0001679 \left( \text{signal}_{\text{H3K4me3}} - \sum_{\text{cond}} \text{signal}_{\text{H3K36me3, cond}} \right) \\
& -0.01338 \left( \ln(2 + \text{signal}_{\text{H3K4me3}}) - \ln\left(2 + \sum_{\text{cond}} \text{signal}_{\text{H3K36me3, cond}}\right) \right) \\
& +0.009823 \, \mathbb{1}_{\{\text{seq}_{-1}=G\}} \\
& +0.01422 \left( \mathbb{1}_{\{\text{seq}_0=G\}} \cdot \mathbb{1}_{\{\text{seq}_1=T\}} \right) \\
& -0.01761 \left( \mathbb{1}_{\{\text{seq}_{-1}=G\}} + \mathbb{1}_{\{\text{seq}_{-1}=C\}} + \mathbb{1}_{\{\text{seq}_0=G\}} + \mathbb{1}_{\{\text{seq}_0=C\}} \right.
\end{aligned}
$$

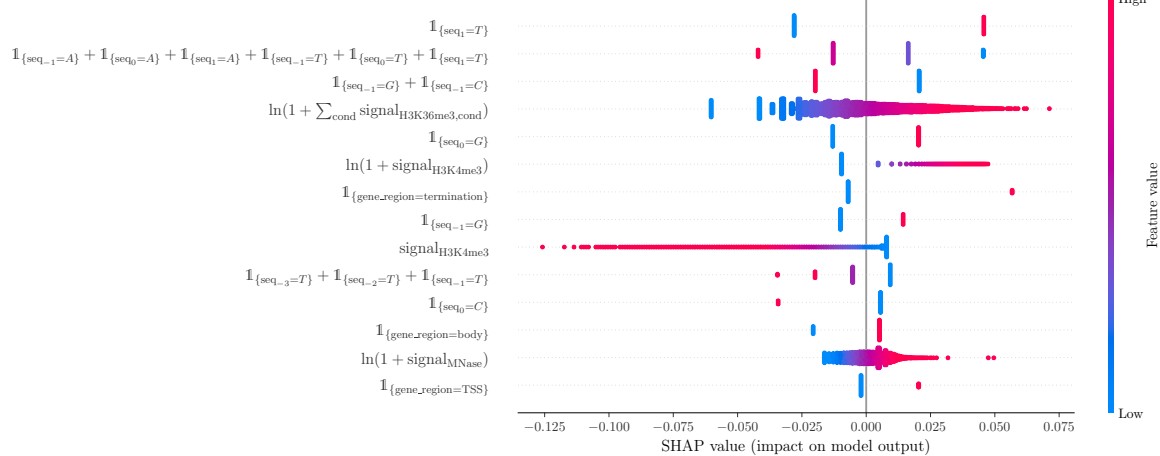

*(a)* SHAP Beeswarm Plot for Equation Terms

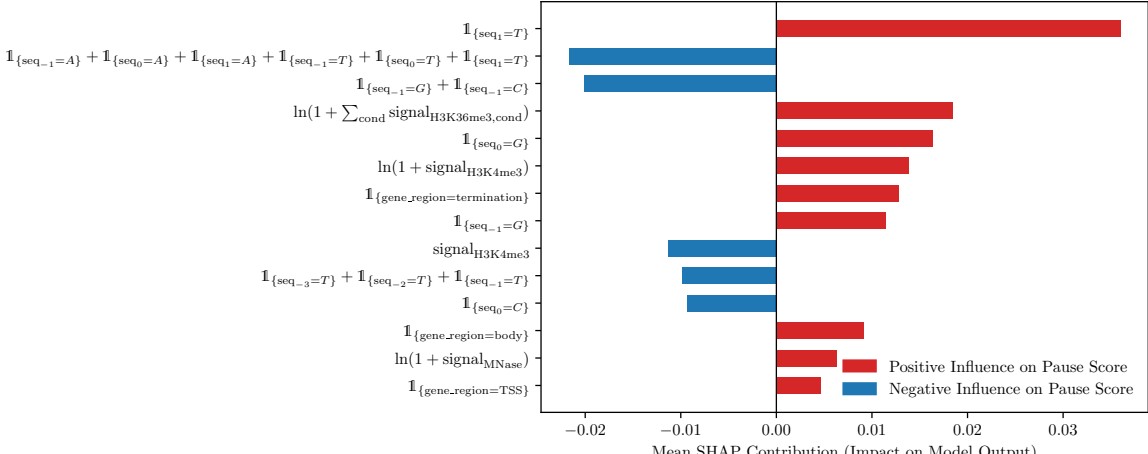

*(b)* SHAP Term Influence Bar Plot

*Figure 6.* SHAP analysis of the IGSR-discovered equation for RNA Polymerase II pausing from **Experiment 1**. Each "feature" in the SHAP analysis corresponds to a basis function term $\psi_j(\mathbf{x})$ from the equation. (**a**) Beeswarm plot showing the distribution of SHAP values for the most important equation terms. Each point is a SHAP value for a term and a sample, colored by the value of the evaluated basis function $\psi_j(\mathbf{x})$ (red for high, blue for low). (**b**) Custom bar plot illustrating global term importance (mean absolute SHAP value) and the direction of influence. Red bars indicate a general positive correlation between the value of $\psi_j(\mathbf{x})$ and its contribution to the pause score (Spearman's $\rho > 0$), while blue bars indicate a negative correlation (Spearman's $\rho < 0$).

$$+ \mathbb{1}_{\{\text{seq}_1=G\}} + \mathbb{1}_{\{\text{seq}_1=C\}}),$$

The SHAP analysis for this equation (Figure 7) illustrates the relative effects of these different terms on the predicted pause score.

**Discussion.** Combining insights from both Experiment 1 and Experiment 2, IGSR has revealed several features of potential functional importance to transcriptional pausing in human cells. A notable result emerging from this comparative analysis is that high H3K4me3 signal is generally associated with reduced pausing (i.e., lower pause scores in Experiment 2), consistent with a proposed a link between this mark and faster transcription (Wang et al., 2023). Regarding H3K36me3, Experiment 1 indicated its role as a significant predictor for the *presence* of pause sites. However, Experiment 2 suggests that higher levels of H3K36me3 are negatively related to the *strength* of these pause sites (lower pause scores), a finding that aligns with some genetic studies (Lee et al., 2025; Wen et al., 2014). Furthermore, IGSR's discovered equations highlight the importance of the relative levels of these two histone marks. Specifically, terms reflecting a high ratio of H3K36me3 to H3K4me3

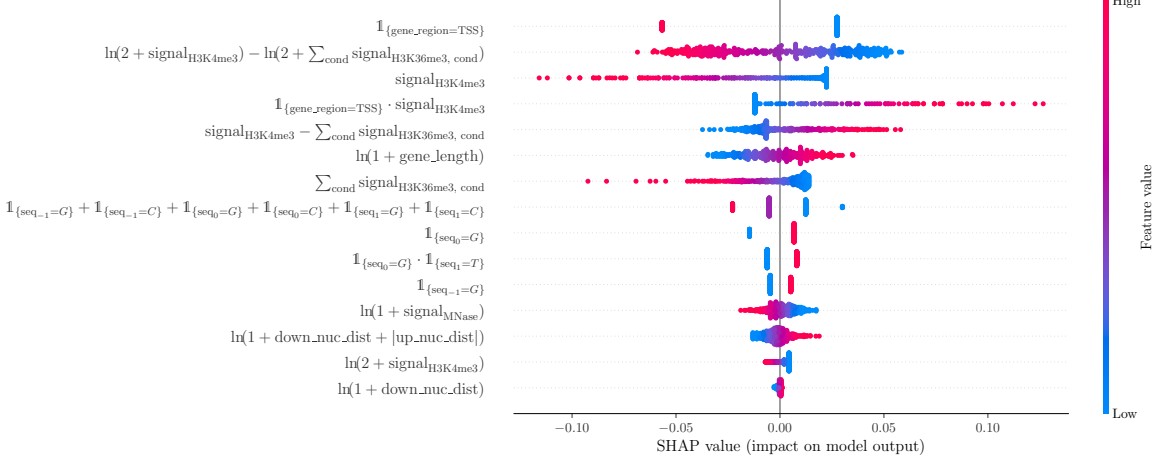

*(a)* SHAP Beeswarm Plot for Equation Terms

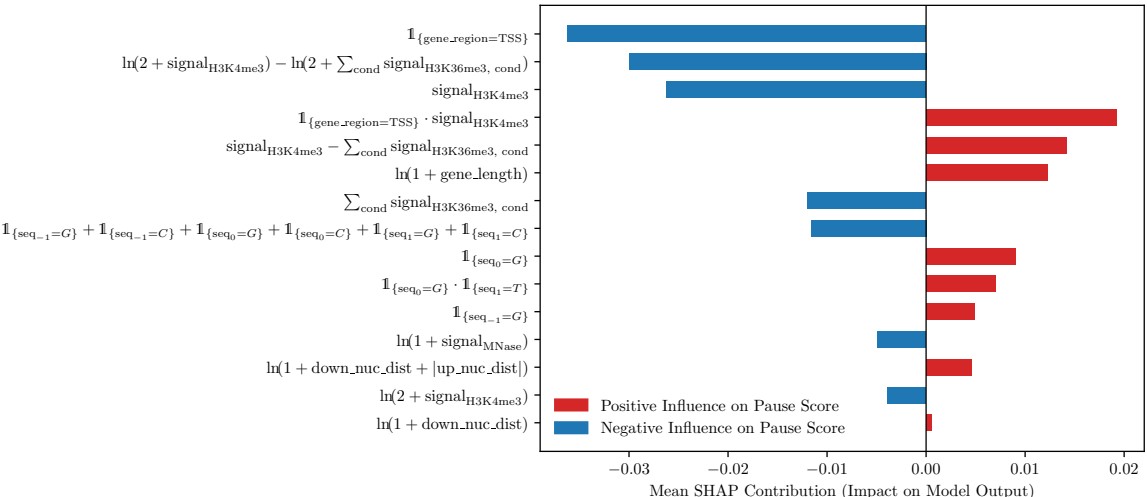

*(b)* SHAP Term Influence Bar Plot

*Figure 7.* SHAP analysis of the IGSR-discovered equation for RNA Polymerase II pausing from **Experiment 2**. Analogous to Figure 6, but for **Experiment 2**.

(e.g., through difference terms like $\text{signal}_{\text{H3K4me3}} - \sum \text{signal}_{\text{H3K36me3, cond}}$ with a positive coefficient, or ratio-like terms in logarithmic form) are associated with higher pause scores, whereas a high H3K4me3 to H3K36me3 ratio appears linked to lower pause scores.

The analysis also found that the region downstream of genes (termination region) is associated with higher pausing scores relative to the gene body and the 5' end (TSS). This is consistent with the hypothesis that Pol II pausing plays a role in facilitating transcription termination (Gromak et al., 2006).

This work also refined our understanding of sequence specificity in pausing. Previous studies identified a G, T/C sequence element as enriched at the pause site (position 0) and the +1 position (Fong et al., 2022; Gajos et al., 2021). Our IGSR models confirm that T at position +1 (often as part of a $G_0$–$T_{+1}$ motif) is a strong positive predictor for pause sites. Conversely, the models from both experiments, particularly when examining factors that disfavor pausing or reduce pause strength, identified new elements: C at position 0 and -1, and T at positions -1, -2, and -3 appear to disfavor pausing. Additionally, a cluster of G/C nucleotides at positions -1, 0, and +1 (captured by terms like $-(\mathbb{1}_{\{\text{seq}_{-1}=G\}} + \cdots + \mathbb{1}_{\{\text{seq}_1=C\}})$) was also found to be associated with reduced pausing or lower pause scores.

These results, derived from the interpretable equations generated by IGSR, provide new, data-driven hypotheses that will help guide future experimental studies into the complex mechanisms of sequence-dependent transcriptional pausing and its regulation by the chromatin environment.

### F.3. Experiment 3: Wet-Lab Validation of IGSR Hypothesis

To rigorously assess the capability of IGSR to generate novel, physically verifiable scientific knowledge – rather than merely fitting hold-out data – we conducted a testable study. This involved running a new experiment using IGSR, analyzing the resulting interpretable equation to formulate a biological hypothesis, and subsequently verifying that hypothesis through a targeted "wet-lab" experiment involving cell treatment and sequencing.

EXPERIMENTAL SETUP: EXPERIMENT 3

We applied IGSR to an extended version of the RNA Polymerase II pausing dataset. Like *Experiment 2*, this experiment (*Experiment 3*) focused on modeling the intensity of pausing (pause score) at pause sites, but unlike previous experiments, we included additional signal features, specifically `H3K27ac` (an acetylation mark) and `CpG_methyl`, a DNA CpG methylation mapping by Bisulphite-seq in HCT116 cells[4].

**The Discovered Equation:** IGSR converged on the following model. Notably, the system incorporated the new signals (`H3K27ac` and `CpG_methyl`) while discarding previous markers like H3K4me3, yet achieving a superior MSE of 0.0102 (compared to 0.0111 in prior experiments). The terms contributing to the novel DNA methylation hypothesis are highlighted in Violet.

$$
\begin{aligned}
\text{pause\_score} \; = \; & 0.1430 \quad \text{(constant baseline term)} \\
& + 0.0878 \, \text{signal}_{\text{CpG\_methyl}} - 0.2199 \, \ln\!\big(1 + \text{signal}_{\text{CpG\_methyl}}\big) \\
& + 0.0103 \, \ln\!\big(1 + \text{down\_nuc} - \text{up\_nuc}\big) \\
& - 0.0745 \, \ln\!\big(1 + \text{signal}_{\text{MNase}}\big) \\
& - 0.0923 \, \mathbb{1}_{\{\text{gene\_region=TSS}\}} + 0.0342 \, \mathbb{1}_{\{\text{gene\_region=TSS}\}} \cdot \ln\!\big(1 + \text{signal}_{\text{MNase}}\big) \\
& - 0.0732 \, \ln\!\big(1 + \text{signal}_{\text{H3K27ac}}\big) \\
& + 0.0172 \, \mathbb{1}_{\{\text{gene\_region=TSS}\}} \cdot \text{signal}_{\text{H3K27ac}} \\
& + 0.0474 \, \mathbb{1}_{\{\text{exon}\}} \cdot \ln\!\big(1 + \text{signal}_{\text{H3K27ac}}\big) \\
& + 0.0274 \, \mathbb{1}_{\{\text{seq}_0=G\}} - 0.0143 \sum_{pos \in \{-1,0,1\}} \big(\mathbb{1}_{\{\text{seq}_{pos}=G\}} + \mathbb{1}_{\{\text{seq}_{pos}=C\}}\big) \\
& + 0.0112 \, \ln\!\big(1 + \text{gene\_length}\big)
\end{aligned}
$$

FROM EQUATION TO HYPOTHESIS

Analysis of the IGSR-discovered equation pointed to a significant role for DNA methylation. Specifically, IGSR predicted that DNA CpG methylation terms (inferred via the `LiftOver` mappability signal and sequence contexts) accounted for approximately 50% of the explanatory power for pause site strength in this context.

Based on the signs of the coefficients, IGSR suggests the novel hypothesis: **DNA methylation suppresses pausing within gene bodies.**

While methylation of promoter sequences is well known to suppress transcription initiation (Deaton & Bird, 2011), far less is known about the function of DNA methylation further downstream within gene bodies. Conflicting reports suggest that DNA methylation of the gene body can either suppress (Lorincz et al., 2004) or enhance gene expression (Yang et al., 2014). How transcription elongation is specifically affected by DNA methylation remains understudied. Based on genome-wide data, IGSR suggests the novel hypothesis that *DNA methylation suppresses pausing*, a connection that has not previously received significant attention.

---

[4]Refer to dataset GSE158010: https://www.ncbi.nlm.nih.gov/geo/query/acc.cgi?acc=GSE158010.

WET-LAB EXPERIMENTAL VERIFICATION

**Methodology:** We tested this hypothesis by treating HCT116 human colorectal carcinoma cells with 5-azacytidine (1 mM), a potent DNA hypomethylating agent approved for the treatment of certain leukemias (Thoms et al., 2025). Cells were treated for 3 days followed by 2 days of recovery to deplete DNA methylation levels. We then examined transcriptional pausing using eNET-seq, including a DMSO-treated vehicle control, across two biological replicates (yielding ~3000 gene-body pause sites per condition).

**Results:** The experimental results revealed a **significant increase** in both the frequency and strength of RNA Polymerase II pauses throughout the length of transcribed genes in the treated (hypomethylated) cells compared to DMSO controls (Wilcoxon $p=4.3 \times 10^{-52}$ for pause frequency and $p=3.5 \times 10^{-33}$ for pause strength). This aligns perfectly with the IGSR prediction: if methylation suppresses pausing (as predicted), removing methylation should increase it.

> ### Validation Conclusion
>
> **Hypothesis Confirmed:**
>
> - **Prediction:** IGSR predicted that methylation suppresses pausing (negative correlation in the discovered equation).
>
> - **Observation:** Removing methylation (hypomethylation via 5-azacytidine) caused pausing to *increase*.
>
> - **Conclusion:** This supports the IGSR-derived hypothesis that CpG methylation in gene bodies functions as a suppressor of pausing, thereby enhancing gene expression at the level of transcription elongation.

This result suggests a *previously unsuspected function for CpG methylation in gene bodies*. In summary, IGSR provides a potent example of how an AI agent can collaborate with biologists to generate entirely original hypotheses that are experimentally verifiable.

ROBUSTNESS OF THE DISCOVERED METHYLATION HYPOTHESIS

Because the methylation hypothesis is the central scientific claim of this case study, we assessed how robustly IGSR recovers it under perturbations of the data and feature set. We deem a run (seed) to support the hypothesis if the combined effect of its discovered methylation terms correlates *negatively* with the pausing target. Table 10 reports recovery over 10 seeds under three conditions: the original setup, an *independent biological replicate*, and a *feature-variation* setting in which the inputs are randomly subsetted to 100 features. The hypothesis is recovered in 7/10 seeds on the original replicate, 6/10 on the independent replicate, and 8/10 under random feature subsetting. This consistent recovery indicates the finding is not an artifact of a single seed, replicate, or feature configuration; the moderate variance on the independent replicate is consistent with expected biological heterogeneity. This complements the equation-level generalization study in Appendix G.9, where equations selected only on Replicate 1 still generalize best to Replicate 2 without refitting.

*Table 10.* **Robustness of the DNA Methylation Hypothesis Discovery.** We evaluate the stability of IGSR's DNA methylation finding under two alternative conditions, over 10 seeds: *(1) Independent Replicate*, using a distinct biological replicate, and *(2) Feature Variation*, randomly subsetting the inputs to 100 features. A run (seed) supports the hypothesis if the combined effect of its discovered methylation terms correlates negatively with the pausing target. This consistent recovery validates IGSR's discovery capabilities, with moderate variance on the independent replicate reflecting expected biological heterogeneity.

| Experimental Setting | Condition Details | Seeds Supporting Hypothesis | Consistency |
|---|---|:---:|:---:|
| Original | Full feature set, original sequencing replicate | 7 / 10 | 70% |
| **Independent Replicate** | Full feature set, distinct biological replicate | 6 / 10 | 60% |
| **Feature Variation** | Features randomly subsetted to 100 | 8 / 10 | 80% |

# G. Additional Results

## G.1. Performance of Agentic Pruning Variant on Benchmark Datasets

We report the performance of the agentic pruning variant of our method, *IGSR-Agent*, on the six Benchmark Datasets, to supplement Table 2a. While the default *IGSR* employs a deterministic selection mechanism based on influence scores for computational efficiency and robustness, *IGSR-Agent* utilizes a secondary LLM agent to interpret these scores and make semantic pruning decisions (see Appendix B). As shown in Table 11, the agentic variant achieves competitive results, slightly outperforming the deterministic variant on two Lung Cancer datasets and the COVID-19 datasets, while performing comparably or slightly worse on others. Note that both variants outperform most of the non-black-box baselines from Table 2a.

We observe that compared to LLM-SRBench results (Table 2b and Appendix E), where the deterministic variant dominates more consistently, here the results are more mixed. We speculate that this may, in part, be due to the greater impact of in-context domain knowledge provided with these datasets – unlike LLM-SRBench, where only a brief description of the problem and its variables is available, the dataset descriptions for the six Benchmark Datasets all feature more detailed context on the domain and the dataset – this could therefore help by providing the "LLM Pruner" with context useful to its term selection decisions.

*Table 11.* **Comparison of IGSR Pruning Variants on Benchmark Datasets.** Test MSE (mean $\pm$ 95% CI) on held-out data. We compare the default deterministic pruning strategy (*IGSR*) against the agentic pruning strategy (*IGSR-Agent*). Experimental setup is identical to Table 2a. **Abbreviations:** LC: Lung Cancer, LC-C: Lung Cancer (with Chemo.), LC-CR: Lung Cancer (with Chemo. & Radio.), C-19: COVID-19, RNAPol: RNA Polymerase, Warf: Warfarin PK.

| Method | LC
MSE↓ | LC-C
MSE↓ | LC-CR
MSE↓ | C-19
MSE↓ | RNAPol
MSE↓ | Warf
MSE↓ |
|---|---|---|---|---|---|---|
| IGSR-Agent | **2.46e-6**$\pm$5.56e-6 | **0.0011**$\pm$0.0011 | 0.0146$\pm$0.0108 | **5.00e-8**$\pm$2.00e-9 | 0.0111$\pm$0.0005 | 0.581$\pm$0.103 |
| IGSR | 5.64e-5$\pm$6.79e-5 | 0.0013$\pm$0.0010 | **0.0141**$\pm$0.0087 | 5.01e-8$\pm$1.78e-9 | 0.0111 $\pm$0.0004 | **0.565**$\pm$0.113 |

## G.2. Extended Ablation of Influence Feedback, Search Strategy, and Pruning Approach

Here we extend the ablation shown in Table 3 to disentangle the "LLM Pruning" vs. "Deterministic Pruning" approaches. Since the deterministic term pruning by necessity uses the the per-term influence values, ablating influence feedback in this setting is not possible. The "LLM Pruning" (following the *IGSR-Agent* variant of our method, as described in Appendix B) *does* allow either per-term influence feedback or basic scalar (val. set MSE) feedback via prompting. Since the "LLM Pruning" approach also allows the ablation of the MCTS component, we include the two additional ablation variants here for completeness:

- LLM Pruning variant with Influence Feedback and with MCTS: "*Full IGSR, LLM Pruning (IGSR-Agent)*" row in Table 12,
- LLM Pruning variant with Influence Feedback and no MCTS: "*w/o MCTS (Iterative + Influence Feedback), LLM Pruning*" row in Table 12.

This allows for clear separation of the ablation components: "MCTS" (yes/no), "Influence Feedback" (yes/no), and "Pruning Approach" (Deterministic/LLM).

From the results in Table 3 we observe that the same pattern of ablating the MCTS leading to worse results (i.e. the importance of the efficient exploration strategy), as discussed in Section 5.2, holds regardless of whether pruning is implemented deterministically (*IGSR*) or via an LLM (*IGSR-Agent*).

**Experiment Configuration Notes.** In all LLM and deterministic ablation experiments, the `keep_n_terms` parameter was set to 6 for both approaches, for equivalent comparison. In Figure 2: for equivalent comparison: both variants start with a candidate pool of 6 terms and 6 terms are kept after each propose-and-prune iteration; at the root node performance *before* pruning is shown to ensure equivalent starting point; the best test MSE achieved so far as of # node expansion is tracked.

*Table 12.* **Extended Ablation Study.** We disentangle the effects of MCTS, Influence Feedback, and the term Pruning Approach (Deterministic vs. LLM). Results are test NMSE (lower is better) averaged over 25 seeds with 95% confidence intervals. (Not Valid) indicates an invalid combination: influence feedback cannot be ablated for deterministic pruning.

| Variant | Pruning Approach | MCTS | Influence Feedback | NMSE ↓ |
|---|---|---|---|---|
| **Full IGSR** | Deterministic | ✓ | ✓ | **0.000787**±0.000532 |
| Full IGSR, LLM Pruning (*IGSR-Agent*) | LLM | ✓ | ✓ | 0.00127±0.000898 |
| w/o MCTS (Iterative + Influence Feedback), Deterministic Pruning | Deterministic | ✗ | ✓ | 0.626±1.35 |
| w/o MCTS (Iterative + Influence Feedback), LLM Pruning | LLM | ✗ | ✓ | 0.303±0.43 |
| w/o Influence Feedback (MCTS + Basic Feedback), Deterministic Pruning | Deterministic | ✓ | ✗ | (Not Valid) |
| w/o Influence Feedback (MCTS + Basic Feedback), LLM Pruning | LLM | ✓ | ✗ | 0.293±0.64 |
| w/o MCTS or Influence Feedback (Iterative + Basic Feedback), Deterministic Pruning | Deterministic | ✗ | ✗ | (Not Valid) |
| w/o MCTS or Influence Feedback (Iterative + Basic Feedback), LLM Pruning | LLM | ✗ | ✗ | 4.85±2.36 |

## G.3. Illustration of IGSR Equation Discovery

To provide a more concrete understanding of how IGSR discovers and refines equations, this section illustrates the process using two example runs (Run A and Run B) on the Lung Cancer (with Chemo. & Radio.) benchmark dataset. Figure 8 depicts the progression of the best Mean Squared Error (MSE) achieved on the test set as the Monte Carlo Tree Search (MCTS) expands more nodes. Each significant drop in MSE, indicating the discovery of a more accurate equation, is annotated on the plot (e.g., A0, B9). Below, we present the specific equations and their corresponding MSEs at these key points, showcasing the iterative improvements made by IGSR. This visualization highlights how the framework navigates the equation space, modifying and selecting terms to progressively enhance model performance.

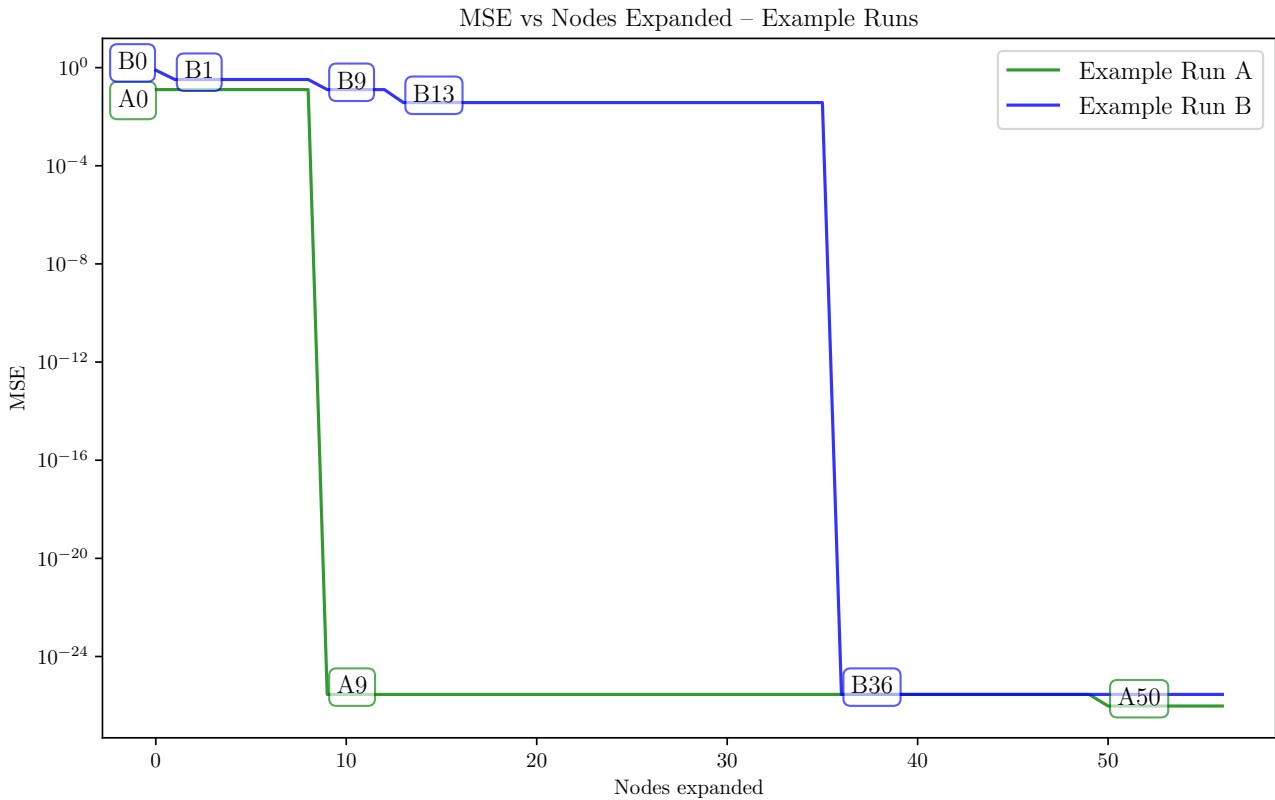

*Figure 8.* **Illustration of IGSR Equation Discovery Process.** Test MSE achieved versus MCTS nodes expanded for two example IGSR runs (Run A: green, Run B: blue) on the Lung Cancer (with Chemo. & Radio.) dataset. Points A0, A9, A50 for Run A, and B0, B1, B9, B13, B36 for Run B, indicate instances where a new, more accurate equation was discovered. The equations corresponding to these points are detailed below.

The equations discovered at pivotal moments during these IGSR runs are as follows:

EXAMPLE RUN A

- **Point A0** (Node 0):
  - MSE: 0.127
  - Equations:

$$dv\_dt = 0.02425 \cdot \text{cancer\_volume} - 0.08865 \cdot \text{chemo\_concentration}$$
$$- 0.03812 \cdot \text{chemo\_dosage}$$
$$- 0.02755 \cdot \text{cancer\_volume} \cdot \text{chemo\_concentration}$$
$$- 0.04786 \cdot \text{cancer\_volume} \cdot \text{radiotherapy\_dosage}$$
$$+ 0.4913\sqrt{\text{cancer\_volume}}$$
$$dc\_dt = -0.5 \cdot \text{chemo\_concentration} + \text{chemo\_dosage}$$

- **Point A9** (Node 9):
  - MSE: $2.85 \times 10^{-26}$
  - Equations:

$$dv\_dt = 0.1389 \cdot \text{cancer\_volume}$$
$$- 0.028 \cdot \text{cancer\_volume} \cdot \text{chemo\_concentration}$$
$$- 0.04776 \cdot \text{cancer\_volume} \cdot \text{radiotherapy\_dosage}$$
$$- 0.01453 \cdot \text{cancer\_volume} \cdot \ln(\text{cancer\_volume})$$
$$dc\_dt = -0.5 \cdot \text{chemo\_concentration} + \text{chemo\_dosage}$$

- **Point A50** (Node 50):
  - MSE: $9.59 \times 10^{-27}$
  - Equations:

$$dv\_dt = 0.1389 \cdot \text{cancer\_volume}$$
$$- 0.028 \cdot \text{cancer\_volume} \cdot \text{chemo\_concentration}$$
$$- 0.01453 \cdot \text{cancer\_volume} \cdot \ln(\text{cancer\_volume})$$
$$- 0.02388 \cdot \text{cancer\_volume} \cdot (\text{radiotherapy\_dosage})^2$$
$$dc\_dt = -0.5 \cdot \text{chemo\_concentration} + \text{chemo\_dosage}$$

EXAMPLE RUN B

- **Point B0** (Node 0):
  - MSE: 0.785
  - Equations:

$$dv\_dt = 0.04326 \cdot \text{cancer\_volume} + 0.1558 \cdot \text{chemo\_concentration}$$
$$+ 0.1157 \cdot \text{chemo\_dosage}$$
$$- 0.02769 \cdot \text{cancer\_volume} \cdot \text{chemo\_concentration}$$
$$- 0.04836 \cdot \text{cancer\_volume} \cdot \text{radiotherapy\_dosage}$$
$$dc\_dt = -0.5 \cdot \text{chemo\_concentration} + \text{chemo\_dosage}$$

- **Point B1** (Node 1):
  - MSE: 0.328
  - Equations:

$$dv\_dt = 0.03643 \cdot \text{cancer\_volume} - 0.08611 \cdot \text{chemo\_concentration}$$
$$- 0.02803 \cdot \text{chemo\_dosage}$$
$$- 0.02736 \cdot \text{cancer\_volume} \cdot \text{chemo\_concentration}$$
$$- 0.04801 \cdot \text{cancer\_volume} \cdot \text{radiotherapy\_dosage}$$
$$+ 0.6666 \ln(\text{cancer\_volume} + 1)$$
$$dc\_dt = -0.5 \cdot \text{chemo\_concentration} + \text{chemo\_dosage}$$

- **Point B9** (Node 9):
  - MSE: 0.127
  - Equations:

$$dv\_dt = 0.02425 \cdot \text{cancer\_volume} - 0.08865 \cdot \text{chemo\_concentration}$$
$$- 0.03812 \cdot \text{chemo\_dosage}$$
$$- 0.02755 \cdot \text{cancer\_volume} \cdot \text{chemo\_concentration}$$
$$- 0.04786 \cdot \text{cancer\_volume} \cdot \text{radiotherapy\_dosage}$$
$$+ 0.4913\sqrt{\text{cancer\_volume}}$$
$$dc\_dt = -0.5 \cdot \text{chemo\_concentration} + \text{chemo\_dosage}$$

- **Point B13** (Node 13):
  - MSE: 0.0376
  - Equations:

$$dv\_dt = 0.005864 \cdot chemo\_concentration + 0.007118 \cdot chemo\_dosage$$
$$- 0.02803 \cdot cancer\_volume \cdot chemo\_concentration$$
$$- 0.04733 \cdot cancer\_volume \cdot radiotherapy\_dosage$$
$$- 1.708 \ln(cancer\_volume + 1)$$
$$+ 1.544 \sqrt{cancer\_volume}$$
$$dc\_dt = -0.5 \cdot chemo\_concentration + chemo\_dosage$$

- **Point B36** (Node 36):
  - MSE: $2.85 \times 10^{-26}$
  - Equations:

$$dv\_dt = 0.1389 \cdot cancer\_volume$$
$$- 0.028 \cdot cancer\_volume \cdot chemo\_concentration$$
$$- 0.04776 \cdot cancer\_volume \cdot radiotherapy\_dosage$$
$$- 0.01453 \cdot cancer\_volume \cdot \ln(cancer\_volume)$$
$$dc\_dt = -0.5 \cdot chemo\_concentration + chemo\_dosage$$

## G.4. Investigation of MCTS rollout depth

The Monte Carlo Tree Search (MCTS) component of IGSR plays a crucial role in systematically exploring the vast space of possible equations. A key parameter within MCTS is the rollout depth, which determines how far into the future the search simulates potential sequences of actions (in our case, "propose then prune terms" equation refinement steps) to estimate the value of a given state (equation). Deeper rollouts can provide more accurate value estimates, potentially leading to better search decisions and ultimately more accurate discovered equations. However, they also incur a higher computational cost. This investigation aims to quantify the impact of varying MCTS rollout depths on the performance of IGSR, helping to understand this trade-off.

We evaluated the effect of rollout depth on two distinct benchmark datasets: the simulated **Lung Cancer (with Chemo. & Radio.)** dataset and the real-world **RNA Polymerase** dataset. Three MCTS configurations were tested:

- **No Rollout (Heuristic MCTS)**: In this configuration, the MCTS does not perform any simulation. Instead, the value of a newly expanded node (representing an equation) is directly estimated using an immediate heuristic, which in our case is the negative validation Mean Squared Error (MSE) of the equation at that node. This reward is then directly backpropagated.
- **Rollout Depth 1**: After selecting a leaf node for expansion, the MCTS performs a simulation of one additional propose-and-prune cycle. The reward obtained from this single-step rollout is used for backpropagation.
- **Rollout Depth 2**: Similar to the above, but the simulation (rollout) extends for two propose-and-prune cycles.

For each configuration and dataset, performance was measured by the test MSE, averaged over 10 random seeds, with 95% confidence intervals reported. The results are presented in Table 13. The total iteration budget of 15 rather than 30 was used in this experiment.

*Table 13.* **Impact of MCTS Rollout Depth on IGSR Performance.** Test MSE (mean ± 95% CI) on the Lung Cancer with Chemo. & Radio. (simulated) and RNA Polymerase (real-world) datasets for different MCTS rollout depths. Results are averaged over 10 seeds. Lower MSE is better. The best performing variation for each dataset is indicated in bold.

| MCTS Rollout Variation | Lung Cancer with Chemo. & Radio. MSE ↓ | RNA Polymerase MSE ↓ |
|---|---|---|
| No Rollout (Heuristic MCTS) | 0.0571 ± 0.0367 | 0.0118 ± 0.000445 |
| Rollout Depth 1 | 0.0211 ± 0.0131 | 0.0114 ± 0.000720 |
| Rollout Depth 2 | **0.0130** ± 0.0123 | **0.0113** ± 0.000523 |

The findings from this investigation, as summarized in Table 13, indicate a trend consistent with expectations for MCTS performance. For both the simulated Lung Cancer with Chemo. & Radio. dataset and the real-world RNA Polymerase dataset, increasing the rollout depth generally leads to improved performance (i.e., lower test MSE). Specifically, moving from no rollout to a rollout depth of 1, and further to a rollout depth of 2, resulted in progressively lower mean MSE values.

On the Lung Cancer with Chemo. & Radio. dataset, a rollout depth of 2 achieved the lowest MSE (0.0130), a noticeable improvement over no rollout (0.0571) and rollout depth 1 (0.0211). Similarly, for the RNA Polymerase dataset, rollout depth 2 yielded the best mean MSE (0.0113), compared to 0.0118 for no rollout and 0.0114 for rollout depth 1. While the confidence intervals show some overlap, particularly for the RNA Polymerase dataset where the improvements are more modest, the consistent reduction in mean MSE suggests that deeper rollouts allow the MCTS to make more informed decisions during the search process. This enhanced lookahead capability helps IGSR to better navigate the complex equation space and identify more accurate symbolic models. This benefit, however, should be weighed against the increased computational time required for deeper rollouts in practical applications.

Elsewhere in this work, unless otherwise specified, we use the Heuristic MCTS configuration.

### G.5. Convergence Efficiency vs. Genetic Programming

A critical aspect of any equation discovery algorithm is its efficiency in navigating the vast search space of potential mathematical expressions. Rapid convergence to accurate and parsimonious solutions is highly desirable, as it reduces computational cost and accelerates the scientific discovery process. This section investigates the convergence efficiency of IGSR compared to a standard Genetic Programming (GP) approach, GPLearn. We aim to demonstrate that the structured, influence-based feedback mechanism within IGSR, particularly when coupled with MCTS, leads to faster identification of high-quality symbolic models.

**Experimental Setup and Iteration Comparability:** To assess convergence efficiency, we tracked the best test Mean Squared Error (MSE) achieved against the number of "major evaluation iterations" for both IGSR and GPLearn on the Lung Cancer (with Chemo. & Radio.) benchmark dataset. Experiments were conducted over 10 random seeds, and we report the mean MSE ± 95% confidence intervals.

For IGSR, an "iteration" is defined as one MCTS node expansion. Each expansion involves the core propose-and-prune cycle: the LLM proposes candidate basis functions, these are evaluated, influence scores are calculated, and the LLM prunes the terms to form a new candidate equation. This new equation is then fully evaluated.

For GPLearn, an "iteration" corresponds to one generation. In each generation, a population of candidate equations undergoes genetic operations (e.g., crossover, mutation), and each new individual (equation) in the subsequent population is evaluated.

While the precise computational operations within an IGSR MCTS expansion (involving LLM calls and influence score calculations) differ from those in a GPLearn generation (dominated by genetic operations and fitness evaluations of a population), both represent a fundamental step where a set of new candidate equations are generated, fully evaluated, and used to guide the subsequent search. By plotting performance against these respective "major evaluation iterations" or, more broadly, "number of candidate equations fully evaluated," we gain insight into how quickly each algorithm explores promising regions of the equation space and refines its solutions. This provides a broadly comparable measure of how efficiently each method utilizes its core search mechanism to improve solution quality.

The iteration count was set to 30 for both methods. GPLearn was run with the following parameters: `population_size = 1000`, `parsimony_coefficient = 0.01`, with the rest of the parameters set to their default values.

**Results:** The convergence behavior of IGSR and GPLearn is illustrated in Figure 9. The plot clearly shows that IGSR converges significantly faster to a substantially lower test MSE compared to GPLearn on this benchmark. IGSR rapidly identifies high-performing equations within a smaller number of major evaluation iterations. Specifically, IGSR reached a final mean test MSE of $0.0752 \pm 0.0295$. In contrast, GPLearn's convergence was slower, and it settled at a considerably higher final mean test MSE of $28.5 \pm 9.34$ within a comparable or number of its own evaluation iterations.

This superior convergence efficiency highlights the effectiveness of IGSR's approach. The detailed, per-term influence feedback allows the LLM to make more informed decisions during the pruning phase, leading to more targeted exploration. Coupled with the systematic search of MCTS, IGSR is able to more quickly discard unpromising avenues and focus on refining equation structures that demonstrate high predictive accuracy. This contrasts with the more stochastic and population-based exploration of GPLearn, which, in this case, required more evaluations to achieve a less optimal solution.

This demonstrates IGSR's ability not only to find accurate equations but to do so with greater efficiency in terms of the number of core search and evaluation steps.

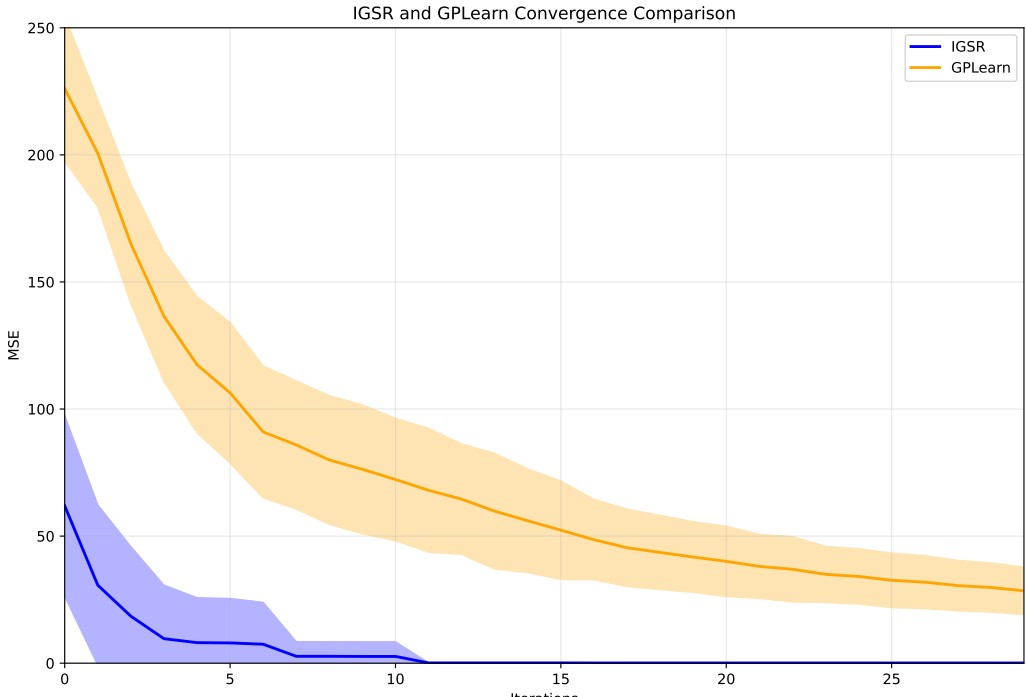

*Figure 9.* **Convergence Efficiency: IGSR vs. GPLearn.** Test MSE (mean ± 95% CI over 10 seeds) versus number of major evaluation iterations for IGSR (MCTS expansions) and GPLearn (generations) on the Lung Cancer (with Chemo. & Radio.) dataset. IGSR demonstrates faster convergence to a lower MSE, indicating superior search efficiency.

### G.6. Investigation of LLM sensitivity

Understanding the robustness of IGSR's advantages across different underlying Large Language Models (LLMs) is crucial for assessing the generalizability of our approach. This investigation aims to determine if the performance gains observed with IGSR are primarily due to its structured, influence-based feedback mechanism and search strategy, rather than being contingent on the specific capabilities of a single LLM. Demonstrating such robustness would imply that the architectural innovations of IGSR provide consistent benefits, offering practical value to users who may employ a variety of LLMs, including proprietary and open-weight models.

To evaluate this sensitivity, we conducted a comprehensive set of experiments comparing IGSR with the baseline LLM-driven methods: ZeroShot, ZeroOptim, and ICL. These four methods were evaluated on all six main benchmark datasets detailed in Section 4 and Appendix C: Lung Cancer, Lung Cancer (with Chemo.), Lung Cancer (with Chemo. & Radio.), COVID-19, RNA Polymerase, and Warfarin PK. For each method and dataset combination, we utilized a diverse suite of nine different base LLMs. The models used, their classification, and version notes are as follows:

- **Open-weight Models:**
    - `Llama-3.3-70B`: `Llama-3.3-70B-Instruct`, version 5.
    - `DeepSeek V3`: `DeepSeek V3`, version `0324`.
- **Proprietary Models:**
    - `GPT-4`: `gpt-4`, version `turbo-2024-04-09`.
    - `GPT-4o-Mini`: `gpt-4o-mini`, version `2024-07-18`.
    - `GPT-4o`: `gpt-4o`, version `2024-11-20`.
    - `GPT-4.1`: `gpt-4.1`, version `2025-04-14`.
    - `OpenAI o1`: `o1`, version `2024-12-17`.
    - `OpenAI o4-Mini`: `o4-mini`, version `2025-04-16`.
    - `OpenAI o3`: `o3`, version `2025-04-16`.

All inference hyperparameters were set to their default values for each model throughout the experiments.

The performance was measured using the Mean Squared Error (MSE) on the respective test sets, consistent with our main evaluations. Each experiment was run with 25 seeds[5] to ensure reliability, and average MSE values are reported.

Figure 10 presents bar plots for all six datasets, showing the MSE outcomes. The bar plots presented in facilitate a visual comparison of method performance across different LLMs. Each subfigure in Figure 10 corresponds to a specific dataset. Within each such dataset-specific plot, results for the four methods – ZeroShot, ZeroOptim, ICL, and IGSR – are displayed in separate panels. In every panel, the x-axis lists the different base LLMs, while the y-axis represents the Mean Squared Error (MSE), where lower bars indicate better performance. The ± 95% CI is shown as error bars. For improved clarity, the y-axis MSE values are presented on a logarithmic scale. Notably, due to its significantly MSE, the ZeroShot method is plotted with a y-axis range distinct from the other three methods within each dataset. Conversely, the y-axis ranges for ZeroOptim, ICL, and IGSR are all aligned with each other within the context of a single dataset to enable their direct comparison.

---

[5]Except `Llama-3.3-70B-Instruct`, where 10 seeds were used, due to limitations to model throughput that was available at the time of the experiments.

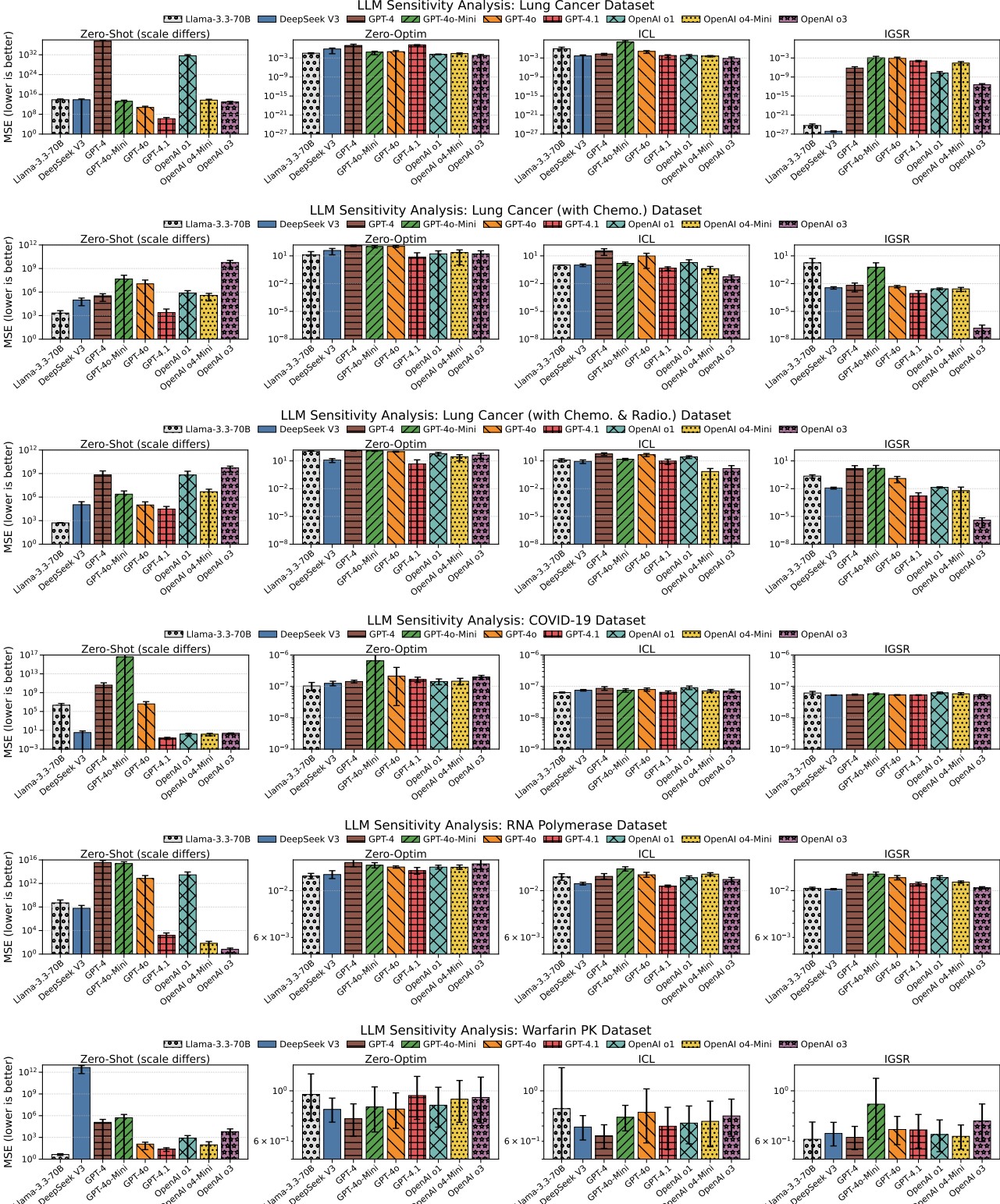

*Figure 10.* MSE (± 95% CI error bars) on benchmark datasets for ZeroShot, ZeroOptim, ICL, and IGSR across various base LLMs, including open-weight models. Each subfigure represents a different dataset. IGSR outperforms the other methods across the datasets and base LLMs with few exceptions. **Note:** The y axis (MSE) scale is logarithmic. For each dataset: the MSE range is different for the ZeroShot baseline, as its performance is significantly worse; the other methods are shown over matching ranges for clarity.

Across all datasets and underlying LLM choices, IGSR consistently outperforms ZeroShot, ZeroOptim, and ICL, with only a few exceptions. This strongly suggests that the structured feedback and systematic search strategy integral to IGSR are key drivers of its enhanced equation discovery capabilities, providing a robust advantage irrespective of the specific foundational LLM employed. These findings underscore the value of IGSR's methodology in leveraging LLMs for scientific discovery.

### G.7. Investigation of Robustness to a Large Number of Irrelevant Features

Scientific datasets, particularly in fields like genomics or high-throughput screening, can often present a high-dimensional feature space where many features may be irrelevant or noisy. A critical aspect of an effective equation discovery method is its ability to identify the truly influential variables amidst a large number of distractors. This section investigates the robustness of IGSR to an increasing number of such irrelevant features.

We conducted experiments using the **Lung Cancer (with Chemo. & Radio.)** benchmark dataset. To simulate scenarios with varying degrees of feature space complexity, we augmented the original dataset by adding 5, 75, or 150 additional "distractor" features. These features were populated with values drawn from a standard normal distribution ($\mathcal{N}(0, 1)$) and were designed to have no causal relationship with the target variables. The performance of IGSR (denoted as "**IGSR**" in the table) was compared against the LLM-based baselines: ZeroShot, ZeroOptim, and ICL. All methods were evaluated based on their Mean Squared Error (MSE) on the test set, averaged over 25 random seeds, with 95% confidence intervals reported.

The results are presented in Table 14.

*Table 14.* **Performance with Increasing Numbers of Irrelevant Features.** Test MSE (mean ± 95% CI) on the Lung Cancer (with Chemo. & Radio.) dataset, augmented with varying numbers of normally distributed, irrelevant features. Results are averaged over 25 seeds. Lower MSE is better. "+0 Features" corresponds to the original dataset.

| Method | +0 Features (Original) | +5 Features | +75 Features | +150 Features |
|---|---|---|---|---|
| ZeroShot | 2.54e+03 ± 2.74e+03 | 1.48e+03 ± 2.35e+03 | 3.08e+03 ± 6.18e+03 | 1.57e+04 ± 2.18e+04 |
| ZeroOptim | 122 ± 6 | 142 ± 12.2 | 131 ± 27.8 | 132 ± 22.7 |
| ICL | 63.3 ± 16.5 | 111 ± 32.6 | 118 ± 32.5 | 126 ± 22.5 |
| **IGSR** | **0.0141 ± 0.0087** | **0.0843 ± 0.0421** | **2.99 ± 5.95** | **4.88 ± 9.8** |

As evident from Table 14, IGSR consistently achieves the lowest MSE across all conditions, even when a substantial number of irrelevant features are introduced. While the performance of IGSR does show some degradation as the number of distractor features increases (MSE from 0.0141 with 0 extra features to 4.88 with 150 extra features), its accuracy remains significantly better – often by orders of magnitude – than the other LLM-based approaches. For instance, with 150 additional irrelevant features, IGSR achieves an MSE of 4.88, whereas the next best LLM-based method, ICL, has an MSE of 126.

The other methods (ZeroShot, ZeroOptim, ICL) show varied responses to the additional features but consistently perform much worse than IGSR. Their inability to effectively filter out the noise, even with optimization (ZeroOptim) or basic iterative feedback (ICL), highlights the challenge distractor features pose. These findings suggest that IGSR's architecture, particularly its use of per-term influence scores and systematic search (with MCTS), equips the LLM with a more robust mechanism to discern and prioritize relevant features.

### G.8. Investigation of Synthetic Model Benchmark

To further assess the robustness and generalization capabilities of IGSR, particularly its performance on model structures that may not be prevalent in its LLM's training data, we conducted evaluations on a procedurally generated synthetic model benchmark. By creating synthetic models with known ground-truth equations incorporating diverse mathematical operators and interactions, we can test the ability of IGSR to discover accurate equations even when faced with potentially unfamiliar functional forms.

These synthetic datasets were derived by introducing specific modifications to the underlying structure of the previously described lung cancer model with chemotherapy and radiotherapy. The modifications included the incorporation of trigonometric operators, division operators, and novel interaction terms, resulting in five distinct synthetic models:

- **Synthetic 1 (inc. $\gamma \sin(\omega t)$):** Introduces a sinusoidal forcing term. The underlying differential equation is:

$$\frac{dx(t)}{dt} = \left( \rho \log \left( \frac{K}{x(t)} \right) - \beta_c C(t) - (\alpha_r d(t) + \beta_r d(t)^2) + \gamma \sin(\omega t) \right) x(t)$$

- **Synthetic 2 (inc. $-\delta I(t)$):** Incorporates an additional linear negative feedback term $I(t)$.

$$\frac{dx(t)}{dt} = \left( \rho \log \left( \frac{K}{x(t)} \right) - \beta_c C(t) - (\alpha_r d(t) + \beta_r d(t)^2) - \delta I(t) \right) x(t)$$

- **Synthetic 3 (inc. $\log(\frac{K}{x(t)+N(t)})$):** Modifies the growth term to include an additional variable $N(t)$ in the denominator of the logistic term.

$$\frac{dx(t)}{dt} = \left( \rho \log \left( \frac{K}{x(t) + N(t)} \right) - \beta_c C(t) - (\alpha_r d(t) + \beta_r d(t)^2) \right) x(t)$$

- **Synthetic 4 (inc. $\epsilon \cos(\phi t)$):** Adds a cosine forcing term.

$$\frac{dx(t)}{dt} = \left( \rho \log \left( \frac{K}{x(t)} \right) - \beta_c C(t) - (\alpha_r d(t) + \beta_r d(t)^2) + \epsilon \cos(\phi t) \right) x(t)$$

- **Synthetic 5 (inc. $\theta C(t)d(t)$):** Introduces a multiplicative interaction term between chemotherapy $C(t)$ and radiotherapy $d(t)$.

$$\frac{dx(t)}{dt} = \left( \rho \log \left( \frac{K}{x(t)} \right) - \beta_c C(t) - (\alpha_r d(t) + \beta_r d(t)^2) - \theta C(t)d(t) \right) x(t)$$

Here, $x(t)$ represents the tumor volume, $C(t)$ is the chemotherapy effect, $d(t)$ is the radiotherapy effect, and other parameters $(\rho, K, \beta_c, \alpha_r, \beta_r, \gamma, \omega, \delta, N(t), \epsilon, \phi, \theta)$ are constants or time-varying inputs specific to each synthetic model.

The performance of IGSR was compared against the LLM-based baselines: ZeroShot, ZeroOptim, and ICL. The results, in terms of test Mean Squared Error (MSE), are presented in Table 15.

*Table 15.* **Synthetic Model Benchmark Performance.** Test MSE (mean ± 95% CI) on five procedurally generated synthetic datasets. Results are averaged over 25 seeds. Lower MSE is better. IGSR consistently outperforms other LLM-based methods.

| Method | Synthetic 1 | Synthetic 2 | Synthetic 3 | Synthetic 4 | Synthetic 5 |
|---|---|---|---|---|---|
| ZeroShot | 3.42e+03 ± 2.59e+03 | 1.14e+04 ± 2.17e+04 | 1.11e+03 ± 470 | 3.12e+05 ± 6.36e+05 | 7.8e+03 ± 5.55e+03 |
| ZeroOptim | 153 ± 13.5 | 65.3 ± 7.53 | 106 ± 14.6 | 147 ± 14.8 | 118 ± 12.3 |
| ICL | 102 ± 17.7 | 48.3 ± 10.7 | 40.4 ± 11.3 | 89.2 ± 12 | 42.6 ± 16.3 |
| **IGSR** | **52.7 ± 0.307** | **0.189 ± 0.158** | **0.0272 ± 0.0153** | **52.5 ± 0.296** | **0.036 ± 0.0202** |

As shown in Table 15, IGSR consistently achieves significantly lower MSE compared to ZeroShot, ZeroOptim, and ICL across all five synthetic datasets. This strong performance on models with varied and potentially novel structures (such as trigonometric terms or modified logistic growth factors) underscores IGSR's ability to effectively search the equation space and adapt its discovery process. The granular, influence-based feedback appears crucial in guiding the LLM to identify relevant terms and construct accurate models, even when the underlying system dynamics deviate from more common forms. These results further highlight the robustness of the IGSR framework and its potential for discovering meaningful equations in diverse scientific domains.

### G.9. Generalization Study on the RNA Polymerase Dataset

A crucial test for any equation discovery method aiming for scientific relevance is its ability to generalize not just to held-out data from the same experiment, but to entirely new, independent measurements. A model that performs well on a new biological replicate is more likely to have captured true underlying biological principles rather than just fitting noise or artifacts specific to a single experiment.

To rigorously assess this, we obtained a second biological replicate for the RNA Polymerase II pausing measurements, representing a separate laboratory experiment (used as the target `pause_score`; henceforth, *"Replicate 2"*). We then took the final equations discovered by IGSR and several LLM-based baselines – which were trained and selected using only the original dataset (*"Replicate 1"*) – and evaluated their predictive performance on this new, unseen replicate without any re-fitting. This provides a strong test of out-of-distribution generalization.

The results, summarized in Table 16, demonstrate the robust generalization of the IGSR-discovered models. While all methods experienced a predictable and modest drop in performance when evaluated on the new replicate – an expected outcome given inter-experiment variability – IGSR maintained its superior accuracy.

On *Replicate 2*, IGSR achieved the lowest Mean Squared Error (MSE) of $0.00793 \pm 0.000214$ and the highest coefficient of determination ($R^2$) of $0.142 \pm 0.0232$. This indicates that the IGSR-discovered equation explained the most variance in the independent dataset, outperforming the next-best method, ICL, which achieved an MSE of $0.00815$ and an $R^2$ of $0.119$. The strong performance on a true biological replicate provides compelling evidence that the structured, influence-based feedback mechanism of IGSR guides the discovery process towards equations that capture genuine, reproducible biological principles.

*Table 16.* **Generalization to an Independent Biological Replicate.** Performance of discovered equations on the original test set (Replicate 1) versus a new, unseen biological replicate (Replicate 2) for the RNA Polymerase dataset. Models were trained/selected using only Replicate 1 data. Results are mean $\pm$ 95% CI over 25 seeds. IGSR shows the best generalization, maintaining the lowest MSE and highest $R^2$ on the new replicate.

| Method | Performance on Original Test Set (Replicate 1) | | | Performance on New Biological Replicate (Replicate 2) | | |
| --- | --- | --- | --- | --- | --- | --- |
| | MSE ↓ | $R^2$ ↑ | NRMSE ↓ | MSE ↓ | $R^2$ ↑ | NRMSE ↓ |
| ZeroShot | 1.35e+05 $\pm$ 1.75e+05 | -9.80e+06 $\pm$ 1.28e+07 | 1.55e+03 $\pm$ 1.15e+03 | 1.40e+05 $\pm$ 1.72e+05 | -1.52e+07 $\pm$ 1.87e+07 | 1.97e+03 $\pm$ 1.42e+03 |
| ZeroOptim | 0.0130 $\pm$ 0.000287 | 0.0724 $\pm$ 0.0199 | 0.963 $\pm$ 0.0101 | 0.00887 $\pm$ 0.000159 | 0.0406 $\pm$ 0.0188 | 0.979 $\pm$ 0.00937 |
| ICL | 0.0119 $\pm$ 0.000352 | 0.149 $\pm$ 0.0251 | 0.922 $\pm$ 0.0137 | 0.00815 $\pm$ 0.000246 | 0.119 $\pm$ 0.0262 | 0.938 $\pm$ 0.0140 |
| **IGSR** | **0.0115 $\pm$ 0.000312** | **0.176 $\pm$ 0.0226** | **0.907 $\pm$ 0.0125** | **0.00793 $\pm$ 0.000214** | **0.142 $\pm$ 0.0232** | **0.926 $\pm$ 0.0126** |

## G.10. Nested Validation Analysis

Both the influence-based pruning decisions and the MCTS node rewards in IGSR are computed on the validation split. A natural concern, raised during review, is whether repeatedly consulting the same validation set during search induces *search-time overfitting*. To test this directly, we re-ran IGSR under a *nested* protocol in which the validation data is partitioned into two disjoint subsets: *Val-inner*, used exclusively for influence-based term pruning, and *Val-outer*, used exclusively for the MCTS reward. If the default (shared-validation) configuration were overfitting the validation set, isolating these two uses should produce a measurable drop in held-out test performance.

Table 17 reports this comparison on two challenging datasets: Lung Cancer (with Chemo. & Radio.) and the high-dimensional RNA Polymerase dataset. Across 10 seeds, the difference in test MSE between the original and nested protocols is small and not statistically significant (paired $t$-test and Wilcoxon, $p > 0.05$ in both cases). This indicates that the default IGSR configuration does not suffer from meaningful search-time overfitting, and that sharing a single validation split for pruning and reward does not come at a measurable cost in generalization.

*Table 17.* **Effect of Nested Validation on Test MSE.** To assess potential search-time overfitting, the validation set was split into disjoint subsets: *Val-inner* (for influence-based pruning) and *Val-outer* (for MCTS reward computation). Evaluated across 10 independent seeds, the test MSE difference between the original and nested protocols is not statistically significant (paired t-test and Wilcoxon $p > 0.05$). This confirms the default IGSR configuration does not suffer from meaningful search-time overfitting.

| Dataset ($n = 10$ seeds) | Original Setup (Test MSE) ↓ | Nested Val Setup (Test MSE) ↓ | Mean Diff. (Nested - Orig.) | Paired t-test ($p$) | Significant? |
| --- | --- | --- | --- | --- | --- |
| Lung Cancer (with Chemo. & Radio.) | 0.0142 $\pm$ 0.0087 | 0.0159 $\pm$ 0.0105 | +0.00171 $\pm$ 0.00898 | 0.677 | No |
| RNA Polymerase | 0.0113 $\pm$ 0.0004 | 0.0115 $\pm$ 0.0006 | +0.000266 $\pm$ 0.000679 | 0.399 | No |

## G.11. Influence Score Variants

In IGSR, the per-term influence score, $\Delta_j$, is computed on a *validation* split by deleting a single term (setting $w_j \leftarrow 0$) while holding the remaining coefficients fixed. This is the explicit definition of $\Delta_j$ used throughout the method and its pruning phase. The corresponding pruning prompt includes, as an illustration, an OLS identity, $\Delta_k = \frac{w_k^2}{n} \sum_i \phi_k(x_i)^2$, with the comment "always $\geq 0$" and an instruction to "treat terms independently; no need to refit or update weights."

The applicability of these statements hinges on the OLS orthogonality that nullifies cross-terms, a property that holds for the *training* data at the optimum but does not necessarily extend to a disjoint validation set. To ensure this distinction is clear, the prompt contains an explicit note that validation-computed values "may not always be $\geq 0$".

Here, as part of an additional investigation, we also implement and evaluate two *refit-aware* influence score alternatives that align the definition more closely with extra-sum-of-squares logic.

INFLUENCE SCORE COMPUTATION

**Notation.** Let $X \in \mathbb{R}^{n \times p}$ and $y \in \mathbb{R}^n$ denote training data; $X_{\mathrm{val}} \in \mathbb{R}^{n_{\mathrm{val}} \times p}$, $y_{\mathrm{val}} \in \mathbb{R}^{n_{\mathrm{val}}}$ denote validation data; $W \in \mathbb{R}^{p \times m}$ the fitted full–model coefficients (one column per output). On validation, $\widehat{Y}_{\mathrm{val}} = X_{\mathrm{val}} W$, with residuals $R_{\mathrm{val}} = Y_{\mathrm{val}} - \widehat{Y}_{\mathrm{val}}$. We use $\mathrm{MSE}_{\mathrm{val}}(W)_j = \frac{1}{n_{\mathrm{val}}} \|y_{\mathrm{val}}^{(j)} - (X_{\mathrm{val}} W)_{:,j}\|_2^2$.

**(A) No–refit** (default in IGSR). Deleting term $k$ is implemented by $\widehat{y}_{\mathrm{val}, -k}^{(j)} = (X_{\mathrm{val}} W)_{:,j} - \varphi_k^{\mathrm{val}} w_{k,j}$, where $\varphi_k^{\mathrm{val}} = X_{\mathrm{val}} e_k$. Expanding the squares yields the exact validation–split change

$$\Delta_{k,j}^{\mathrm{val}} = \frac{2 w_{k,j}}{n_{\mathrm{val}}} (\varphi_k^{\mathrm{val}})^\top r_{\mathrm{val}}^{(j)} + \frac{w_{k,j}^2}{n_{\mathrm{val}}} \|\varphi_k^{\mathrm{val}}\|_2^2, \quad r_{\mathrm{val}}^{(j)} = y_{\mathrm{val}}^{(j)} - (X_{\mathrm{val}} W)_{:,j}.$$

The cross–term vanishes on the *training* split at the OLS optimum (residuals orthogonal to columns of $X$), making the classic "$\geq 0$" identity a good approximation there; on validation the cross–term generally does not vanish and $\Delta_{k,j}^{\mathrm{val}}$ can be slightly negative. This approach is how IGSR computes $\Delta_j$ before pruning, by default.

**(B) Refit–aware (full refit).** Define the refit–aware influence as the *validation* MSE change after *refitting on train* with column $k$ removed:

$$\Delta_{k,j}^{\mathrm{refit}} = \mathrm{MSE}_{\mathrm{val}}\big(W^{(-k)}\big)_j - \mathrm{MSE}_{\mathrm{val}}(W)_j, \qquad W_{:,j}^{(-k)} = \arg \min_{u \in \mathbb{R}^{p-1}} \frac{1}{n} \|y^{(j)} - X_{(-k)} u\|_2^2$$

(or ridge adds $\lambda \|u\|_2^2$). This aligns with extra-sum-of-squares on train and yields a clean validation readout.

**(C) Refit–aware (efficient)**, compatible with OLS and ridge. Let $A = X^\top X$ (OLS) or $A = X^\top X + \lambda I$ (ridge), $B = A^{-1}$. Partition index $k$ from the rest and denote $\alpha_k = B_{kk}$, $\beta_{-k} = B_{-k,k}$. Writing $W = \begin{bmatrix} W_{-k,:} \\ W_{k,:} \end{bmatrix}$,

$$W_{-k,:}^{(-k)} = W_{-k,:} - \frac{\beta_{-k}}{\alpha_k} W_{k,:}$$
$$\widehat{Y}_{\mathrm{val}}^{(-k)} = X_{\mathrm{val}}^{(-k)} W_{-k,:}^{(-k)}$$
$$\Delta_{k,j}^{\mathrm{refit}} = \frac{1}{n_{\mathrm{val}}} \left\| y_{\mathrm{val}}^{(j)} - \widehat{y}_{\mathrm{val}}^{(j)(-k)} \right\|_2^2 - \mathrm{MSE}_{\mathrm{val}}(W)_j.$$

This avoids an explicit fit–loop, reusing $B$ to update all $p$ leave–one–out solutions in $O(p^2)$ per term (after a one–time $O(p^3)$ factorization), and works columnwise for multi–output.

**Where this is used in IGSR.** The pipeline computes $\Delta_j$ on validation to guide pruning (Section 2, Appendix B.2); the prompt receives the per–term weights $w_j$ and validation–computed $\Delta_j$ to guide its term keep/drop decisions.

ABLATION ACROSS SIX BENCHMARKS: ACCURACY VS. COST

We compared the three influence score variants – (i) No–refit, (ii) Refit–aware (full), and (iii) Refit–aware (efficient) – across the six benchmark datasets described in Tables 18 and 19. To ensure a fair comparison of test performance, the same set of 15 random seeds was used for each variant.

A repeated-measures ANOVA confirmed that there was *no significant difference* in test MSE among the three variants on any of the six datasets (all $p \geq 0.09$).[6] As expected, the refit–aware variants incur a non–trivial extra cost (Table 19) without providing a corresponding gain in accuracy (Table 18).

---

[6]We conducted a one-factor repeated-measures ANOVA with Greenhouse-Geisser correction for each dataset. The statistics ($F(2, 18)$,

*Table 18.* Test MSE (mean $\pm$ 95% CI) across datasets for the three influence variants.

| Variant | COVID-19 | Lung Cancer (with Chemo. & Radio.) | Lung Cancer (with Chemo.) | Lung Cancer | RNA Polymerase | Warfarin PK |
|---|---|---|---|---|---|---|
| No–refit | 5.35e-08 $\pm$ 1.93e-09 | 0.0311 $\pm$ 0.027 | 0.00304 $\pm$ 0.00194 | 3.55e-09 $\pm$ 5.3e-09 | 0.0114 $\pm$ 0.000398 | 0.597 $\pm$ 0.0821 |
| Refit–aware (full) | 5.24e-08 $\pm$ 1.67e-09 | 0.0337 $\pm$ 0.0064 | 0.00903 $\pm$ 0.0043 | 0.0065 $\pm$ 0.00561 | 0.0118 $\pm$ 0.000373 | 0.622 $\pm$ 0.0943 |
| Refit–aware (efficient) | 5.14e-08 $\pm$ 1.97e-09 | 0.0476 $\pm$ 0.0226 | 0.00756 $\pm$ 0.00169 | 0.0028 $\pm$ 0.00192 | 0.0119 $\pm$ 0.000391 | 0.639 $\pm$ 0.0888 |

*Table 19.* Wall–clock optimization time (mean $\pm$ 95% CI; seconds).

| Variant | COVID-19 | Lung Cancer (with Chemo. & Radio.) | Lung Cancer (with Chemo.) | Lung Cancer | RNA Polymerase | Warfarin PK |
|---|---|---|---|---|---|---|
| No–refit | 0.752 $\pm$ 0.0103 | 4.46 $\pm$ 0.564 | 5.54 $\pm$ 0.712 | 2.83 $\pm$ 0.193 | 13.5 $\pm$ 1.17 | 0.482 $\pm$ 0.0415 |
| Refit–aware (full) | 0.891 $\pm$ 0.0159 | 10.6 $\pm$ 1.56 | 10.7 $\pm$ 1.22 | 9.87 $\pm$ 1.14 | 23.1 $\pm$ 1.59 | 0.607 $\pm$ 0.0848 |
| Refit–aware (efficient) | 0.779 $\pm$ 0.0139 | 4.55 $\pm$ 0.512 | 4.31 $\pm$ 0.61 | 5.04 $\pm$ 0.662 | 18 $\pm$ 1.69 | 0.53 $\pm$ 0.0753 |

DISTRIBUTION OF INFLUENCE VALUES

We examined the empirical distribution of $\Delta$ values encountered during runs for *Lung Cancer (with Chemo. & Radio.)* and *RNA Polymerase* across all three variants. Two representative histograms (symlog $x$–axis for readability; outliers lightly clipped at extreme quantiles) are included in Fig. 11–12.

**Key summaries:**

**Lung Cancer (with Chemo. & Radio.)**: share of influences (%)

| Variant | $\Pr(\Delta < 0)$ | $\Pr(\Delta = 0)$ | $\Pr(\Delta > 0)$ | $n$ |
|---|---|---|---|---|
| No–refit | 18.19 | – | 81.81 | 21,912 |
| Refit–aware (full) | 24.59 | 0.05 | 75.37 | 21,590 |
| Refit–aware (efficient) | 29.53 | 0.14 | 70.33 | 22,398 |

**RNA Polymerase**: share of influences (%)

| Variant | $\Pr(\Delta < 0)$ | $\Pr(\Delta = 0)$ | $\Pr(\Delta > 0)$ | $n$ |
|---|---|---|---|---|
| No–refit | 4.71 | – | 95.29 | 7,190 |
| Refit–aware (full) | 9.08 | 1.40 | 89.52 | 6,696 |
| Refit–aware (efficient) | 8.75 | 2.28 | 88.97 | 6,526 |

**Observations:** (i) negative values are infrequent, and where present their magnitudes are typically small compared to the bulk positive mass (see symlog histograms); (ii) the *shape* of the distribution is broadly similar across the three variants; (iii) taken together with the within–seed ANOVA analysis, these results suggest the validation–computed $\Delta$ is *stable enough* for pruning decisions even without per–term refits.

SUMMARY OF FINDINGS

- **The Nature of the Influence Score.** Because the influence score, $\Delta$, is computed on a validation set *without* refitting the model, it is not constrained to be non-negative. Occasional small negative values are an expected result of this design and indicate that a feature has low or redundant influence in the context of the full model.

- **Performance vs. Cost.** Our experiments across six diverse datasets show that the test accuracy of the default, no-refit, and refit-aware variants is statistically indistinguishable. Given that the refit-aware variants are more computationally expensive, this finding establishes the default **no-refit** method as an efficient approach that does not compromise test accuracy.

- **Implementation Flexibility.** For applications where a refit-aware analysis is specifically required, two alternatives are available via configuration. Setting `refit_aware=True` performs a full refit for each term, while `refit_aware_efficient=True` uses a faster computational update. Both methods provide the extra-sum-of-squares style score, $\Delta^{\text{refit}}$, evaluated on the validation set.

---

$p_{\text{GG}}$, partial $\eta^2$) were: Lung Cancer ($F = 0.19, p = 0.72, \eta_p^2 = 0.02$); Lung Cancer (with Chemo.) ($F = 2.17, p = 0.17, \eta_p^2 = 0.20$); Lung Cancer (with Chemo. & Radio.) ($F = 1.74, p = 0.22, \eta_p^2 = 0.16$); Warfarin PK ($F = 2.78, p = 0.09, \eta_p^2 = 0.24$); RNA Polymerase ($F = 1.49, p = 0.26, \eta_p^2 = 0.14$); and COVID-19 ($F = 1.05, p = 0.34, \eta_p^2 = 0.10$).

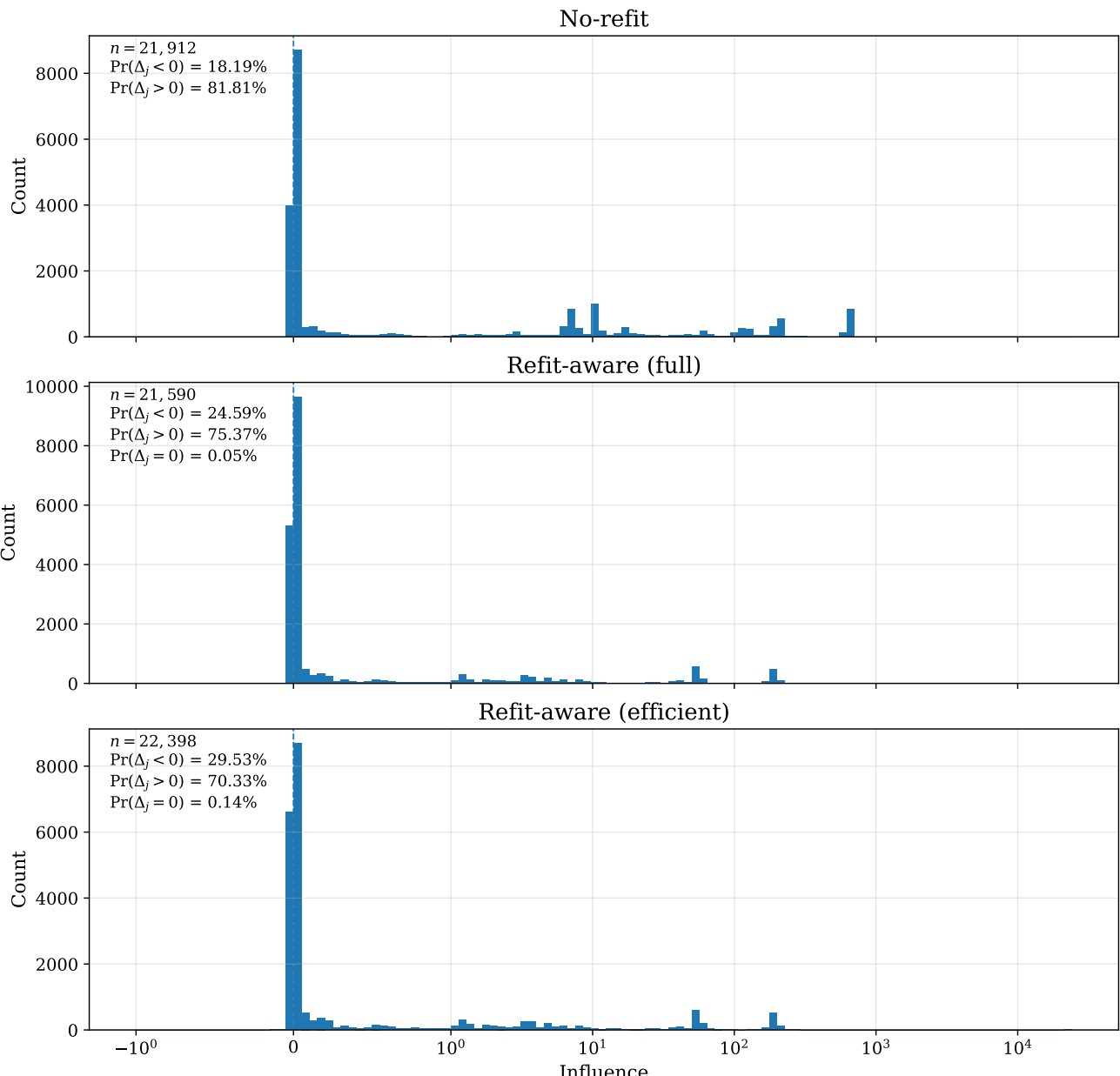

*Figure 11.* Distribution of influence values $\Delta$ on **Lung Cancer (with Chemo. & Radio.)** (symlog $x$–axis; extreme outliers clipped at the 0.01% tails for legibility).

## G.12. Robustness of Influence-Based Pruning

The cross-seed stability reported in the main text (Section 5.3, Figure 4) relies on the influence-based pruning step reliably selecting the right terms. Here we stress-test that step in two controlled settings that are classically challenging for a no-refit, marginal influence score: (i) *extreme multicollinearity* among candidate terms, and (ii) *interaction-only* (epistasis-like) signal whose individual constituents are uninformative. In both cases we compare the default no-refit influence against the two refit-aware variants of Appendix G.11.

**Collinearity stress test.** Figure 13 mixes the six ground-truth basis functions with four highly correlated "clones" and four distractors, and asks pruning to recover all six signal *groups*. The default no-refit influence recovers every signal group in all 15 seeds, even at clone correlation $\rho = 0.999$, with zero duplicate-group retentions; the refit-aware variants give identical

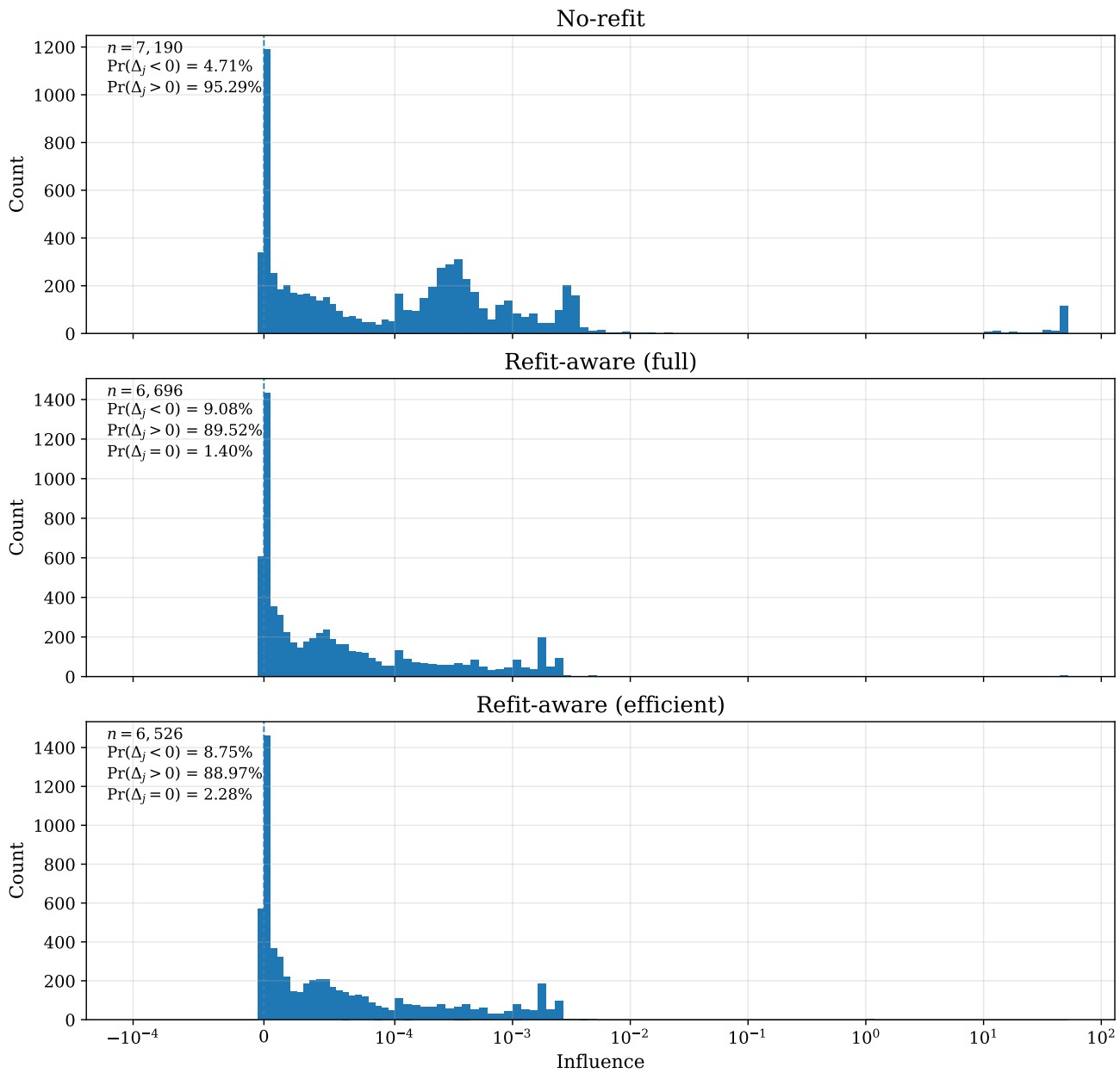

*Figure 12.* Distribution of influence values $\Delta$ on **RNA Polymerase** (symlog $x$–axis; extreme outliers clipped).

selections but are 2–7× slower.

**Interaction-only (epistasis-like) test.** Figure 14 constructs targets whose signal resides solely in interaction terms (e.g. $x_1 x_2$), so that the constituent marginals carry zero individual linear signal. Once the interaction term is present in the candidate pool, pruning retains it in all 15/15 seeds (ranks 1–2), while the marginal variables receive near-zero influence ($\sim 10^{-7}$–$10^{-18}$). This confirms that pruning operates on candidate *terms* rather than raw features, and does not discard a synergistic term merely because its constituents are individually weak.

## G.13. Cross-Seed Stability of Discovered Equations: Extended Analysis

This section expands the cross-seed stability summary in the main text (Section 5.3, Figure 4) with the full pairwise-similarity analysis over the 25 seeds on Lung Cancer (with Chemo. & Radio.). Figure 15 shows the Jaccard similarity between the

| Signal | Term | Group |
|---|---|---|
| | $x$ | 1 |
| | $x \cdot \log x$ | 2 |
| | $x \cdot C$ | 3 |
| | $x \cdot d$ | 4 |
| | $C$ | 5 |
| | $u^c$ | 6 |

| | Term | Group |
|---|---|---|
| Clone | $C_\rho$ | 5 |
| | $u^c_\rho$ | 6 |
| | $x \cdot C_\rho$ | 3 |
| | $x \cdot d_\rho$ | 4 |
| Distr. | $\sqrt{x}$ | – |
| | $\log x$ | – |
| | $x^2$ | – |
| | $u^c \cdot d$ | – |

*(a)* 14-term candidate pool. A signal *group* is covered if $\geq 1$ of its members is retained.

| | Signal-group recall (out of 6) | | | |
|---|---|---|---|---|
| Influence variant | $\rho=0.95$ | $\rho=0.99$ | $\rho=0.999$ | Time (s) |
| No-refit (default) | 6/6 | 6/6 | 6/6 | 0.03 |
| Refit-aware (full) | 6/6 | 6/6 | 6/6 | 0.20 |
| Refit-aware (efficient) | 6/6 | 6/6 | 6/6 | 0.05 |

*(b)* Results (15 seeds per cell). All three variants perfectly separate the 6 true signal components from 4 correlated clones and 4 distractors for all $\rho$. Test MSE $< 10^{-27}$; zero duplicate-group retentions (groups where both original and clone are retained). Time is mean per seed.

*Figure 13.* **Collinearity stress test for influence-based pruning** on Lung Cancer (with Chemo. & Radio.), 15 seeds. Variables: $x =$ tumor volume, $C =$ chemo. concentration, $u^c =$ chemo. dosage, $d =$ radiotherapy dose. **Setup.** A 14-term candidate pool mixes the 6 ground-truth basis functions with 4 correlated *clones* and 4 *distractors* (a). Each clone $v_\rho$ of a variable $v \in \{C, u^c, d\}$ is constructed as $v_\rho = v + \text{sd}(v)\sqrt{\rho^{-2}-1}\,\varepsilon$, $\varepsilon \sim \mathcal{N}(0,1)$, so that $\text{Corr}(v, v_\rho) \approx \rho$. Clones share the signal group of their original (e.g. $\{C, C_\rho\}$). After fitting with all 14 terms on train, per-term influence is computed on validation, aggregated across outputs via $\max$, and the top-$K$ ($K{=}6$) terms are retained, matching the default pruning protocol. **Result.** The default no-refit influence correctly identifies all 6 signal groups in every seed, even at $\rho{=}0.999$. Refit-aware variants yield identical selections but are $\approx$2–7$\times$ slower (full refit requires $p$ separate least-squares solves).

| | | Signal? | |
|---|---|---|---|
| | Term | Exp. 1 | Exp. 2 |
| Marginals | $x_1$ | | |
| | $x_2$ | | |
| | $x_1 + x_2$ | | |
| Interactions | $x_1 x_2$ | ✓ | ✓ |
| | $x_3 x_4$ | † | ✓ |
| | $x_1 x_3$ | | |
| | $x_2 x_4$ | | |
| Other | $x_1^2$ | | |
| | $x_2^2$ | | |
| | $x_3$ ‡ | | |
| | $x_4$ ‡ | | |

† Not in pool for Exp. 1 (10 terms); added for Exp. 2 (11 terms).
‡ In Exp. 2, $x_3$ and $x_4$ are zero-marginal constituents of the signal term $x_3 x_4$, analogous to $x_1$, $x_2$ in Exp. 1.

*(a)* Candidate pool. Marginals $x_1$, $x_2$ (and also $x_3$, $x_4$ in case of Exp. 2) have zero individual contribution by construction.

**Experiment 1** $(y = x_1 x_2 + \varepsilon, \; K{=}1)$

| Influence variant | $x_1 x_2$ retained | Test MSE | Time (s) |
|---|---|---|---|
| No-refit (default) | 15/15 | 0.0025 | 0.001 |
| Refit-aware (full) | 15/15 | 0.0025 | 0.005 |
| Refit-aware (efficient) | 15/15 | 0.0025 | 0.002 |

**Experiment 2** $(y = x_1 x_2 + x_3 x_4 + \varepsilon, \; K{=}2)$

| Influence variant | Both retained | Test MSE | Time (s) |
|---|---|---|---|
| No-refit (default) | 15/15 | 0.0025 | 0.001 |
| Refit-aware (full) | 15/15 | 0.0025 | 0.006 |
| Refit-aware (efficient) | 15/15 | 0.0025 | 0.002 |

*(b)* Results (15 seeds each). Experiment 2 adds $x_3 x_4$ to the pool (11 terms) and prunes to $K{=}2$. Test MSE matches the noise floor ($\sigma^2 = 0.05^2 = 0.0025$) in all runs.

*Figure 14.* **Interaction-only signal recovery under pruning. Setup.** Features $x_1, \ldots, x_{10} \overset{\text{iid}}{\sim} \mathcal{N}(0,1)$; targets are pure interaction terms plus noise $\varepsilon = 0.05\,\mathcal{N}(0,1)$. By construction, the marginal variables $(x_1, x_2, \ldots)$ carry zero individual linear signal; all predictive power resides in the interaction(s). This tests whether influence-based pruning can recover synergistic terms whose individual constituents are uninformative. **Result.** All three influence variants assign rank 1 (Exp. 1) or ranks 1–2 (Exp. 2) to the true interaction(s) in every seed. We also observed that e.g. the marginals $x_1$, $x_2$ received near zero influence ($\sim$$10^{-7}$–$10^{-18}$). These results demonstrate that pruning operates on candidate terms, rather than raw features, and correctly preserves the interaction-only signal.

canonicalized final term sets of every pair of seeds (seeds sorted by validation MSE), and Figure 16 shows the distribution of these pairwise similarities. The heatmap exhibits a clear block structure that mirrors the two performance tiers of Figure 4: the low-validation-MSE seeds (top-left) form a near-identical block ($J \approx 1$, the near-exact-recovery tier), while the remaining seeds form a second, internally-coherent cluster that differs from the first mainly in how the Gompertz growth nonlinearity is represented. Consistent with this, the distribution is concentrated away from zero (mean $J = 0.65$, median $0.50$), with a substantial spike at $J = 1$: 86 of the 300 seed pairs share *identical* term sets. This corroborates the main-text conclusion that the discovered structure is stable across seeds, with variation confined to near-equivalent approximations of a single nonlinear term.

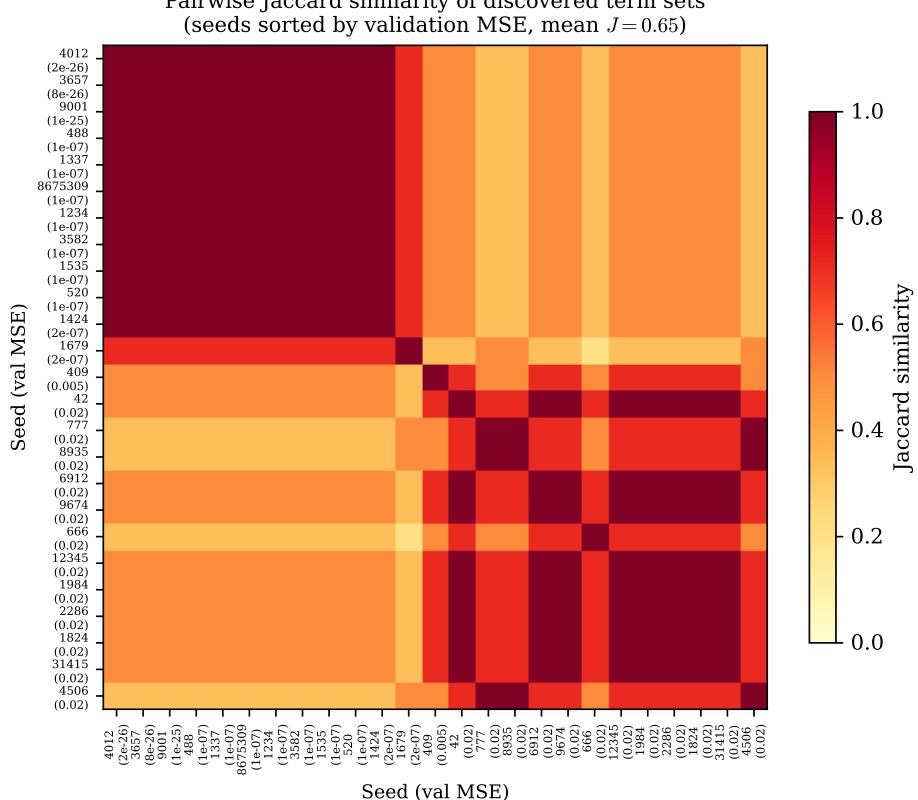

*Figure 15.* **Pairwise Jaccard similarity of discovered term sets** across 25 seeds on Lung Cancer (with Chemo. & Radio.), after canonicalization. Seeds are sorted by validation MSE (shown in parentheses). The dark top-left block ($J \approx 1$) is the near-exact-recovery tier; the lower-right cluster is internally similar but approximates the Gompertz nonlinearity differently. Mean pairwise $J = 0.65$.

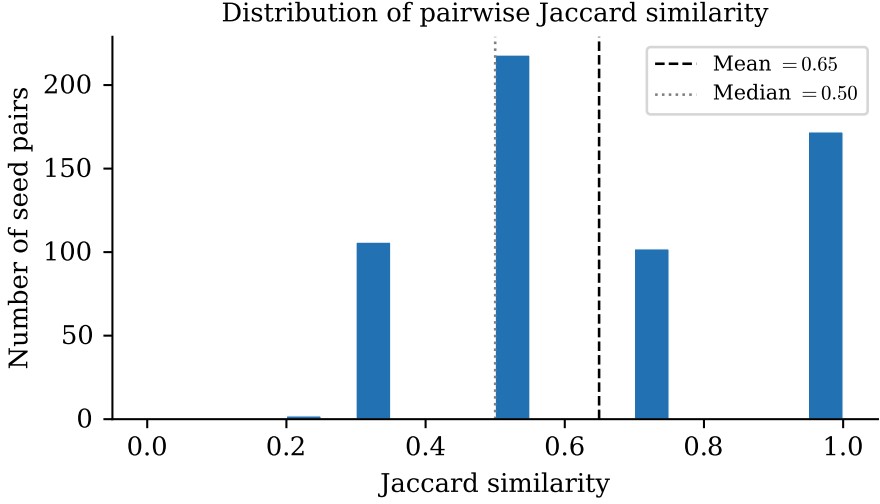

*Figure 16.* **Distribution of pairwise Jaccard similarity** over the 300 seed pairs (same data as Figure 15). Mean $= 0.65$, median $= 0.50$. The mass at $J = 1$ corresponds to seed pairs with identical canonical term sets (86 of 300 pairs).

### G.14. Term-local Optimization

To enhance IGSR's flexibility, we introduce a variant that allows for the discovery of optimal scalar constants *within* the basis functions proposed by the LLM. This extension, which we refer to as IGSR with Term-Local Optimization (IGSR-TLO), addresses scenarios where an equation's true functional form involves specific constants that an LLM is unlikely to guess *a priori* (e.g., a decay rate in an exponential term or a frequency in a sinusoidal term). IGSR-TLO integrates a dedicated optimization step to fine-tune these constants, enabling the discovery of more precise and potentially more accurate symbolic models.

The core principle is to augment the standard IGSR loop: the "Propose" agent can now suggest basis functions containing tunable parameters, which are subsequently optimized before the "Prune" agent performs its influence-guided term selection.

**Parametric Basis Functions**    In this variant, the "Propose" LLM agent can include tunable scalar constants in its proposed terms using the syntax `c(init)`, where `init` is an initial value for the parameter. For example, an agent might propose a term like `np.sin(c(1.0) * x1)`. Each basis function $\psi_j$ can thus be a parametric function $\psi_j(\mathbf{x}; \theta_j)$, where $\theta_j$ is a vector of the tunable constants within that term. The collection of all such constants across all candidate terms is denoted by $\theta$.

To maintain the model's primary structure as a linear combination of basis functions, the use of `c()` is disallowed as a direct outer multiplier (e.g., `c(1.5) * x1`), as the per-term linear coefficient $w_j$ is already learned by the outer model. The "Propose" agent's prompt in IGSR-TLO variant contains clear instructions and examples to reflect this setup.

**Term-Local Optimization Objective**    After the "Propose" agent suggests a new set of candidate terms, an additional optimization step is introduced before pruning. This step tunes the vector of all constants $\theta$ by minimizing an objective function $J(\theta)$. Crucially, this is a nested optimization problem. For any given set of constants $\theta$, the outer linear model weights $W$ are first re-computed by fitting on the training data. The objective $J(\theta)$ is then the Mean Squared Error (MSE) evaluated on the validation set using these optimal weights $W(\theta)$. This ensures that the constants are optimized for generalization performance.

Formally, for a given $\theta$, the design matrix on a data split $S \in \{\text{train, val}\}$ is $\Phi_S(\theta)$, where $[\Phi_S(\theta)]_{ik} = \psi_k(\mathbf{x}_i; \theta_k)$. The optimal outer weights are a function of $\theta$:

$$W(\theta) = \arg\min_{W'} \frac{1}{n_{\text{train}}m} \|Y_{\text{train}} - \Phi_{\text{train}}(\theta)W'\|_F^2 \ (+ \lambda\|W'\|_F^2 \text{ for ridge})$$

The objective for the inner-term constants is then to minimize the validation loss:

$$J(\theta) = \frac{1}{n_{\text{val}}m} \left\|Y_{\text{val}} - \Phi_{\text{val}}(\theta) \, W(\theta)\right\|_F^2$$

This objective is minimized with respect to $\theta$ using a quasi-Newton method (L-BFGS-B by default), with gradients estimated via finite differences.

**Dynamic Re-optimization and Agent Interaction**    A key feature of this variant is that the constants are not optimized once and then fixed. After optimization, the symbolic representation of the terms retains the `c(value)` markers, where `value` is now the optimized value. These symbolic forms are what the "Prune" agent sees in its feedback tables.

In each subsequent iteration of the IGSR loop, the constants within *all* surviving and newly proposed terms are re-optimized together. This dynamic re-optimization allows the ideal value for a constant in one term to adapt to the presence or absence of other terms in the model, preserving maximal flexibility throughout the discovery process. The "Prune" agent is explicitly prompted that these constants are not static and will be re-tuned in subsequent rounds.

**Algorithm Flow**    The IGSR-TLO propose-and-prune cycle proceeds as follows:

1. The LLM Propose agent suggests new terms, which may contain `c(init)` syntax. These are combined with surviving terms from the previous iteration.
2. The set of all constants $\theta$ from all current candidate terms is optimized by minimizing $J(\theta)$.
3. After optimization, the resulting terms are evaluated on the data splits (train, validation, test).

4. The standard pruning phase proceeds. The Prune agent receives the terms (with their optimized `c(value)` markers), their fitted outer weights $w_j$, and their per-term influence scores $\Delta_j$.

5. The agent returns keep/drop decisions. Surviving terms, retaining their `c()` markers, are passed to the next iteration.

This entire cycle is embedded within either the linear iterative refinement or the MCTS search strategy, just as in the standard IGSR framework.

### G.14.1. PROOF OF CONCEPT EXPERIMENT

To provide a clear illustration of the specific advantage offered by term-local optimization, we conducted a proof-of-concept experiment on a simple synthetic dataset. The data was generated from the ground-truth equation $y = 1/(0.123 + x_1^2)$, which contains a non-trivial constant, $0.123$, that an LLM is highly unlikely to propose spontaneously. We ran both the standard IGSR and the IGSR-TLO variant on this dataset, using their non-tree-based iterative refinement modes for simplicity.

As hypothesized, IGSR-TLO was uniquely capable of recovering the exact ground-truth equation. During its iterative search, the "Propose" agent eventually suggested the correct functional form with a tunable constant: `1 / (x_1**2 + c(init))`. The subsequent optimization step successfully tuned the constant to match the ground truth value of $0.123$. In the final pruning round, the feedback provided to the "Prune" agent was unambiguous. The correct parametric term had an influence score of $17.59$, while all other candidate terms had negligible influence (on the order of $10^{-17}$ or less). Guided by this overwhelming signal, the agent correctly kept only the single correct term and discarded all others, resulting in the exact solution with a final MSE of effectively zero ($1.07 \times 10^{-29}$). The final discovered equation was:

$$y = 1 \;/\; (x\_1**2 \;+\; 0.123)$$

By contrast, the standard IGSR variant was unable to discover the true equation. Without the ability to tune the constant, it could not find a single basis function to accurately model the data. Instead, it was forced to approximate the target function by constructing a linear combination of multiple, non-parametric basis functions. After ten iterations, the best model it found was a complex five-term approximation:

$$y = 17.33\ x\_1 + 102.2\ np.exp(-x\_1) + 38.73\ x\_1 * np.exp(-x\_1)$$
$$- 18.34\ np.abs(x\_1) - 186.4 * 1\ /\ (1 + np.exp(x\_1))$$

While this model achieved a respectable test MSE of $0.0372$, it failed to capture the simple, parsimonious structure of the underlying data-generating process. In this experimental setting, with a 10 iteration budget, IGSR-TLO was able to discover the exact expression in 10/25 runs, while vanilla IGSR was able discover it in 0/25 runs (never). The experiment clearly demonstrates that for problems where precise constants are integral to the model's form, the term-local optimization capability of IGSR-TLO is essential for discovering the correct symbolic solution.

### G.14.2. BENCHMARK RESULTS WITH TERM-LOCAL OPTIMIZATION

To assess the practical impact of term-local optimization, we compared the performance of IGSR-TLO against the vanilla IGSR framework across our six benchmark datasets. The results, presented in Table 20, show that the inclusion of tunable constants yields mixed outcomes depending on the dataset's characteristics. The experimental runs in this section used fewer total budget iterations (15 rather than 30) compared to the main results set in Table 2a.

*Table 20.* **Comparison of IGSR vs. IGSR-TLO.** Test MSE (mean $\pm$ 95% CI) across all benchmark datasets. Results are averaged over 25 seeds. Lower is better. Bold indicates the better-performing variant for each dataset (if confidence intervals overlap, only the mean of the better-performing variant is bold.)

| Benchmark Dataset | IGSR | IGSR-TLO |
|---|---|---|
| Lung Cancer | $0.0033 \pm 0.0034$ | **7.45e-06 $\pm$ 1.30e-05** |
| Lung Cancer (with Chemo.) | $0.0055 \pm 0.00109$ | **0.000905 $\pm$ 0.000530** |
| Lung Cancer (with Chemo. & Radio.) | **0.0518 $\pm$ 0.0171** | $0.234 \pm 0.127$ |
| COVID-19 | **5.12e-08 $\pm$ 1.32e-09** | $5.39e-08 \pm 1.78e-09$ |
| RNA Polymerase | **0.0112 $\pm$ 0.000314** | $0.0122 \pm 0.000119$ |
| Warfarin PK | **0.641 $\pm$ 0.107** | $0.655 \pm 0.108$ |

For the **Lung Cancer** and **Lung Cancer (with Chemo.)** datasets, IGSR-TLO demonstrates a marked improvement in accuracy. This is likely attributable to the increased flexibility afforded by parametric terms, which can more closely

approximate the underlying system dynamics. In the discovered equations for these datasets, we observe the selection of basis functions with optimized constants, such as those with the functional form $w_j \log(x + \theta)$ or $w_j \exp(\theta \cdot x)$. For instance, a high-performing equation discovered for the Lung Cancer (with Chemo.) task includes such adaptive terms:

$$
\begin{aligned}
\text{dv\_dt} = & - 0.0002279 \cdot \text{chemo\_concentration} + 0.0001333 \cdot \text{chemo\_dosage} \\
& - 2.358 \cdot \text{np.log}(\text{cancer\_volume} + \mathbf{4.06858}) \\
& - 0.02799 \cdot \text{cancer\_volume} \cdot \text{chemo\_concentration} \\
& + 1.736 \cdot \text{np.sqrt}(\text{cancer\_volume}) \\
& + 1.945 \cdot \text{np.exp}(\mathbf{-0.00875506} \cdot \text{cancer\_volume}) \\
\text{dc\_dt} = & - 0.5 \cdot \text{chemo\_concentration} + \text{chemo\_dosage}
\end{aligned}
$$

Conversely, on datasets like **Lung Cancer (with Chemo. & Radio.)** and **RNA Polymerase**, the performance of IGSR-TLO is slightly worse than that of vanilla IGSR. We speculate that this may be due to the significantly more complex and potentially non-convex optimization landscape introduced by the tunable constants $\theta$. The L-BFGS-B optimizer, while effective, may converge to local minima, particularly if the LLM's initial guesses for the constants are far from an optimal region. This could result in a set of basis functions that are locally optimal with respect to their internal constants but globally suboptimal for the final linear model, leading to a higher test MSE compared to the simpler, non-parametric terms found by the standard IGSR. Performance on the COVID-19 and Warfarin PK datasets was comparable between the two variants.

The promising results on several datasets indicate that IGSR-TLO is a valuable extension of the core framework. The approach and its variants warrant further investigation in future work, which could explore more sophisticated global optimization algorithms or improved heuristics for initializing the tunable constants. We provide the implementation of IGSR-TLO as described in this section with the codebase for this work.

### G.15. Investigation in the Context of Automated Feature Engineering

As discussed in Appendix A.6, IGSR shares conceptual parallels with Automated Feature Engineering (AFE). While IGSR focuses on discovering a concise, interpretable symbolic equation, the basis functions $\psi_j(\mathbf{x})$ can be viewed as engineered features that transform the input space to make it linearly separable. A relevant question, therefore, is how IGSR compares to established AFE frameworks and whether the restriction to a linear downstream model limits predictive performance compared to tree-based predictors.

To investigate this, we conducted a comparative study against *OpenFE* (Zhang et al., 2023), a state-of-the-art non-LLM AFE method known for its effectiveness on tabular data. We chose OpenFE as a robust representative of the AFE landscape because many recent LLM-based AFE tools (e.g., CAAFE (Hollmann et al., 2023)[7], FeatLLM (Han et al., 2024)[8]) are primarily intended for classification tasks or lack standardized regression implementations.

**Experimental Setup.** We evaluated four configurations:

1. **IGSR:** The standard IGSR framework where features are discovered by the LLM and the predictor is a linear model.

2. **IGSR + XGBoost:** The features $\psi_j(\mathbf{x})$ discovered by IGSR are frozen and used as inputs to an XGBoost regressor. This tests whether a non-linear predictor can extract more value from the discovered symbolic features.

3. **OpenFE + Linear Regression:** OpenFE is used to generate and select features, followed by a linear predictor.

4. **OpenFE + XGBoost:** OpenFE is used to generate and select features, followed by an XGBoost predictor.

To ensure fair comparison, we constrained OpenFE to select the top 6 features, matching the parsimony constraint imposed on IGSR (`keep_n_terms = 6`). We utilized the *Lung Cancer (with Chemo. & Radio.)* dataset (complex dynamics, low dimensionality) and the *RNA Polymerase* dataset (high dimensionality, 263 features). Performance is reported using Normalized Mean Squared Error (NMSE) as defined in (Shojaee et al., 2025b), over 10 seeds.

**Results on Complex Dynamics (Cancer Dataset).** As shown in Table 21, IGSR significantly outperforms the OpenFE baselines on the Cancer dataset. OpenFE, which relies on expanding a predefined set of operators, struggles to reconstruct

---

[7]Implementation at https://github.com/noahho/CAAFE
[8]Implementation at https://github.com/Sungwon-Han/FeatLLM

the specific functional forms (e.g., specific interaction terms or logarithmic growth laws) required to model the tumor dynamics accurately.

Interestingly, IGSR with a linear backend outperforms IGSR + XGBoost. Since the influence scores $\Delta_j$ used during the discovery process explicitly optimize for the reduction of linear least-squares error, the discovered basis functions tend to linearize the problem representation. In this context, the additional complexity of XGBoost does not yield a performance gain and may lead to slight overfitting compared to the linear analytic form.

*Table 21.* **AFE Comparison on Lung Cancer (with Chemo. & Radio.).** NMSE and Wall Clock time averaged over 10 seeds. IGSR outperforms standard AFE baselines by orders of magnitude. Replacing the linear solver with XGBoost does not improve performance, confirming that IGSR successfully discovers features that linearize the dynamics.

| Method (FE) | Predictor | NMSE ↓ | Wall Clock (s) ↓ |
|---|---|---|---|
| OpenFE | XGBoost | $0.183 \pm 0.038$ | $17.8 \pm 0.4$ |
| OpenFE | Linear Reg. | $0.202 \pm 0.041$ | $\mathbf{17.7} \pm 0.4$ |
| IGSR | XGBoost | $0.0084 \pm 0.0005$ | $340.4 \pm 56.3$ |
| IGSR | Linear Reg. | $\mathbf{0.0013} \pm \mathbf{0.0004}$ | $336.3 \pm 98.7$ |

*Table 22.* **AFE Scalability Comparison on RNA Polymerase (263 features).** NMSE and Wall Clock time averaged over 10 seeds. Traditional AFE (OpenFE) suffers from combinatorial explosion and times out. IGSR leverages LLM priors to navigate the high-dimensional space efficiently.

| Method (FE) | Predictor | NMSE ↓ | Wall Clock (s) ↓ |
|---|---|---|---|
| OpenFE | XGBoost | DNF | DNF |
| OpenFE | Linear Reg. | DNF | DNF |
| IGSR | XGBoost | $\mathbf{0.773} \pm 0.030$ | $318.9 \pm 22.1$ |
| IGSR | Linear Reg. | $0.791 \pm 0.039$ | $\mathbf{310.0} \pm 37.5$ |

DNF: Did Not Finish within the allocated 3-hour limit.

**Scalability on High-Dimensional Data (RNA Pol II).** A critical limitation of traditional AFE methods is the combinatorial explosion of the search space as the number of input features increases. The RNA Polymerase dataset contains 263 input features. As detailed in Table 22, OpenFE failed to complete a single run within a 3-hour cutoff. Analysis of the logs revealed that OpenFE attempted to evaluate a pool of $716,320$ candidate features, rendering it computationally intractable for this regime.

In contrast, IGSR leverages the LLM's semantic priors and the influence-based pruning mechanism to navigate this large feature space efficiently, converging in approximately 5 minutes. The performance of IGSR (with default Linear Regression) and IGSR + XGBoost was comparable on this dataset, suggesting that the discovered features captured the bulk of the signal regardless of the downstream predictor.

**Summary.** This investigation highlights two key advantages of IGSR over traditional AFE pipelines for scientific discovery:

1. **Scalability via Priors:** IGSR avoids the combinatorial explosion typical of generative feature engineering by using LLM priors to propose semantically relevant features, making it viable for high-dimensional datasets where traditional methods fail.

2. **Inherent Interpretability:** By optimizing features specifically for a linear backend, IGSR produces a model that is fully interpretable ($f(\mathbf{x}) = \sum w_j \psi_j(\mathbf{x})$). The results show that this interpretability does not come at the cost of accuracy.

**Extended comparison against additional AFE baselines.** The comparison above focuses on OpenFE as a representative non-LLM AFE method. To broaden it, Table 23 evaluates IGSR against a wider set of AFE approaches across all six benchmark datasets: a traditional white-box pipeline (AutoFeat with a LASSO regressor), a tree-optimized method (OpenFE + XGBoost), the SISSO-style pipeline SyMANTIC (Muthyala et al., 2025), and the LLM-based AFE framework

CAAFE (Hollmann et al., 2023), which we adapted to this regression setting. We additionally include an *IGSR + Base Feature Importances* variant in which the "Propose" agent also receives the relative importances of the top 20 raw features (computed once *a priori* via Ridge regression on standardized training data), to aid navigation of complex feature spaces while preserving symbolic sparsity. Against these baselines, plain IGSR attains the best mean MSE on 5/6 datasets; on RNA Polymerase it is comparable to CAAFE (overlapping 95% CIs), whereas the traditional AFE methods either time out (AutoFeat, OpenFE) or exceed memory limits (SyMANTIC) on this high-dimensional dataset. Table 24 additionally evaluates a traditional AFE baseline (AutoFeat + LASSO) on the LLM-SRBench discovery problems, where IGSR achieves superior predictive accuracy and symbolic recovery on both the in-distribution (ID) and out-of-distribution (OOD) sets.

*Table 23.* **Comparison against Automated Feature Engineering (AFE) Baselines.** We evaluate IGSR against *traditional* white-box AFE (AutoFeat with LASSO regressor, SyMANTIC), a tree-optimized AFE method (OpenFE + XGBoost), and an *LLM-based* AFE framework (CAAFE, adapted for regression). Because methods like CAAFE inherently evaluate all raw features globally, we also report an *IGSR + Base Feat. Imp.* variant where the LLM is provided with the relative importances of the top 20 raw features (computed once *a priori* via Ridge regression on standardized training data) to aid in navigating complex feature spaces while preserving symbolic sparsity. Evaluated over 25 seeds, IGSR and its variant maintain superior predictive accuracy, though CAAFE performs comparably to default IGSR on the RNAPol dataset. Notably, traditional AFE methods struggled on the high-dimensional RNAPol dataset: AutoFeat and OpenFE timed out due to combinatorial explosion, while SyMANTIC exceeded memory limits (496GB RAM). IGSR overcomes these limitations via semantic priors. **Abbreviations and configuration:** As per the main benchmark experiment (Table 2a).

| Method | LC
MSE ↓ | LC-C
MSE ↓ | LC-CR
MSE ↓ | C-19
MSE ↓ | RNAPol
MSE ↓ | Warf
MSE ↓ |
|---|---|---|---|---|---|---|
| AutoFeat (Horn et al., 2019) + LASSO | 15.7±26.8 | 0.0803±0.0445 | 0.0157±0.0029 | 2.23e-7±5.01e-8 | —[†] | 0.958±0.125 |
| OpenFE (Zhang et al., 2023) + XGBoost | 0.010±0.001 | 0.429±0.065 | 2.83±0.35 | 1.94e-7±2.11e-8 | —[†] | 14.8±0.8 |
| SyMANTIC (Muthyala et al., 2025) | 3.55e6±4.86e5 | 5.70e4±5.26e4 | 161±3.8 | 1.19e-3±1.99e-4 | —[‡] | 49.6±6.99 |
| CAAFE (Hollmann et al., 2023) | 6.64±8.68 | 92.5±39.6 | 150±22 | 2.32e-6±4.46e-7 | 0.0107±0.0002 | 0.807±0.143 |
| IGSR | **5.64e-5±6.79e-5** | **0.0013±0.0010** | **0.0141±0.0087** | 5.01e-8±1.78e-9 | 0.0111±0.0004 | **0.565±0.113** |
| IGSR + Base Feat. Imp. | 9.42e-5±1.46e-4 | 0.00255±0.00062 | 0.0177±0.0071 | **4.99e-8±1.27e-9** | **0.0103±0.0001** | 0.637±0.086 |

[†] Method timed out prior to reaching the 3-hour limit due to the combinatorial explosion resulting from the dataset's high feature count ($d = 263$).

[‡] Method exceeded memory limits on a machine with 496GB RAM due to the combinatorial explosion associated with the dataset's high feature count ($d = 263$).

*Table 24.* **Traditional AFE Baseline on LLM-SRBench.**
We also evaluate a traditional Automated Feature Engineering baseline (AutoFeat + LASSO) on the LLM-SRBench discovery problems. While AutoFeat explores a large space of analytical operator expansions, it lacks the semantic priors provided by an LLM. IGSR demonstrates superior predictive accuracy and symbolic recovery across both In-Distribution (ID) and Out-Of-Distribution (OOD) sets, highlighting the advantage of combining LLM proposals with influence-guided search.
**Experimental configuration:** Matches the main LLM-SRBench results (Table 2b).
**Formatting:** For each column, the **best** result is bolded.
**Values:** Metrics are reported as Median [IQR] for NMSE, and Mean ± Standard Deviation for others.

| Method | ID Test Set | | OOD Test Set | | Symbolic Metrics | |
|---|---|---|---|---|---|---|
| | NMSE ↓ | $Acc_{0.1}$ ↑ | NMSE ↓ | $Acc_{0.1}$ ↑ | Term Recall ↑ | Symbolic Accuracy ↑ |
| AutoFeat + LASSO | $9.20 \times 10^{-5}$ [$1.26 \times 10^{-3}$] | 0.423 ± 0.457 | 0.475 [109] | 0.405 ± 0.453 | 0.225 ± 0.281 | 0.00930 ± 0.0897 |
| **IGSR** | $\mathbf{7.51 \times 10^{-7}}$ [$2.40 \times 10^{-5}$] | **0.781 ± 0.368** | $\mathbf{2.90 \times 10^{-5}}$ [0.127] | **0.744 ± 0.394** | **0.344 ± 0.297** | **0.0558 ± 0.180** |

## G.16. Neurosymbolic Baseline Comparison

As discussed in Appendix A.4, IGSR can be viewed as a neurosymbolic system. To position it against a dedicated neurosymbolic regressor, we compare against LIES (Montazerin et al., 2025), a method based on operator-encoded networks with pruning, across all six benchmark datasets (Table 25). IGSR achieves a lower test MSE on *all six* datasets while also being substantially cheaper to run: on the high-dimensional RNA Polymerase dataset, IGSR's local model fitting and search averaged $137.0 \pm 6.8$ seconds, versus $5106.5 \pm 8.3$ seconds for LIES. This indicates that, on the tasks studied here, the more flexible neurosymbolic parameterization does not yield an accuracy advantage over IGSR's sparse linear-in-basis form, while the latter preserves full equation-level interpretability.

## G.17. Extended SINDy Baseline Comparison

The SINDy baseline reported in Table 2a uses the polynomial-feature library of its original configuration (derived from Holt et al. 2024). To ensure parity in expressive capacity with IGSR and the other baselines, we additionally evaluate

*Table 25.* **Comparison against Neurosymbolic Method.** We compare IGSR against LIES ([Montazerin et al.](), [2025]()) across all six benchmark datasets. IGSR consistently achieves superior predictive accuracy, Test MSE (mean±95 % CI). Furthermore, IGSR maintains a highly efficient local computational footprint: for example, on the RNAPol dataset, its local model fitting and search averaged just $137.0 \pm 6.8$ seconds of wall-clock runtime, compared to $5106.5 \pm 8.3$ seconds for LIES. **Abbreviations and configuration:** As per the main benchmark experiment (Table [2a]()).

| Method | LC MSE ↓ | LC-C MSE ↓ | LC-CR MSE ↓ | C-19 MSE ↓ | RNAPol MSE ↓ | Warf MSE ↓ |
|---|---|---|---|---|---|---|
| LIES | 5.32±2.84 | 3.66±0.977 | 5.66±0.850 | 8.75e-5±1.10e-5 | 0.0143±1.06e-4 | 3.39±2.30 |
| **IGSR** | **5.64e-5±6.79e-5** | **0.0013±0.0010** | **0.0141±0.0087** | **5.01e-8±1.78e-9** | **0.0111±0.0004** | **0.565±0.113** |

SINDy with an augmented functional library that includes non-linear and transcendental operators (sin, cos, tan, exp, log, $\sqrt{\cdot}$, and $|\cdot|$). As shown in Table [26](), this augmentation substantially improves SINDy on the Cancer PKPD and Warfarin datasets, but IGSR still maintains superior predictive accuracy across all benchmarks; SINDy remains unable to run on the high-dimensional RNA Polymerase dataset due to combinatorial explosion of its candidate library.

*Table 26.* **Extended SINDy Baseline Evaluation.** We evaluate SINDy with an augmented functional library to ensure parity in expressive capacity with IGSR and other baselines. The original SINDy configuration (derived from [Holt et al. 2024]()) relied on polynomial features only. Here, we extend it to include non-linear and transcendental operators (sin, cos, tan, exp, log, $\sqrt{\cdot}$, and $|\cdot|$). While this augmentation improves SINDy's performance on the Cancer PKPD and Warfarin datasets, IGSR maintains superior predictive accuracy across all benchmarks. **Abbreviations and configuration:** As per the main benchmark experiment (Table [2a]()).

| Method | LC MSE ↓ | LC-C MSE ↓ | LC-CR MSE ↓ | C-19 MSE ↓ | RNAPol MSE ↓ | Warf MSE ↓ |
|---|---|---|---|---|---|---|
| SINDy (Original, Polynomial) | 335±3.14 | 0.838±6.35e-3 | 0.664±5.14e-3 | 1.43e-7±4.48e-9 | — [†] | 0.914±0.169 |
| SINDy (Extended Library) | 3.74±0.103 | 0.0374±3.10e-4 | 0.0429±8.02e-4 | 1.58e-4±2.66e-7 | — [†] | 0.723±0.112 |
| **IGSR** | **5.64e-5±6.79e-5** | **0.0013±0.0010** | **0.0141±0.0087** | **5.01e-8±1.78e-9** | **0.0111±0.0004** | **0.565±0.113** |

[†] Unable to run due to combinatorial explosion from the high feature count ($d = 263$).

## G.18. Investigation of Proposed Terms Count

The `terms_per_round` hyperparameter controls the breadth of the search during the term generation phase. It dictates how many new candidate basis functions the "Propose" LLM agent suggests in each iteration of the propose-and-prune cycle. A higher number implies a wider exploration of the local equation space per step, potentially increasing the likelihood of finding correct terms, but also increasing the cognitive load on the LLM (context length) and the computational cost of evaluation.

To quantify this trade-off and justify our default configuration, we evaluated IGSR on the *Lung Cancer (with Chemo. & Radio.)* dataset while varying `terms_per_round` across the set $\{1, 3, 5, 10, 20\}$. All other hyperparameters were maintained at their default values (including a maximum sparsity of 6 terms). We report the Normalized Mean Squared Error (NMSE, defined and implemented as per [Shojaee et al., 2025b]()) on the test set, along with computational costs (Total Tokens and Wall Clock time), averaged over 25 independent seeds.

*Table 27.* **Impact of `terms_per_round` on Performance and Cost.** Evaluation on the *Lung Cancer (with Chemo. & Radio.)* dataset. Results are mean ± 95% confidence intervals over 25 seeds. Increasing the number of proposed terms improves predictive accuracy (lower NMSE) up to a saturation point, at the expense of increased computational cost. The default setting of 5 represents an efficient trade-off.

| Terms Per Round | NMSE ↓ | Total Tokens ↓ | Wall Clock (s) ↓ |
|---|---|---|---|
| 1 | $0.00223 \pm 0.000535$ | $271{,}785 \pm 34{,}154$ | $216.0 \pm 28.5$ |
| 3 | $0.00197 \pm 0.000809$ | $299{,}619 \pm 43{,}488$ | $248.4 \pm 33.2$ |
| **5 (Default)** | $\mathbf{0.00107 \pm 0.000483}$ | $\mathbf{303{,}615 \pm 42{,}967}$ | $\mathbf{280.5 \pm 39.2}$ |
| 10 | $0.000742 \pm 0.000435$ | $324{,}178 \pm 34{,}697$ | $349.9 \pm 41.8$ |
| 20 | $0.000877 \pm 0.000445$ | $343{,}885 \pm 59{,}708$ | $351.1 \pm 51.5$ |

The results are summarized in Table 27. The data reveals a clear trend: increasing the proposal width generally leads to improved model accuracy. Moving from suggesting 1 term to 5 terms per round reduces the NMSE by approximately half (from 0.00223 to 0.00107). This improvement highlights the benefit of providing the "Prune" agent with a sufficiently diverse pool of candidates; the influence-based feedback mechanism ($\Delta_j$) is most effective when the candidate set contains high-quality options to select from.

However, diminishing returns are observed at higher values. While increasing the proposal count to 10 yields the lowest mean NMSE (0.000742), this represents a smaller marginal gain compared to the jump from 1 to 5. Furthermore, proposing 20 terms does not improve performance beyond the 10-term setting, potentially due to the increased difficulty for the LLM to attend to and reason about a larger list of candidates within the prompt context or simply due to saturation of the search space.

In terms of computational efficiency, costs scale with the number of terms. The setting of 10 terms increases the wall-clock time by approximately 25% compared to the setting of 5 terms. The default value of `terms_per_round = 5` was thus selected as the operational "sweet spot" for IGSR. It achieves a substantial performance improvement over lower values (nearly halving the error compared to 3 terms) with only a modest increase in runtime, ensuring the method remains computationally efficient while maintaining high discovery capability. Acknowledging that the optimal configuration is contingent upon specific dataset characteristics and computational constraints, we recommend 5 to 10 terms per round as a robust heuristic for general application.

### G.19. Investigation of Prompt Variations

The reliance of IGSR on Large Language Models raises the question of sensitivity to prompt engineering: to what extent does performance depend on the specific, detailed instructions and context provided in the default prompts? Furthermore, the context window length and associated token costs are practical concerns, particularly when dealing with high-dimensional datasets where providing data previews can consume significant bandwidth.

To address these questions, we investigated the performance of IGSR using a set of *Simplified* prompts. We compared this variant against the *Default* prompt configuration described in Appendix B.7. The simplified prompts differ from the default prompts in the following key aspects:

- **Data Preview Removal:** In the "Propose" phase, the snippet of raw input data and target variable preview was removed. This is the primary driver of context length reduction, particularly for datasets with many features.
- **Instruction Compression:** For both "Propose" and "Prune" agents, verbose instructions and detailed task descriptions were collapsed into concise lists.
- **Pruning Simplification:** The extensive "Input You Receive" and "Your Task" sections, along with the example input/output tables, were removed from the "Prune" agent prompt. Only the core input tables (terms, weights, influence), history, and essential output formatting constraints were retained.

We evaluated both variants on the *Lung Cancer (with Chemo. & Radio.)* dataset (representing complex dynamics) and the *RNA Polymerase* dataset (representing high-dimensional data). Experiments were conducted over 10 random seeds. We report the Normalized Mean Squared Error (NMSE) on the test set, along with the total token consumption and wall-clock time.

The predictive performance results are presented in Table 28. While the *Default* prompts yield the lowest mean NMSE on both datasets, the *Simplified* prompts remain competitive. On the RNA Polymerase dataset, the performance gap is relatively narrow (0.824 vs 0.854), and the confidence intervals overlap significantly. On the Lung Cancer dataset, the *Simplified* variant exhibits higher variance and a higher mean error (0.0219 vs 0.0013), suggesting that for complex dynamical systems, the detailed instructions and examples in the *Default* prompt help stabilize the LLM's reasoning.

*Table 28.* **Impact of Prompt Variations on Predictive Accuracy.** NMSE (mean $\pm$ 95% CI) on test sets over 10 seeds. *Default* prompts achieve lower mean error, but *Simplified* prompts demonstrate reasonable robustness.

| Dataset | IGSR (Default) | IGSR (Simplified) |
|---|---|---|
| Lung Cancer (with Chemo. & Radio.) | **0.00127** $\pm$ 0.00087 | 0.0219 $\pm$ 0.0439 |
| RNA Polymerase | **0.824** $\pm$ 0.031 | 0.854 $\pm$ 0.009 |

However, the trade-off becomes distinct when analyzing computational costs, as shown in Table 29. The *Simplified* variant achieves a substantial reduction in token usage. For the high-dimensional RNA Polymerase dataset (263 features), removing the data preview reduced the average token consumption from $\approx$1.68 million to $\approx$140,000. This also translated to a wall-clock speedup of approximately $2.5\times$.

*Table 29.* **Computational Cost of Prompt Variations.** Comparison of total token usage and wall-clock time (seconds) averaged over 10 seeds. The *Simplified* prompts offer significant efficiency gains, particularly for high-dimensional data (RNA Polymerase).

| Dataset | Metric | IGSR (Default) | IGSR (Simplified) |
|---|---|---|---|
| Lung Cancer (with Chemo. & Radio.) | Total Tokens | $291,540 \pm 45,848$ | $116,301 \pm 19,301$ |
| | Wall Clock (s) | $271.8 \pm 39.9$ | $153.7 \pm 16.0$ |
| RNA Polymerase | Total Tokens | $1,675,638 \pm 151,933$ | $139,835 \pm 8,714$ |
| | Wall Clock (s) | $296.5 \pm 28.2$ | $112.0 \pm 11.6$ |

These findings suggest that IGSR is reasonably robust to variations in prompt detail. While detailed instructions and examples (as in the *Default* configuration) maximize discovery accuracy and stability, the core efficacy of the method is driven by the influence-based feedback mechanism rather than prompt engineering alone. For resource-constrained scenarios or high-dimensional datasets, simplified prompts offer a viable, highly efficient alternative that retains the majority of the method's performance capabilities.

### G.20. Term Candidate Diversity in Search

To quantify the exploration capability of the IGSR framework, specifically within the stochastic term generation phase of the Monte Carlo Tree Search (MCTS), we analyzed the structural diversity of the candidate basis functions proposed by the LLM. High diversity among candidate terms implies that the search is effectively exploring the hypothesis space rather than collapsing into a narrow mode of similar expressions.

**Methodology.** We define a *Diversity Index* based on the symbolic composition of the proposed terms. For a given candidate term $t$, we parse its abstract syntax tree to extract a "bag-of-symbols" set $S(t)$, comprising all unique variables, operators, functions, and constants utilized in the expression (e.g., $S(3x + \sin(y)) = \{3, x, y, +, \sin\}$).

For a set of unique candidate terms $\mathcal{T}$ generated at a specific search node, we compute the mean pairwise Jaccard similarity $\bar{s}$:

$$\bar{s} = \frac{1}{|\mathcal{P}|} \sum_{(t_i, t_j) \in \mathcal{P}} \frac{|S(t_i) \cap S(t_j)|}{|S(t_i) \cup S(t_j)|}$$

where $\mathcal{P}$ is the set of all unique pairs in $\mathcal{T}$. The *Diversity Index* is defined as $D = 1 - \bar{s}$. A score of $1.0$ indicates disjoint symbol sets (maximum diversity), while $0.0$ indicates identical symbol usage. We also track the *Distinct Ratio*, defined as the number of unique symbolic strings divided by the total number of proposed terms, to measure repetition.

**Results.** We evaluated this metric across all expansion points in the MCTS for two representative dataset: *Lung Cancer (with Chemo. & Radio.)*, low-dimensional, 4 state/action variables, and *RNA Polymerase*, high-dimensional, 263 features. All default IGSR (MCTS variant) hyperparameters were used, with early stopping disabled to obtain a larger pool on node expansions. The results are summarized in Table 30.

*Table 30.* **Candidate Term Diversity Statistics.** Metrics are averaged across all MCTS expansion nodes (mean $\pm$ 95% CI). The *Diversity Index* measures structural variation ($1 -$ Jaccard), while *Distinct Ratio* measures string uniqueness.

| Dataset | Diversity Index ($D$) | Distinct Ratio |
|---|---|---|
| Lung Cancer (with Chemo. & Radio.) | $0.812 \pm 0.020$ | $0.683 \pm 0.088$ |
| RNA Polymerase | $0.987 \pm 0.021$ | $0.384 \pm 0.060$ |

**Analysis.** The *Lung Cancer* dataset exhibits a robust diversity index of approximately $0.81$. Given the small number of available variables in this problem domain, some overlap in symbol usage (e.g., repeated use of the variable `cancer_volume`) is structurally inevitable; essentially, terms must share variables to be relevant. Despite this constraint, the high index confirms that the operators and functional transformations proposed are varied.

In contrast, the *RNA Polymerase* dataset achieves a near-maximal diversity index of $\approx 0.99$. This higher diversity is expected and desirable: the dataset contains 263 input features, allowing the LLM to propose candidate terms involving disjoint subsets of features (e.g., different nucleotide positions or histone markers) with very little symbolic overlap.

These results indicate that the stochastic sampling parameters used in the MCTS induce a high degree of structural variety in the candidate generation phase, ensuring broad coverage of the search space.

# H. Computational and Scalability Analysis

To address the practical feasibility of IGSR, we conducted a comprehensive analysis of its computational cost and scalability.

## H.1. Cost and Wall-Clock Time Comparison

We benchmarked IGSR's monetary cost (in USD, based on LLM API calls) and total wall-clock time against several baseline methods on the Lung Cancer (with Chemo. & Radio.) dataset. Cost is based on GPT-4o API pricing at the time of experimentation: $0.000005 per input token and $0.00002 per output token. The total cost and wall clock results, averaged over 5 seeds, are shown in Table 31, as well as the Test MSE taken from the Table 2a main experiment results where available. Hyperparameters used were the same as in the Table 2a experiment.

IGSR's runtime is competitive with other modern symbolic regression methods and significantly faster than computationally intensive approaches like LaSR, while achieving a substantially lower Test MSE. For results under a comparable LLM token budget, see Appendix H.3, which makes the per-token performance efficiency of IGSR especially clear.

*Table 31.* **Computational Cost and Performance Comparison.** Benchmarks on the Lung Cancer (with Chemo. & Radio.) dataset. Cost is based on GPT-4o API pricing at the time of experimentation.

| Method | Total Cost (USD) ↓ | Total Tokens ↓ | Wall Clock (s) ↓ | LLM Calls Wall Clock (s) ↓ | LLM Call Count ↓ | Test MSE ↓ |
|---|---|---|---|---|---|---|
| GPLearn | - | - | $92.2 \pm 5.8$ | - | - | $46.8 \pm 4.9$ |
| PySR (Cramner, 2023) | - | - | $282.7 \pm 6.0$ | - | - | $0.399 \pm 0.123$ |
| D3-white-box (Holt et al., 2024) | $0.589 \pm 0.138$ | $262,636 \pm 2,406$ | $203.4 \pm 39.5$ | $178.6 \pm 23.8$ | $23.4 \pm 6.2$ | $253 \pm 273$ |
| LaSR (Grayeli et al., 2024) | $72.40 \pm 3.48$ | $5,171,391 \pm 238,935$ | $1444.5 \pm 137.5$ | $-$ [†] | $-$ [†] | $3.97 \pm 3.21$ |
| ICSR (Merler et al., 2024) | $1.08 \pm 0.16$ | $261,989 \pm 71,813$ | $774.2 \pm 140.6$ | $392.6 \pm 117.5$ | $80.0 \pm 0.0$ | $6.1 \pm 1.05$ |
| LLM-SR (Shojaee et al., 2025a) | $0.80 \pm 0.02$ | $61,905 \pm 2,220$ | $371.7 \pm 18.7$ | $355.1 \pm 19.2$ | $32.0 \pm 0.0$ | $32.1 \pm 48.4$ |
| **IGSR** | $1.87 \pm 0.71$ | $272,743 \pm 111,859$ | $382.8 \pm 137.6$ | $376.8 \pm 135.2$ | $93.4 \pm 17.7$ | $\mathbf{0.0141 \pm 0.0087}$ |

[†] Due to implementational difficulties, LaSR being implemented in Python *via Julia*, while we were able to obtain *"Total Tokens"* from the logs, we were unable to measure *"LLM Call Wall Clock (s)"* or *"LLM Call Count"* for this method.

## H.2. Peak Memory Usage

Because IGSR runs its linear-algebra computations (model fitting and influence evaluation) on CPU and can expand several MCTS branches, a natural question is how peak memory scales on the largest problem. We profiled the peak resident set size (peak RSS, via /usr/bin/time) of a full IGSR run on the RNA Polymerase dataset (263 features) – the most demanding of our benchmarks – over 5 seeds. The peak RSS was $2.782 \pm 0.003$ GB (over 5 seeds), i.e. consistently under 3 GB and well within the memory of a commodity workstation. The additional search memory grows with the number of concurrent branches and candidate terms rather than with the raw feature count: in practice the active candidate set remains small (about 11 terms before pruning, 6 after), so memory does not explode with input dimensionality. This confirms that, alongside the sub-linear runtime scaling of Figure 3, memory usage is not a practical bottleneck for IGSR even in the high-dimensional regime.

## H.3. Performance Under a Fixed Computational Budget

To provide a direct and fair comparison of computational efficiency, we conducted an experiment where all LLM-based baselines were benchmarked under a fixed computational budget with respect to LLM usage – as the LLM calls are the most computationally intensive aspect of these methods. In order to ensure the methods consumed a nearly identical number of total LLM tokens (approximately 265k) we either adjusted the hyperparameters controlling itrative execution length of the method (number of iterations / generations or equivalent), or added early stopping based on the cumulative number of tokens consumed in the experiment run. All other hyperparameters were unchanged from the Table 2a main experiment setting. The Lung Cancer (with Chemo. & Radio.) dataset was used and the results were averaged over 5 seeds.

The results, presented in Table 32, are conclusive. When the LLM computational resources are held equal, IGSR's performance advantage is clear. It achieves an MSE that is nearly 50 times better than the next-best performing baseline method (ICSR). This demonstrates that IGSR's architecture uses its computational budget more efficiently to find a superior solutions. While some simpler methods (like ICL, with Basic Feedback) have a faster wall-clock time, they produce substantially less accurate results. IGSR strikes an optimal balance, achieving a competitive runtime while delivering superior final equation(s).

*Table 32.* **Performance Comparison Under a Fixed Token Budget ($\approx$265k Tokens).** Results on the Lung Cancer (with Chemo. & Radio.) dataset, averaged over 5 seeds.

| Method | Test MSE ↓ | Total Tokens | Wall Clock (s) | LLM Call Count |
|---|---|---|---|---|
| ICL | $16.04 \pm 22.1$ | $265,199 \pm 8,729$ | $82.4 \pm 3.8$ | $36.5 \pm 1.1$ |
| D3-white-box (Holt et al., 2024) | $809.7 \pm 1754.4$ | $262,636 \pm 2,406$ | $302.1 \pm 33.3$ | $40.2 \pm 0.7$ |
| LaSR (Grayeli et al., 2024) | $101.3 \pm 112.1$ | $272,606 \pm 15,209$ | $409.3 \pm 28.1$ | - [†] |
| ICSR (Merler et al., 2024) | $2.66 \pm 2.11$ | $261,989 \pm 71,813$ | $1031.5 \pm 319.3$ | $146.0 \pm 0.0$ |
| LLM-SR (Shojaee et al., 2025a) | $9.88 \pm 20.1$ | $259,639 \pm 12,820$ | $1208.2 \pm 58.1$ | $116.0 \pm 0.0$ |
| **IGSR** | $\mathbf{0.016 \pm 0.032}$ | $272,743 \pm 111,859$ | $382.8 \pm 137.6$ | $105.5 \pm 3.7$ |

[†] Due to implementational difficulties, LaSR being implemented in Python *via Julia*, while we were able to obtain *"Total Tokens"* from the logs, we were unable to measure *"LLM Call Count"* for this method.

# I. Reproducibility and Settings

## I.1. Software and Data

For the code and datasets (or instructions on obtaining them, according to the dataset usage terms), see the project codebase (Footnote 1).

To ensure reproducibility of the results, we detail the default hyperparameters and configuration settings used for the Influence-Guided Symbolic Regression (IGSR) framework and the baseline methods. Unless otherwise specified in the specific experimental sections, these default values were maintained throughout the experiments reported in this paper.

## I.2. IGSR Default Hyperparameters

The IGSR framework involves parameters governing the LLM interaction, the linear model fitting, and the Monte Carlo Tree Search (MCTS) strategy.

**LLM Agents:**

- **Model Version:** `gpt-4o` (version `2024-11-20`) was used as the backbone for both the "Propose" and "Prune" agents, unless otherwise noted in the LLM sensitivity analysis.
- **Inference parameters:** *Temperature* set to default 1.0 to allow for stochastic diversity in term proposal; all others kept at their OpenAI defaults.

**Term Generation and Pruning (Propose-and-Prune Cycle):**

- `terms_per_round`: 5. The number of new basis functions the LLM is prompted to suggest in standard iterations.
- `first_round_n_candidates`: 10. The number of terms requested in the initial iteration to seed the search.
- `keep_n_terms` ($K$): 6. The maximum number of terms retained after the pruning phase to ensure equation parsimony.
- `linear_model_type`: OLS (Ordinary Least Squares). Used for fitting weights $\mathbf{w}$ and calculating influence scores.
- `influence_calculation`: No-refit. Influence scores $\Delta_j$ are calculated on the validation set without refitting the model for every term removal, ensuring computational efficiency.

**Search Strategy (MCTS):**

- `total_budget`: 30. The total number of node expansions (iterations) allowed before termination.
- `n_successors`: 5. The number of distinct child nodes (variations) generated from a parent node during expansion.
- `exploration_constant` ($c$): $\sqrt{2}$. The constant used in the UCT formula to balance exploration and exploitation.
- `depth_limit`: 10. The maximum depth of the search tree.
- `rollout_is_just_node_reward`: True. We utilize Heuristic MCTS where the node's immediate validation MSE serves as the reward, without performing deep rollouts.

## I.3. Baseline Method Hyperparameters

We compared IGSR against several white-box and black-box baselines. Below are the specific settings used to ensure fair comparison, particularly regarding model complexity (parsimony). All LLM-based methods use `gpt-4o` (version `2024-11-20`), unless stated otherwise in the particular experiment section.

**GPLearn** (Stephens, 2015):

- `population_size`: 1000.
- `generations`: 30.
- `parsimony_coefficient`: 1.0. Selected to penalize complexity and encourage concise equations comparable to IGSR.
- The remaining hyperparameters were set to the default values provided in the `gplearn` library.

**PySR** (Cranmer, 2023):

- `model_selection`: ``best''.
- `niterations`: 40 (default for version 0.19).
- Library version: 0.19, as bundled with the LaSR implementation[9].
- The remaining hyperparameters were set to the default values of the PySR 0.19 package.

**LLM-based Baselines (ZeroShot, ZeroOptim, ICL):**

- `max_terms`: 6. Constrained via prompting to match IGSR's parsimony.
- `iterations`: 30 (for ICL). Matches the IGSR search budget.

**D3 (white-box)** (Holt et al., 2024):

- `max_features`: 6. Constrained to match IGSR's parsimony.
- The remaining hyperparameters were set to the default values as described in (Holt et al., 2024).

**LaSR** (Grayeli et al., 2024):

- `model_selection`: ``best''.
- `maxsize`: 30. Set to allow equations of sufficient complexity (at least as complex as the 6-term limit used elsewhere).
- `populations`: 15.
- `cycles`: 75 (`niterations` (5) × `populations` (15)).
- The remaining hyperparameters were set to the default values as in LaSR implementation[9] (Grayeli et al., 2024).

**LLM-SR** (Shojaee et al., 2025a):

- `max_nparams`: 8.
- `global_max_sample_num`: 30.
- The remaining hyperparameters were set to the default values as described in (Shojaee et al., 2025a).

**ICSR** (Merler et al., 2024):

- `max_nodes`: 30. Set to allow equations of sufficient complexity.
- `iterations`: 30.
- The remaining hyperparameters were set to the default values as described in (Merler et al., 2024).

**DyNODE, SINDy, RNN, Transformer**:

---

[9]https://github.com/trishullab/LibraryAugmentedSymbolicRegression.jl

- Implementation and hyperparameters follow (Holt et al., 2024).

**XGBoost**:

- All default hyperparameters of the `xgboost` library were used.

**SyMANTIC** (Muthyala et al., 2025):

- Operator set $\{+, -, \times, \div, \exp, \ln, \sin, \cos, (\cdot)^2, \sqrt{\cdot}\}$, generally matching other baselines.
- SIS screening retained the top $20$ features per step, with feature-space expansion to depth $2$ and mutual-information pre-screening.
- Model dimension (`n_term`): 3; higher values were computationally prohibitive on these datasets.
- The remaining hyperparameters were set to the package defaults.

**CAAFE** (Hollmann et al., 2023):

- `iterations`: 30 (LLM feature-generation rounds; matched to IGSR's budget, above the CAAFE default of 10).
- Downstream estimator: Ridge regression, retaining the top $6$ engineered features by coefficient magnitude to match IGSR's parsimony.
- `temperature`: 0.5.
- The remaining hyperparameters were set to the CAAFE defaults.

**LIES** (Montazerin et al., 2025):

- Per-neuron operators: $\{\text{identity}, \exp, \sin, \log\}$; the number of hidden layers was auto-set to $\min(d, 10)$.
- Training: learning rate $0.015$, batch size $128$, with two-phase ADMM sparsification and gradient-based pruning.
- `n_trials`: 3 (best by validation MSE retained).
- The remaining hyperparameters follow the reference LIES implementation.

For any additional hyperparameters or configuration, please refer to the codebase (Footnote 1).

### I.4. Other Notes on Experimental Setup

Finally, please note below any additional details on the experimental setup:

- **Multi-target Regression:** For baseline methods that do not natively support multi-target regression, a separate independent model was instantiated and fitted for each target variable.
- **Model Selection and Data Splits:** Where a method supports early stopping, the validation set was utilized to determine the optimal stopping point to prevent overfitting. For detailed information regarding the specific breakdown of training, validation, and test sets for each benchmark dataset, please refer to Appendix C.

### I.5. Computational Resources

The computational experiments for this research were conducted using a combination of cloud-based services for Large Language Model (LLM) inference and local or server-based machines for model optimization and equation evaluation.

**LLM Inference:** The inference for the majority of the Large Language Models employed in this study was performed using the serverless API provided by Azure AI Foundry. This allowed for scalable access to various proprietary LLM endpoints.

For the `Llama-3.3-70B` model, inference was conducted on a dedicated Azure Virtual Machine, specifically a Standard_NC96ads_A100_v4 instance. Key specifications of this VM include:

- **Processor:** 96 non-multithreaded 3rd Gen AMD EPYC™ 7V13 (Milan) cores.
- **GPU Accelerators:** 4 NVIDIA A100 PCIe GPUs, each with 80GB of memory.
- **System Memory:** 880 GiB.

**Model Optimization and Equation Evaluation:** The optimization of model parameters (i.e., fitting the linear model $\mathbf{w}$ for basis functions $\psi_j(\mathbf{x})$) and the evaluation of equations were carried out on the following types of local workstations and servers:

- A workstation equipped with a 10-core Intel Core i9-10900K CPU and 64 GiB of RAM.
- A server equipped with an 80-Core AMD EPYC 9V84 CPU and approximately 630 GiB of RAM.

It is important to note that the optimization step of IGSR, which involves fitting a linear model (we use the `scikit-learn` library's implementation), does not require GPU acceleration and can be efficiently performed on any reasonably powerful desktop computer.

**Execution Time:** The time of execution for IGSR varied depending on several factors, including the complexity of the dataset, the number of terms explored, the depth and breadth of the Monte Carlo Tree Search (if utilized), and the response latency of the LLM APIs. Individual propose-and-prune cycles involving LLM calls typically took seconds to minutes, while full MCTS runs could extend to several hours for comprehensive exploration. No single run took longer than 3 hours total wall-clock time. Fitting the linear models for equation evaluation was generally swift, on the order of seconds.

The total compute resources for the entire research project, including preliminary experiments and hyperparameter tuning, naturally exceeded that of the final reported experimental runs. However, the resources outlined above are representative of those required to reproduce the main findings.

