# OpenReview forum: "Influence-Guided Symbolic Regression: Scientific Discovery via LLM-Driven Equation Search with Granular Feedback"
_ICML.cc/2026/Conference — ICML 2026 regular_

### Official Review · Reviewer_XQ7n · 2026-03-09

**Soundness:** 3
**Presentation:** 3
**Significance:** 3
**Originality:** 3
**Overall Recommendation:** 4
**Confidence:** 4

**Summary:**

This paper proposes Influence-Guided Symbolic Regression (IGSR), an LLM-driven framework for discovering sparse, interpretable equations of the form f(x) = Σ wj ψj(x). An LLM proposes candidate basis functions, which are evaluated using per-term influence scores Δj that quantify each term’s marginal contribution to validation performance; a pruning step retains top-influence terms. Embedding the propose-and-prune cycle in a Monte Carlo Tree Search (MCTS) balances exploration and exploitation to navigate the combinatorial space. The authors report strong performance on six biological/clinical datasets, leading ranks on LLM-SRBench across predictive and structural metrics, ablations showing the importance of influence feedback and MCTS, and a case study culminating in a wet-lab experiment that supports an IGSR-derived biological hypothesis.

**Compliance With Llm Reviewing Policy:**

Affirmed.

**Key Questions For Authors:**

1. How exactly is $\Delta j$ computed and aggregated across iterations? Please clarify the sign convention (the text shows $\Delta j = \text{MSE(full)} - \text{MSE}(w_j \leftarrow 0)$, which is the negative of an "increase") and define $A_j$ (aggregate influence) used for ranking. Do you refit after term removal, or hold other coefficients fixed? What do ablations show for refit-aware vs. no-refit $\Delta j$?

2. How do you mitigate overfitting to the validation set during search? Is there a nested validation or cross-fitting scheme for computing $\Delta j$ and MCTS rewards? Please report any performance drop when adopting a stricter nested protocol.

3. What is the operator set and complexity allowed for $\psi_j$? How do you prevent pathological terms (e.g., singularities, invalid domains) during proposal and evaluation? Are there dimensional-consistency or unit-aware constraints?

4. Can you add baselines that combine non-LLM library expansion with sparse selection (e.g., LASSO/OMP/SISSO/SyMANTIC) and report results on at least one of your datasets and a subset of LLM-SRBench? This would help isolate the value of LLM proposals versus the influence-pruning/MCTS machinery.

5. For the wet-lab validation, could you summarize sample sizes, effect sizes, statistical tests, and controls in the main text? How sensitive is the discovered relationship to alternative feature sets or to different sequencing replicates?

6. What is the typical $K$ (sparsity) used in pruning, and how is it selected? Do you consider an explicit complexity penalty during MCTS (multi-objective search) or only post hoc model selection on a Pareto frontier?

7. How robust is IGSR to multicollinearity among proposed terms and to redundant proposals across MCTS branches? Do you employ grouping, decorrelation, or stability-selection mechanisms to avoid spurious retention?

**Limitations:**

**Technical limitations**

- Influence definition and refitting: $\Delta j$ is described as MSE(full) – MSE($w_j \leftarrow 0$) while “increase on removal” would be the opposite sign; the notation and sign convention are inconsistent. More importantly, computing $\Delta j$ without refitting can be biased under collinearity; leave-one-term-out with refitting or cross-fitting would be more principled.
- Potential overfitting to validation: both pruning decisions and MCTS rewards use the same validation set repeatedly, which risks search overfitting. No explicit nested validation or cross-fitting protocol is described in the main text.
- The linear-in-basis model class $f(x) = \sum w_j \psi_j(x)$ limits nested/nonlinear compositions (though $\psi_j$ can be complex). This is a known trade-off but should be quantified against methods that learn deeper compositions.

**Experimental gaps/issues**

- Missing strong white-box baselines for sparse model selection over expanded libraries, such as LASSO/Elastic Net, orthogonal matching pursuit, SISSO/C2-SISSO, or recent pipelines like SyMANTIC (C2-SISSO with adaptive expansion) that are purpose-built for interpretable SR. Comparisons against AFE+linear selection are discussed conceptually but not empirically.
- No head-to-head vs. neural SR methods like LIES (operator-encoded networks with pruning) on overlapping benchmarks; these are competitive in symbolic recovery and would contextualize IGSR’s strengths and limits.
- For dynamical tasks, the fit between SINDy-style baselines (which assume derivative libraries) and the paper’s setup is unclear; ensuring apples-to-apples configurations is important.

**Clarity and presentation issues**

- Inconsistent notation between $\Delta j$ and $A_j$ (aggregate influence), with $A_j$ referenced but not defined in the main text; details deferred to appendices should be briefly summarized to avoid ambiguity.
- Precise algorithmic specification of influence variants, pruning thresholds ($K$), and how history is summarized for prompting are largely in appendices; core choices should be in the main paper.

**Missing related work or comparisons**

- Limited discussion of leave-one-feature-out importance (LOFO), SAGE, and model-class-reliance (Fisher et al., 2019) as principled per-feature importance; more explicit connections and ablations versus refit-aware LOFO would strengthen the case.
- Recent scalable SR pipelines (e.g., SyMANTIC’s C2-SISSO; parameter-aware SINDy variants for robust term selection) are discussed tangentially but not directly compared.

**Strengths And Weaknesses:**

**Technical novelty and innovation**

- Introduces granular per-term influence scoring as a feedback signal for LLM-guided equation discovery, moving beyond scalar loss guidance and enabling fine-grained credit assignment.
- Decouples generative proposal (LLM) from deterministic, data-driven selection via influence-based pruning; clean separation reduces hallucination risk and enforces rigor.
- Integrates the propose-and-prune loop into MCTS to manage combinatorial exploration with an explicit exploitation signal, a thoughtful design choice for search over equation structures.
- Provides an agentic variant to incorporate domain knowledge in pruning, which is a pragmatic addition for scientific workflows.

**Experimental rigor and validation**

- Broad empirical suite: six domain-diverse datasets, plus large-scale evaluation on LLM-SRBench with a fixed token budget, and thorough ablations isolating the contributions of influence feedback and MCTS.
- Sensitivity and robustness analyses (LLM backbones, high-dimensional scaling, hyperparameters) and runtime scaling experiments add credibility.
- The RNA Polymerase II case study bridges to wet-lab validation, demonstrating real-world scientific utility beyond benchmark wins.

**Clarity of presentation**

- High-level workflow and search strategy are clearly explained; Figure 1 effectively communicates the MCTS-embedded propose-and-prune pipeline.
- Positioning relative to symbolic regression, LLM-based discovery, and automated feature engineering is reasonably articulated.

**Significance of contributions**

- Addresses a central pain point in LLM-SR—coarse feedback without component-level accountability—by providing tractable, interpretable term-level signals.
- If reproduced, the results and the biological validation would have impact for neurosymbolic scientific discovery, where interpretability and generalization are critical.

---

> ### Author Rebuttal · Authors · 2026-03-30
>
> Thank you for your thorough comments on notation/soundness, stronger baselines, and biological validation, these prompted us to strengthen the paper by adding new analyses (**Tables S.1-S.5**, **Figure S.6**), detailed in full in the anonymous supplementary figures/tables at https://pdfhost.io/v/NMx8D9UWRA_figs, and discussed below.
>
> ### 1. Influence definition, defaults, and validation reuse.
>
> We agree the notation should be clearer. Default pruning uses the no-refit validation change $\Delta_{j,m}=MSE_{-j,m}-MSE_{full,m}$ with others fixed; positive $\Delta$ meaning removal *hurts*. For multi-output pruning, $\Delta_j^{agg}=\max_m \Delta_{j,m}$, with no across-iteration aggregation. *Appendix G.10* shows no-refit, full-refit, and efficient-refit are statistically indistinguishable on all six datasets (all $p\ge0.09$), so we keep the no-refit default. Complexity is controlled by top-$K$ pruning ($K=6$); default search is heuristic MCTS.
>
> To address the validation-overfitting concern directly, **Table S.1** splits validation into disjoint *Val-inner* (pruning) and *Val-outer* (MCTS reward). We find no significant test-MSE drop: LC-CR $0.0142\pm0.0087 \to 0.0159\pm0.0105$ ($p=0.677$) and RNAPol $0.0113\pm0.0004 \to 0.0115\pm0.0006$ ($p=0.399$; Wilcoxon also $p>0.05$). *App. G.9* shows generalization on an independent RNAPol replicate (MSE $0.00793\pm0.000214$, $R^2=0.142\pm0.0232$).
>
> ### 2. Operator constraints and SINDy fairness.
>
> We agree the implementation safeguards should be surfaced more explicitly. The system is practical rather than rule-based: we do not impose formal unit-aware constraints, but candidate terms are evaluated and `nan`/`inf` failures are fed back to the LLM to handle pathological cases.
>
> On SINDy fairness, **Table S.2** adds $\sin,\cos,\tan,\exp,\log,\sqrt{\cdot},|\cdot|$. This improves SINDy (e.g. LC-CR $0.664\to0.0429$), but IGSR remains better on all six datasets (e.g. LC-CR $0.0141$, Warfarin $0.565$). For library-based baselines we also supplied standard nonlinear operators `exp`, `log`, `sin`, and `cos` (with `sqrt` / powers where needed).
>
> ### 3. Stronger baselines and model-class scope.
>
> We added three direct comparisons:
>
> | Comparison | Supp. | Main takeaway |
> | --- | --- | --- |
> | AFE / sparse baselines | **S.4.1** | IGSR best mean MSE on 5/6 datasets; RNAPol near CAAFE |
> | AutoFeat on LLM-SRBench | **S.4.2** | Better OOD and symbolic metrics |
> | LIES | **S.5** | Lower MSE on all six datasets; faster on RNAPol |
>
> In **S.4.1**, RNAPol is the only near-tie: IGSR $0.0111\pm0.0004$ vs CAAFE $0.0107\pm0.0002$ (CIs overlap); AutoFeat/OpenFE time out, and SyMANTIC exceeds memory. In **S.4.2**, IGSR improves OOD NMSE $0.475\to2.90\times10^{-5}$ and Symbolic Accuracy $0.00930\to0.0558$. In **S.5**, IGSR also beats LIES on all six datasets; on RNAPol, MSE is $0.0111$ vs $0.0143$ and runtime $137.0\pm6.8$s << $5106.5\pm8.3$s. We agree the class remains linear in the final readout, but it is already competitive on the tasks studied.
>
> ### 4. Wet-lab detail and robustness of the methylation finding.
>
> We agree the wet-lab essentials should be surfaced more directly. The setup was HCT116, 1 mM 5-azacytidine, 3-day treatment + 2-day recovery, DMSO control, eNET-seq, two biological replicates, and about 3000 gene-body pauses per condition. Hypomethylation increased both pause frequency and strength (Wilcoxon $p=4.3\times10^{-52}$ and $3.5\times10^{-33}$), consistent with the IGSR hypothesis that gene-body methylation suppresses pausing.
>
> To address robustness directly, **Table S.3** shows that the methylation hypothesis is recovered in 7/10 seeds on the original replicate, 6/10 on an independent replicate, and 8/10 under random 100-feature subsets. This complements *App. G.9*, where equations trained/selected only on Replicate 1 still generalize best to Replicate 2 without refitting.
>
> ### 5. Multicollinearity, redundancy, and related work.
>
> We agree that no explicit grouping/decorrelation/stability-selection module is implemented. Our answer is empirical: *App. G.10* shows no held-out accuracy loss from the no-refit default, suggesting that collinearity is not materially harming final held-out performance. *App. G.15* also shows high proposal diversity across MCTS branches ($0.812\pm0.020$ on LC-CR and $0.987\pm0.021$ on RNAPol). **Fig. S.6** complements this with final-equation stability: after canonicalization, only 11 unique terms remain across 25 LC-CR seeds; 12/25 seeds recover the same 6 ground-truth terms, and 86/300 seed pairs are identical. In *App. A.5* we position IGSR relative to LOFO / Fisher et al. (2019); in the camera-ready we will clarify this and include SAGE. We view $\Delta_j$ as leave-one-$term$-out importance over discovered basis functions, rather than raw features.
>
> We hope these clarifications and new analyses address your concerns, and that you would consider updating your score if so. We thank you for the detailed review and for helping us strengthen the paper.

---

> > ### Author Rebuttal · Reviewer_XQ7n · 2026-04-03
> >
> > The authors have addressed my concerns.

---

> > > ### Author Response · Authors · 2026-04-04
> > >
> > > Dear Reviewer XQ7n,
> > >
> > > Thank you for your rigorous evaluation and for confirming that our additional analyses resolved your concerns. Your constructive feedback regarding our baselines and validation has significantly strengthened our paper. If you deem it appropriate, we would be grateful if you considered updating your score to reflect this resolution.
> > >
> > > Thank you again for your valuable guidance.
> > >
> > > ---
> > >
> > > The previously shared PDF supplementary document appears to have expired, and we have anonymously reuploaded it at:
> > > https://figs1.hostmora.app/ (Supplementary Figures S.1-S.6).

---

### Official Review · Reviewer_6bP6 · 2026-03-12

**Soundness:** 4
**Presentation:** 3
**Significance:** 4
**Originality:** 4
**Overall Recommendation:** 5
**Confidence:** 4

**Summary:**

This paper proposes a two-step process, using per-term influence scores to guide LLM-driven search of scientific equation using symbolic search with MCTS. Current LLM-based approaches to symbolic regression typically rely on coarse, global feedback such as overall mean squared error, which makes it difficult to identify which specific terms are driving prediction errors. Influence-Guided Symbolic Regression (IGSR) instead treats equation discovery as an iterative propose-and-prune process. First, the LLM generates candidate basis functions. Each candidate term is then evaluated using a per-term influence score that measures its marginal contribution to generalization. Terms with low influence are pruned, and the cycle repeats within a Monte Carlo Tree Search framework. The experimental validation is broad and well structured across 6 benchmarks. The wet-lab validation corroborates the results.

**Compliance With Llm Reviewing Policy:**

Affirmed.

**Final Justification:**

Overall, this is an interesting and well-executed paper that provides clear experimental validation of its proposed methods. Furthermore, the authors submitted a thoughtful and compelling rebuttal that effectively addressed any initial concerns. I think this paper should be accepted.

**Key Questions For Authors:**

- Have you experimented with non-linear models? How would you adapt the IGSR framework when interpretability can only be partial?

- How would you deal with confounding factors in a non-linear model?

- MCTS is an interesting search mechanism. However, your methodology is also amenable to classic probabilistic inference approaches like MCMC and SCM. Did you think about exploring such approaches with LLM and search?

**Limitations:**

yes

**Strengths And Weaknesses:**

**Strengths**:

- The core idea is well grounded and convincingly motivated. An interpretable, modular influence function is especially valuable in domains such as medicine.

- The experimental evaluation is impressive, covering multiple models and benchmarks.

- The ablation studies clearly isolate the contributions of both the influence score and MCTS.

- The wet-lab validation is a major strength and is uncommon in machine learning papers.

---------

**Weaknesses**:

- The notation is dense and, in my view, more complicated than necessary.

- The quality of the LLM remains a bottleneck for discovery, even if MCTS partially mitigates this limitation.

---

> ### Author Rebuttal · Authors · 2026-03-30
>
> Thank you for the thoughtful review. Your questions on notation, intended scope, and alternative search strategies helped us further develop the paper. To address them, we summarize the nonlinear-extension analyses included in the appendix and, in addition, have now added a comparison with a nonlinear nerusymbolic model. Throughout, labels beginning with "**S.**" refer to the additional supplementary figures/tables PDF shared anonymously at https://pdfhost.io/v/NMx8D9UWRA_figs, especially **Table S.5** below.
>
> ### 1. The notation can be stated more simply.
>
> We agree. The default workflow is: (i) the LLM proposes 5 candidate terms, (ii) we fit the linear readout, (iii) we compute each term's validation-set no-refit influence $\Delta_j=MSE_{-j}-MSE_{full}$, (iv) we keep the top $K=6$ terms (for multi-output, rank by $\max_m \Delta_{j,m}$), and (v) we explore these edits with heuristic MCTS using immediate node reward. In the camera-ready, we will make this default workflow and its main settings easier to see in the main text.
>
> ### 2. LLM quality is still a bottleneck.
>
> We agree. Our claim is not that IGSR removes dependence on LLM priors; rather, it uses more informative feedback and search to make better use of each proposal. Appendix G.6 evaluates nine backbones, including open-weight models, and the performance gains from IGSR remain stable. So the benefit appears to come from the architecture rather than from one particular backbone, while better proposal models should still directly improve discovery quality.
>
> ### 3. Nonlinear downstream models and partial interpretability.
>
> We agree this should be discussed more explicitly. Even in the default model, the basis functions can be nonlinear; the main restriction is that the *final readout* is linear in the discovered terms. This means symbolic feature discovery and the final predictor can be decoupled: if one uses a nonlinear downstream model, interpretability becomes *partial* because the discovered $\psi_j(\mathbf{x})$ remain human-readable, but the final combiner is no longer a single sparse closed-form equation.
>
> Three points of analysis below explore this aspect experimentally:
>
> | Setting | What it tests | Main takeaway |
> | --- | --- | --- |
> | IGSR + XGBoost (App. G.12) | Nonlinear predictor on frozen symbolic features | Lung Cancer (with Chemo. & Radio.) NMSE $0.0013 \to 0.0084$; RNAPol $0.791 \to 0.773$: no consistent gain vs linear |
> | LIES (**Table S.5**) | More flexible neurosymbolic baseline | IGSR has lower test MSE on all 6 benchmarks; e.g. RNAPol $0.0111$ vs $0.0143$ |
> | IGSR-TLO (App. G.11) | Richer nonlinear terms via optimized internal constants | This IGSR extension is able to achieve recovery on a *within-term* parameter benchmark in $10/25$ runs |
>
> Thus, on the tasks studied here, the more flexible variants do not show a clear empirical advantage over the linear readout, while the linear form preserves full equation-level interpretability.
>
> ### 4. Confounding requires a causal design choice, not just a nonlinear predictor.
>
> We agree, and this should be separated from the predictive-model discussion more clearly. Changing the final model from linear to nonlinear would still require explicit confounding-control choices. If the goal were causal claims, confounding control would need to be built into the data and objective, e.g. explicit adjustment variables, temporal ordering, and an identification strategy before or alongside symbolic search. The current paper's primary scope is predictive equation discovery rather than causal identification. The Cancer PKPD simulator does include time-dependent confounding in treatment assignment, but we use it as a challenging prediction/equation-discovery benchmark, not as a claim that IGSR alone identifies treatment effects.
>
> ### 5. Why MCTS rather than MCMC-style or other probabilistic search?
>
> We agree these are natural alternatives. We chose MCTS because the search naturally forms a branching tree of propose-and-prune edits, UCT gives a clean exploration/exploitation trade-off, and validation MSE provides a simple node reward without requiring a full posterior over equation space. The default is also lightweight: heuristic MCTS with immediate node reward backpropagation (no deep rollout). This choice matters empirically: in the main ablation on Lung Cancer (with Chemo. & Radio.), removing MCTS while keeping influence feedback degrades NMSE from $7.87\times10^{-4}$ to $0.626$. We therefore chose MCTS as the most direct search mechanism for this paper, while viewing MCMC-style and related probabilistic-search formulations as promising future work, likely with the LLM used as an informed proposal mechanism.
>
> We hope this discussion and the additional **Table S.5** comparison address your questions and concerns. We are sincerely grateful for your thoughtful comments, which helped us strengthen both the presentation and empirical evaluation of our work, and we would be happy to discuss anything further.

---

> > ### Author Rebuttal · Reviewer_6bP6 · 2026-04-01
> >
> > Thank you for the clarifications. I think this work and experimental protocol are relevant and interesting for the community.

---

> > > ### Author Response · Authors · 2026-04-02
> > >
> > > Dear Reviewer 6bP6,
> > >
> > > Thank you for your time and effort, and for confirming that our rebuttal addressed your questions. We are glad to hear that you find the work and experimental protocol relevant and interesting for the community. If you feel it is appropriate, we would appreciate any score update reflecting that your concerns were resolved.
> > >
> > > Thank you again for your constructive review.
> > >
> > > ---
> > >
> > > The previously shared PDF supplementary document appears to have expired, and we have anonymously reuploaded it at:
> > > https://figs1.hostmora.app/ (Supplementary Figures S.1-S.6).

---

### Official Review · Reviewer_FiLv · 2026-03-13

**Soundness:** 2
**Presentation:** 4
**Significance:** 3
**Originality:** 2
**Overall Recommendation:** 4
**Confidence:** 4

**Summary:**

This work proposes IGSR, a framework for symbolic regression that uses LLMs to iteratively propose basis functions for a sparse linear model, guided by per-term influence scores measuring each term's marginal contribution to score. The search is embedded in MCTS to avoid local optima. The method is evaluated on biological and clinical benchmarks, LLM-SRBench, and a genomics case study where a discovered hypothesis was subsequently validated via wet-lab experiment.

**Compliance With Llm Reviewing Policy:**

Affirmed.

**Final Justification:**

The rebuttal addresses most of the concerns; my overall evaluation of the work, however, remains the same, and I keep the score.

**Key Questions For Authors:**

1. On positioning and missing baselines: the method discovers symbolic features for a linear model, which closely overlaps with LLM-based AFE methods (like CAAFE, FeatLLM, and OCTree -- CAAFE in particular has been applied to regression settings). These are discussed in Appendix but not evaluated against. How does IGSR compare to these methods, particularly when paired with a linear downstream model?

2. On the linear model assumption: for problems like tumor growth under combined chemo and radiotherapy, or RNA Pol II pausing, a sparse linear combination of basis functions is a strong structural assumption. What is the method actually recovering in these settings: a true mechanistic model, a useful approximation, or something in between? Also, how stable are the discovered equations across seeds in terms of which specific terms are selected?

**Limitations:**

yes

**Strengths And Weaknesses:**

Strengths:
* The core idea of providing granular influence scores rather than a single global loss signal is interesting
* Ablations are thorough and extensive; the paper is well-written
* Extensive set of benchmark problems and wet-lab experiments

Weaknesses:
* The paper positions itself primarily as symbolic regression, but the method is structurally closer to automated feature engineering for linear models. This affects how the contributions and comparisons should be framed, and the current framing understates the relationship to a large body of relevant AFE literature
* The linear model assumption is a significant structural constraint that is not well-justified for the chosen benchmarks, which involve complex and likely nonlinear dynamics.

---

> ### Author Rebuttal · Authors · 2026-03-30
>
> Thank you for the constructive review. Your questions on positioning, baselines, model scope, and equation stability helped us refine the paper. To address them during rebuttal, we ran substantial new analyses, including an expanded AFE comparison and a dedicated cross-seed stability study. Throughout, labels beginning with "**S.**" refer to the additional supplementary figures/tables PDF, especially **Table S.3**, **Table S.4.1**, **Table S.5**, and **Figure S.6**, shared anonymously at https://pdfhost.io/v/NMx8D9UWRA_figs; these materials will be included in the camera-ready.
>
> ### 1. AFE framing.
>
> We agree IGSR should be framed more explicitly at the symbolic regression / automated feature engineering boundary. It uses AFE-like symbolic transformations, but targets a single sparse closed-form equation rather than a feature set for an arbitrary downstream model, with search guided by per-term influence scores. In the camera-ready, we will use the additional space to surface the discussion of AutoFeat/OpenFE/CAAFE/FeatLLM/OCTree more prominently in the main paper and sharpen this positioning.
>
> ### 2. AFE baseline comparison.
>
> We added a direct post-review AFE comparison in **Table S.4.1**, including the LLM-AFE baseline CAAFE and the non-LLM AFE baseline AutoFeat+LASSO; Appendix G.12 also reports OpenFE with linear and XGBoost downstream models. For the new LLM-AFE head-to-head, we prioritized CAAFE because its published setup maps most directly to this regression setting, whereas FeatLLM and OCTree are presented more as classifier/tabular pipelines.
>
> | Dataset | IGSR | CAAFE | AutoFeat+LASSO |
> | --- | --- | --- | --- |
> | RNA Polymerase | 0.0111 ± 0.0004 | 0.0107 ± 0.0002 | DNF |
> | Lung Cancer (with Chemo.) | **0.0013 ± 0.0010** | 92.5 ± 39.6 | 0.0803 ± 0.0445 |
> | COVID-19 | **5.01e-8 ± 1.78e-9** | 2.32e-6 ± 4.46e-7 | 2.23e-7 ± 5.01e-8 |
>
>
> Against these added AFE baselines, plain IGSR has the best mean MSE on **5/6** datasets; on RNA Polymerase, IGSR and CAAFE are comparable and the 95% CIs overlap.
>
> ### 3. Linear-in-basis model assumption.
>
> We agree the linear readout is a real structural assumption. The model is linear only in the final readout, while the discovered basis functions can themselves be nonlinear (e.g., $x_1x_2$, $x\log x$). Empirically, more flexible downstream models do not show a consistent gain here: in Appendix G.12, the XGBoost variant yields $0.0084$ on Lung Cancer (with Chemo. & Radio.) and $0.773$ on RNA Polymerase, versus $0.0013$ and $0.791$ for the linear readout. In **Table S.5**, IGSR also outperforms a neurosymbolic baseline (LIES) across all six datasets, including **0.0141** vs **5.66** on Lung Cancer (with Chemo. & Radio.) and **0.0111** vs **0.0143** on RNA Polymerase. This supports the linear readout as a reasonable default when the goal is a sparse interpretable equation.
>
> ### 4. What the discovered equations represent.
>
> We agree this should be calibrated more sharply. For the simulated PKPD tasks with known ground-truth dynamics, the natural interpretation is often close to mechanistic recovery, and **Figure S.6** further shows that the discovered term sets concentrate strongly on the true structure. On LLM-SRBench, the two IGSR variants occupy the top two average ranks on both Term Recall and Symbolic Accuracy, supporting better exact structural recovery than the baselines. For RNA Polymerase, by contrast, we think the right framing is interpretable approximation / hypothesis generation rather than exact mechanism recovery. In **Table S.3**, the DNA methylation hypothesis is recovered in **7/10** seeds on the original replicate, **6/10** on an independent replicate, and **8/10** under random 100-feature subsets; the wet-lab follow-up was directionally consistent, with hypomethylation increasing both pause frequency and pause strength ($p=4.3\times 10^{-52}$, $p=3.5\times 10^{-33}$). In the camera-ready, we will make this calibration and these statistics explicit in the manuscript text.
>
> ### 5. Cross-seed stability of selected terms.
>
> We added a dedicated cross-seed analysis in **Figure S.6** on Lung Cancer (with Chemo. & Radio.). After canonicalization, only 11 unique terms remain across 25 seeds; the four most frequent, appearing in 84-100% of runs, are all ground-truth. Moreover, 12/25 seeds recover the same 6 ground-truth terms with mean pairwise Jaccard 0.95, and 86/300 seed pairs yield identical final term sets. Remaining variation is concentrated in near-equivalent Gompertz approximations ($\sqrt{x}$, $\log x$, $x\log x$), while the core interaction terms stay stable.
>
> We hope these clarifications, together with the new AFE baselines and **Figure S.6**, address your concerns on positioning, model assumptions and stability, and that you would consider updating your score. We are grateful for your insightful suggestions that motivated the additional experiments and analyses, and would be very happy to discuss any remaining questions.

---

> > ### Author Rebuttal · Reviewer_FiLv · 2026-04-04
> >
> > Thanks for the detailed rebuttal. While the authors address most of the concerns, my overall evaluation of the work remains the same, and I decided to keep the score.

---

> > > ### Author Response · Authors · 2026-04-05
> > >
> > > Dear Reviewer FiLv,
> > >
> > > Thank you again for the thoughtful discussion, and for indicating that our rebuttal addressed your concerns. Your questions directly motivated the added AFE head-to-heads and the dedicated cross-seed stability analysis, which we believe materially strengthen the empirical support for the paper and better contextualize its contribution.
> > >
> > > We deeply appreciate your constructive feedback throughout this process and hope these additions prove helpful during the final deliberation.
> > >
> > > ---
> > >
> > > The previously shared PDF supplementary document appears to have expired, and we have anonymously reuploaded it at:
> > > https://figs1.hostmora.app/ (Supplementary Figures S.1-S.6).

---

### Official Review · Reviewer_ZJYk · 2026-03-15

**Soundness:** 4
**Presentation:** 4
**Significance:** 3
**Originality:** 3
**Overall Recommendation:** 4
**Confidence:** 5

**Summary:**

The paper presents Influence-Guided Symbolic Regression (IGSR), a framework leveraging Large Language Models (LLMs) for scientific equation discovery. Unlike prior LLM-based SR methods that rely on coarse, global scalar metrics (e.g., MSE) for feedback, IGSR evaluates proposed basis functions using granular, per-term "influence scores". This metric quantifies each term's marginal contribution to generalization accuracy, driving a rigorous propose-and-prune cycle. Integrated with Monte Carlo Tree Search (MCTS), IGSR efficiently navigates the combinatorial space of functional forms. The authors demonstrate strong performance across benchmarks such as LLM-SRBench, pharmacological PKPD models, and an epidemiological simulation. Furthermore, they validate the framework's capacity for genuine discovery through a biological case study, where an IGSR-identified hypothesis regarding DNA methylation and RNA Polymerase II pausing was supported by prospective wet-lab experimentation.

**Compliance With Llm Reviewing Policy:**

Affirmed.

**Key Questions For Authors:**

1. How does the IGSR system detect and filter out mathematically equivalent but syntactically distinct expressions generated by the language model before computing their influence scores?
2. Biological systems are heavily driven by synergistic interactions. If two features only show significant predictive power when combined (similar to genetic epistasis) but have very low individual influence scores, how does the greedy pruning mechanism ensure these synergistic pairs are not prematurely discarded early in the tree search?
3. Since the linear optimization step relies on CPU computations, how does the peak memory usage scale during parallel tree search rollouts when processing high-dimensional datasets?

**Limitations:**

yes

**Strengths And Weaknesses:**

Strengths:
1. The shift from global scalar loss to granular, per-term influence scores for guiding LLM generation and pruning is a highly effective and intuitive contribution.
2. The study features extensive benchmarking against baselines (e.g., PySR, D3, LLM-SR) and notably goes beyond standard in-silico metrics by validating a novel, AI-generated biological hypothesis through prospective wet-lab experiments.
3. Integrating the propose-and-prune cycle into an MCTS framework provides a systematic way to balance exploration and exploitation, which is well-supported by thorough ablation studies showing improvements over a linear iterative refinement approach.
Weaknesses:
1. The core framework fundamentally assumes an additive structure of non-linear basis functions. While the authors briefly explore Term-Local Optimization (IGSR-TLO) in the appendix, the default methodology may struggle to discover deeply nested or recurrent non-linear dynamics compared to fully differentiable or evolutionary approaches.
2. The default computation of influence scores—setting a term's weight to zero without refitting the model—might inaccurately estimate term importance in the presence of highly correlated basis functions, potentially leading to suboptimal pruning.
3. The reliance on iterative LLM API calls within an MCTS tree search raises concerns about computational cost and scalability. Although the linear optimization step is swift, simulating multiple future steps (rollouts) using high-capacity LLMs could be prohibitively expensive for broader adoption.

---

> ### Author Rebuttal · Authors · 2026-03-30
>
> Thank you for the positive and careful review. Your questions on model scope, correlated terms, efficiency, and implementation details helped us sharpen this work. We have added two new analyses to aid in addressing your concerns: **Figure S.6** on cross-seed stability/redundancy and a peak-RSS profiling measurement, noted below. New supplementary figures and tables are referred to with the prefix "**S.**" here, and are anonymously shared at https://pdfhost.io/v/NMx8D9UWRA_figs. These will be included in the camera-ready version.
>
> ### 1. Additive basis-function structure and nested dynamics.
>
> We agree the default IGSR class is more restrictive than fully nested or recurrent formulations. Our target is a sparse interpretable equation $f(\mathbf{x})=\sum_j w_j\psi_j(\mathbf{x})$: terms can be nonlinear, but the final readout is linear in the discovered basis functions. This limitation is noted in the Discussion, and Appendix G.11 probes one adjacent issue via IGSR-TLO. On a proof-of-concept benchmark where the target contains a non-trivial internal constant, IGSR-TLO is able to recover the exact parametric form in 10/25 runs; vanilla IGSR still finds a reasonable approximation, but, as expected, lacks a direct mechanism to tune such internal constants. Across the six main datasets, TLO improves on Lung Cancer and Lung Cancer (with Chemo.), is comparable on COVID-19 and Warfarin PK, and is slightly worse on Lung Cancer (with Chemo. & Radio.) and RNA Polymerase. We therefore view TLO as useful evidence that internal constants can matter, while deeply nested/recurrent forms remain outside the default scope.
>
> ### 2. No-refit influence under correlated terms.
>
> We agree this is important. Appendix G.10 compares the default no-refit score with full-refit and efficient refit-aware variants on all six datasets. The test-MSE differences are not statistically significant on any dataset (repeated-measures ANOVA, all $p\geq 0.09$), while the refit-aware variants are slower. This is why we keep the no-refit default: it gives the relevant marginal signal in the current model context at much lower cost, without measurable held-out loss in these benchmarks.
>
> ### 3. Cost, scalability, and memory.
>
> We agree these points should be stated more concretely. The main results use heuristic MCTS, not deep rollouts: immediate node reward is backpropagated directly, with defaults $n_{\text{successors}}=5$, $K=6$, and budget 30 node expansions. Empirically, runtime on RNA Polymerase grows only about 25% when the feature count increases from 10 to 263; prompt simplification reduces token usage from $\approx 1.68$M to $\approx 1.40\times 10^5$ and wall-clock from 296.5s to 112.0s; and under a fixed $\approx 265$k-token budget IGSR attains 0.016 test MSE versus 2.66 for the next-best LLM baseline. With $B$ concurrent branches/rollouts and $M$ candidate terms, the additional search memory scales roughly as $O(B((N_{\text{train}}+N_{\text{val}})M+M^2))$; in practice $M$ remains small (about 11 terms before pruning, $K=6$ after), so memory grows roughly linearly with the number of concurrent branches rather than exploding with feature count. We have additionally *measured peak-RSS on the largest dataset*, RNA Polymerase, finding it at a reasonable $2.782\pm0.002$ GB (over 5 seeds), empirically confirming that memory usage is not a prohibitive concern in practice.
>
> ### 4. Equivalent expressions and greedy pruning of interactions.
>
> We agree these implementation details should be clearer. The current implementation does not do explicit search-time de-duplication/canonicalization before influence evaluation. Instead, invalid expressions are rejected at execution time, and redundant or near-equivalent candidates can often be reduced downstream by low marginal influence and pruning; stochastic branching also allows related alternatives to reappear later. In the added rebuttal analysis, **Figure S.6** shows that after post hoc canonicalization, 11 unique canonical terms remain across 25 LC-CR seeds, and 12/25 seeds recover the same 6 ground-truth terms; the remaining variation is concentrated in near-equivalent approximations of one Gompertz nonlinearity. On synergy, IGSR prunes proposed basis functions, rather than raw variables: if an interaction is proposed directly, it is evaluated and can survive as a single unit, and MCTS keeps multiple branches alive. We agree, however, that there is no formal safeguard for purely joint-only effects if the interaction is never proposed; synergy-aware/group-retention mechanisms would be valuable future work.
>
> We hope that these clarifications, together with the added efficiency/memory analysis and **Figure S.6**, address your concerns regarding model scope, pruning, and scalability, and would justify an improved score. We sincerely thank you for helping us improve this work and would be very happy to engage in any further discussion.

---

> > ### Author Rebuttal · Reviewer_ZJYk · 2026-04-03
> >
> > Thank you for the comprehensive rebuttal; however, I will keep my current score as some methodological limitations are still present.
> >
> > I thank the authors for the comprehensive rebuttal and the additional empirical validations, which resolve my previous methodological concerns. I have no further comments.

---

> > > ### Author Response · Authors · 2026-04-05
> > >
> > > Thank you again for the positive and careful discussion. Your comments helped us sharpen the methodological picture, and they motivated **two additional targeted analyses** that strengthen the paper. The new supplementary figures are shared anonymously at https://figs2.hostmora.app/.
> > >
> > > The manuscript already frames default IGSR as searching for sparse interpretable equations of the form $f(\mathbf{x})=\sum_j w_j\psi_j(\mathbf{x})$, rather than arbitrary deeply nested or recurrent dynamics, and within that intended scope the method performs strongly across diverse settings. Empirically, this includes 6 benchmark datasets, 128 LLM-SRBench discovery problems under a fixed token budget (including Biology-domain problems), and a prospective wet-lab validation case study. Appendix G.8 supports this point further: on synthetic models with trigonometric forcing, a modified denominator inside the growth term, and a multiplicative interaction $C(t)d(t)$, IGSR consistently outperforms scalar-feedback LLM baselines. We will make this scope framing even clearer in the camera-ready version.
> > >
> > > ### Collinearity Stress Test
> > >
> > > [**Figure S.7.1**](https://figs2.hostmora.app/) shows that default no-refit pruning remained fully stable even under extreme multicollinearity, recovering all signal groups in every seed up to $\rho=0.999$. We tested this on Lung Cancer (with Chemo. & Radio.) using a 14-term pool with the 6 ground-truth terms ($x$, $x\log x$, $xC$, $xd$, $C$, $u^c$), 4 highly correlated clones, and 4 distractors. Clone correlations were set to $\rho\in\{0.95,0.99,0.999\}$. Each term/clone pair counted as one signal group (e.g., $\{C,C_\rho\}$ and $\{xC,xC_\rho\}$), so full recovery means the retained top-$K$ terms cover all 6 groups.
> > >
> > > | $\rho$ | Default no-refit signal-group recall |
> > > | --- | --- |
> > > | 0.95 | 6/6 groups in 15/15 seeds |
> > > | 0.99 | 6/6 groups in 15/15 seeds |
> > > | 0.999 | 6/6 groups in 15/15 seeds |
> > >
> > > Across all 15 seeds and all three $\rho$ values, the default no-refit score recovered all 6 groups with *zero* duplicate-group retentions. The refit-aware variants from Appendix G.10 gave the same selections, but were slower (0.03s no-refit, 0.05s efficient refit-aware, 0.20s full refit; per seed). This addresses the correlated-term concern directly: even under severe multicollinearity, default pruning did not lose true signal groups or over-select redundant clones.
> > >
> > > ### Interaction-Only Pruning Test
> > >
> > > [**Figure S.7.2**](https://figs2.hostmora.app/) shows that default pruning retained epistasis-like interaction terms in every seed, even when their constituent variables had zero marginal linear signal. We used i.i.d. Gaussian inputs and targets whose signal resided only in interaction terms, and considered (i) $y=x_1x_2+\varepsilon$ with $K=1$ from a 10-term pool, and (ii) $y=x_1x_2+x_3x_4+\varepsilon$ with $K=2$ from an 11-term pool.
> > >
> > > | Setting | Default no-refit retention |
> > > | --- | --- |
> > > | $y=x_1x_2+\varepsilon$, $K=1$ | $x_1x_2$ retained in 15/15 seeds |
> > > | $y=x_1x_2+x_3x_4+\varepsilon$, $K=2$ | both true interactions retained in 15/15 seeds |
> > >
> > > Refit-aware variants again gave identical selections, and the marginal variables themselves received near-zero influence ($\sim10^{-7}$ to $10^{-18}$). In this controlled interaction-only setting, the default pruning rule preserved the epistasis-like interaction terms once proposed, rather than discarding them because their constituents were individually weak.
> > >
> > > The added evidence addresses the methodological concerns point by point: Appendix G.8 supports the adequacy of the chosen model class within its intended scope, **Figure S.7.1** addresses correlated-term pruning, and **Figure S.7.2** addresses synergy / epistasis-like pruning. Taken together, this suggests that the linear-in-basis form $f(\mathbf{x})=\sum_j w_j\psi_j(\mathbf{x})$ is best understood as the modeling scope of the current paper, while the pruning mechanism within that scope is empirically reliable rather than a fundamental limitation.
> > >
> > > We sincerely thank you for the constructive and insightful discussion. Your feedback helped us run additional analyses that strengthened the paper, and these would be included in the camera-ready version.
> > >
> > > ---
> > >
> > > The previously shared PDF supplementary document appears to have expired, and we have anonymously reuploaded it at:
> > > https://figs1.hostmora.app/ (Supplementary Figures S.1-S.6).
> > >
> > > The additional two supplementary figures for this response comment are shared anonymously at:
> > > https://figs2.hostmora.app/ (Supplementary Figures S.7.1 and S.7.2).

---

### Decision · Program_Chairs · 2026-04-30

**Decision:**

Accept (regular)

**Comment:**

The authors introduce a method for equation discovery (e.g., discovering equations that explain observed phenomena). They score LLM generated candidates (sparse basis functions for a linear model), and use this scoring mechanism to drive an MCTS search.

The reviewers requested a few additional baselines , had some questions about overfitting and stability, and had some questions about he model assumptions (e.g., linear vs nonlinear models). These requests were met by including new benchmarks, a nested validation experiment, and clarifying that a major driving goal was interpretability of the final equations, which of course makes sense if our goal is to model physical phenomena through simple equations rather than, for example, learned functions.

Ultimately, all reviewers agreed that the authors fully addressed their concerns.